# Effects of Dropout on Performance in Long-range Graph Learning Tasks

**Jasraj Singh**[1][*]  **Keyue Jiang**[2][†]  **Brooks Paige**[2]  **Laura Toni**[2]
[1]Nanyang Technological University  [2]University College London

## Abstract

Message Passing Neural Networks (MPNNs) are a class of Graph Neural Networks (GNNs) that propagate information across the graph via local neighborhoods. The scheme gives rise to two key challenges: *over-smoothing* and *over-squashing*. While several Dropout-style algorithms, such as DropEdge and DropMessage, have successfully addressed over-smoothing, their impact on over-squashing remains largely unexplored. This represents a critical gap in the literature, as failure to mitigate over-squashing would make these methods unsuitable for long-range tasks – the intended use case of deep MPNNs. In this work, we study the aforementioned algorithms, and closely related edge-dropping algorithms – DropNode, DropAgg and DropGNN – in the context of over-squashing. We present theoretical results showing that DropEdge-variants reduce sensitivity between distant nodes, limiting their suitability for long-range tasks. To address this, we introduce DropSens, a sensitivity-aware variant of DropEdge, which is developed following the message-passing scheme of GCN. DropSens explicitly controls the proportion of information lost due to edge-dropping, thereby increasing sensitivity to distant nodes despite dropping the same number of edges. Our experiments on long-range synthetic and real-world datasets confirm the predicted limitations of existing edge-dropping and feature-dropping methods. Moreover, DropSens with GCN consistently outperforms graph rewiring techniques designed to mitigate over-squashing, suggesting that simple, targeted modifications can substantially improve a model's ability to capture long-range interactions. Our conclusions highlight the need to re-evaluate and re-design existing methods for training deep GNNs, with a renewed focus on modelling long-range interactions. The code for reproducing the results in our this work is available at https://github.com/ignasa007/Dropout-Effects-GNNs.

## 1 Introduction

*Graph neural networks* (GNNs) [51, 73] are powerful neural models developed for modelling graph-structured data, and have found applications in several real-world scenarios [29, 61, 84, 90–94, 96, 101]. A popular class of GNNs, called *message-passing neural networks* (MPNNs) [32], recursively process neighborhood information using message-passing layers. These layers are stacked to allow each node to aggregate information from increasingly larger neighborhoods, akin to how *convolutional neural networks* (CNNs) learn hierarchical features for images [48]. However, unlike in image-based deep learning, where *ultra-deep* CNN architectures have led to performance breakthroughs [38, 80], shallow GNNs often outperform deeper models on many graph learning

---

[*]Part of the work done as a master's student at UCL. [†]Primary supervisor. Correspondence to Jasraj Singh <`jasraj.singh00150@gmail.com`> and Keyue Jiang <`keyue.jiang.18@ucl.ac.uk`>.

39th Conference on Neural Information Processing Systems (NeurIPS 2025).

tasks [99]. This is because deep GNNs suffer from unique issues like *over-smoothing* [65] and *over-squashing* [4], which makes training them notoriously difficult.

Over-smoothing refers to the problem of node representations becoming *too similar* as they are recursively processed. This is undesirable since it limits the GNN from effectively utilizing the information in the input features. The problem has garnered significant attention from the research community, resulting in a suite of algorithms designed to address it [72] (see Appendix A.1 for an overview of representative methods). Amongst these methods are a collection of random edge-dropping algorithms, including DropEdge [70], DropNode [23], DropAgg [43] and DropGNN [66] – which we will collectively refer to as *DropEdge-variants* – which act as *message-passing reducers*. In addition, we have DropMessage [21], which performs Dropout [79] on the message matrices, instead of the feature matrices; we will collectively refer to these two methods as *Dropout-variants* since they are applied along the feature dimensions.

The other issue specific to GNNs is over-squashing. In certain graph structures, neighborhood size grows exponentially with distance from the source [12], causing information to be lost as it passes through graph bottlenecks [4]. This limits MPNNs' ability to enable communication between distant nodes, which is crucial for good performance on long-range tasks [52] . To alleviate over-squashing, several graph-rewiring techniques have been proposed, which aim to improve graph connectivity by adding edges in a strategic manner [4, 8, 16, 44, 64] (see Appendix A.3 for an overview of representative methods).[2] In contrast, the DropEdge-variants only remove edges, which should, in principle, amplify over-squashing levels. The same can be intuitively argued about Dropout-variants.

Empirical evidence in support of methods designed for training deep GNNs has been majorly collected on short-range tasks (see Appendix A.2 for a detailed discussion). That is, it simply suggests that *these methods prevent loss of local information, but it remains inconclusive if they facilitate capturing long-range interactions* (LRIs). Of course, on long-range tasks, deep GNNs are useless if they cannot capture LRIs. This is especially a concern for DropEdge-variants since evidence suggests that alleviating over-smoothing with graph rewiring could exacerbate over-squashing [34, 64].

**Contributions.** In this work, we precisely characterize the effects of random edge-dropping algorithms on over-squashing in MPNNs. By explicitly computing the expected *sensitivity* of the node representations to the node features [81] (inversely related to over-squashing) in a linear Graph Convolutional Network (GCN) [47], we show that these methods provably reduce the *effective receptive field* of the model. Precisely speaking, the rate at which sensitivity between nodes decays is *exponential* w.r.t. the distance between them. We also extend the existing theoretical results on sensitivity in nonlinear MPNNs [8, 18, 86] to the random edge-dropping setting, concluding that these algorithms exacerbate the over-squashing problem. We use our analysis of GCNs to design a sensitivity-aware DropEdge-variant, named *DropSens*, that enjoys the representational expressivity of DropEdge without suffering from over-squashing, thereby demonstrating how algorithms can be readily adapted for long-range tasks.

We evaluated the DropEdge- and Dropout-variants on long-range datasets using GCN, Graph Isomorphism Network (GIN) [87] and Graph Attention Network (GAT) [83] architectures. Specifically, we follow the setup in [33] with the SyntheticZINC dataset, in [81] with real-world homophilic (corresponding to short-range tasks) and heterophilic (long-range tasks) node classification datasets, and in [8, 44] with graph classification datasets. Our results indicate that while the random dropping methods improve model performance in short-range tasks, they are often ineffective, and sometimes even detrimental, to long-range task performance. Finally, we present results for DropSens, which outperforms state-of-the-art graph rewiring methods aimed at addressing over-squashing at node classification and graph-classification tasks. These findings point to the importance of re-evaluating the methods used to train deep GNNs, especially in terms of how well they capture LRIs.

## 2   Background

Consider a directed graph $\mathcal{G} = (\mathcal{V}, \mathcal{E})$, with $\mathcal{V} = [N] := \{1, \ldots, N\}$ denoting the node set and $\mathcal{E} \subset \mathcal{V} \times \mathcal{V}$ the edge set; $(j \rightarrow i) \in \mathcal{E}$ if there's an edge from node $j$ to node $i$. Let $\boldsymbol{A} \in \{0, 1\}^{N \times N}$ denote its adjacency matrix, such that $\boldsymbol{A}_{ij} = 1$ if and only if $(j \rightarrow i) \in \mathcal{E}$, and let $\boldsymbol{D} := \operatorname{diag}(\boldsymbol{A}\boldsymbol{1}_N)$ denote the in-degree matrix. The geodesic distance, $d_{\mathcal{G}}(j, i)$, from node $j$ to node $i$ is the length

---

[2]Sometimes, along with removal of some edges to preserve statistical properties of the original topology.

of the shortest path starting at node $j$ and ending at node $i$. Accordingly, the $\ell$-hop neighborhood of a node $i$ can be defined as the set of nodes that can reach it in exactly $\ell \in \mathbb{N}_0$ steps, $\mathbb{S}^{(\ell)}(i) = \{j \in \mathcal{V} : d_{\mathcal{G}}(j, i) = \ell\}$.

## 2.1 Graph Neural Networks

Graph Neural Networks (GNNs) operate on inputs of the form $(\mathcal{G}, \boldsymbol{X})$, where $\mathcal{G}$ encodes the graph topology and $\boldsymbol{X} \in \mathbb{R}^{N \times H^{(0)}}$ collects the node features.[3] Message-Passing Neural Networks (MPNNs) [32] are a special class of GNNs which recursively aggregate information from the 1-hop neighborhood of each node using *message-passing layers*. An L-layer MPNN is given as

$$
\begin{aligned}
\boldsymbol{z}_i^{(\ell)} &= \mathsf{Upd}^{(\ell)}\left(\boldsymbol{z}_i^{(\ell-1)}, \mathsf{Agg}^{(\ell)}\left(\boldsymbol{z}_i^{(\ell-1)}, \left\{\boldsymbol{z}_j^{(\ell-1)} : j \in \mathbb{S}^{(1)}(i)\right\}\right)\right), \quad \forall \ell \in [L] \\
\mathsf{MPNN}_\theta(\mathcal{G}, \boldsymbol{X}) &= \mathsf{Out}\left(\left\{\boldsymbol{z}_i^{(L)} : i \in \mathcal{V}\right\}\right)
\end{aligned}
\tag{2.1}
$$

where $\boldsymbol{Z}^{(0)} = \boldsymbol{X}$, $\mathsf{Agg}^{(\ell)}$ denotes the *aggregation functions*, $\mathsf{Upd}^{(\ell)}$ the *update functions*, and $\mathsf{Out}$ the *readout function*. Since $\boldsymbol{z}_i^{(L)}$ is a function of the input features of nodes at most L-hops away from it, its *receptive field* is given by $\mathbb{B}^{(L)}(i) := \{j \in \mathcal{V} : d_{\mathcal{G}}(j, i) \leq L\}$.

For example, a GCN [47] updates node representations as the weighted sum of its neighbors' representations:

$$
\boldsymbol{Z}^{(\ell)} = \sigma\left(\hat{\boldsymbol{A}} \boldsymbol{Z}^{(\ell-1)} \boldsymbol{W}^{(\ell)}\right)
\tag{2.2}
$$

where $\sigma$ is a point-wise nonlinearity, e.g. ReLU, the propagation matrix, $\hat{\boldsymbol{A}}$, is a *graph shift operator*, i.e. $\hat{\boldsymbol{A}}_{ij} \neq 0$ if and only if $(j \to i) \in \mathcal{E}$ or $i = j$, and $\boldsymbol{W}^{(\ell)} \in \mathbb{R}^{H^{(\ell-1)} \times H^{(\ell)}}$ is a weight matrix. The original choice for $\hat{\boldsymbol{A}}$ was the symmetrically normalized adjacency matrix $\hat{\boldsymbol{A}}^{\mathrm{sym}} := \widetilde{\boldsymbol{D}}^{-1/2} \widetilde{\boldsymbol{A}} \widetilde{\boldsymbol{D}}^{-1/2}$ [47], where $\widetilde{\boldsymbol{A}} = \boldsymbol{A} + \boldsymbol{I}_N$ and $\widetilde{\boldsymbol{D}} = \mathrm{diag}(\widetilde{\boldsymbol{A}} \boldsymbol{1}_N)$. However, several influential works have also used the asymmetrically normalized adjacency, $\hat{\boldsymbol{A}}^{\mathrm{asym}} := \widetilde{\boldsymbol{D}}^{-1} \widetilde{\boldsymbol{A}}$ [36, 50, 74].

## 2.2 DropEdge-variants

DropEdge [70] is a random data augmentation technique that works by sampling a subgraph of the input graph in each layer, followed by the addition of self-loops, and uses that for message passing. Several variants of DropEdge have also been proposed, forming a family of random edge-dropping algorithms for tackling the over-smoothing problem. For example, DropNode [23] independently samples nodes and sets their features to $\boldsymbol{0}$, followed by rescaling to make the feature matrix unbiased. This is equivalent to setting the corresponding columns of the propagation matrix to $\boldsymbol{0}$. In a similar vein, DropAgg [43] samples nodes that don't aggregate messages from their neighbors. This is equivalent to dropping the corresponding rows of the adjacency matrix. Combining these two approaches, DropGNN [66] samples nodes which neither propagate nor aggregate messages in a given layer. These algorithms alleviate over-smoothing by reducing the number of messages being propagated in the graph, thereby slowing down the convergence of node representations.

## 2.3 Dropout-variants

Dropout is a stochastic regularization technique which reduces over-fitting by randomly dropping features before each layer. It has been successful with various architectures, like CNNs [79] and transformers [82], and has also found applications in GNN training. DropMessage [21] is a variant of Dropout designed specifically for message-passing schemes – it acts directly on the messages over each edge, instead of the node representations. This reduces the induced variance in the messages compared to Dropout, DropEdge and DropNode, while at the same time making the method more effective at alleviating over-smoothing and enabling the training of deep GNNs.

---

[3]To keep things simple, we will ignore edge features.

## 2.4 Over-squashing

Over-squashing refers to the problem of information from exponentially growing neighborhoods [11] being squashed into finite-sized node representations [4]. [81] formally characterized over-squashing in terms of the Jacobian of the node-level representations w.r.t. the input features: $\|\partial \boldsymbol{z}_i^{(L)}/\partial \boldsymbol{x}_j\|_1$. Accordingly, over-squashing can be understood as low sensitivity between distant nodes, i.e. small perturbations in a node's features don't effect other distant nodes' representations.

See Appendix A for an extensive discussion of related works addressing the problems of over-smoothing and over-squashing, and a unified treatment of the two.

## 3 Sensitivity Analysis

In this section, we perform a theoretical analysis of the expectation – w.r.t. random edge masks – of sensitivity of node representations. This will allow us to predict how DropEdge-variants affect communication between nodes at various distances, which is relevant for predicting their suitability towards learning LRIs.

Here, we present our analysis for linear GCNs, and treat more general nonlinear MPNN architectures in Appendix C.1. In this model, the final node representations can be summarised as

$$\boldsymbol{Z}^{(L)} = \left( \prod_{\ell=1}^{L} \hat{\boldsymbol{A}}^{(\ell)} \right) \boldsymbol{X} \boldsymbol{W} \in \mathbb{R}^{N \times H^{(L)}} \tag{3.1}$$

where $\boldsymbol{W} \coloneqq \prod_{\ell=1}^{L} \boldsymbol{W}^{(\ell)} \in \mathbb{R}^{H^{(0)} \times H^{(L)}}$. Using the i.i.d. assumption on the distribution of edge masks in each layer, $\{\mathbf{M}^{(\ell)}\}_{\ell=1}^{L}$, the expected sensitivity of node $i$ to node $j$ can be shown to be

$$\mathbb{E}_{\mathbf{M}^{(1)}, \ldots, \mathbf{M}^{(L)}} \left[ \left\| \frac{\partial \boldsymbol{z}_i^{(L)}}{\partial \boldsymbol{x}_j} \right\|_1 \right] = \left( \mathbb{E} \left[ \hat{\boldsymbol{A}} \right]^L \right)_{ij} \|\boldsymbol{W}\|_1 \tag{3.2}$$

To keep things simple, we will ignore the effect of DropEdge-variants on the optimization trajectory. Accordingly, it is sufficient to study $\mathbb{E}[\hat{\boldsymbol{A}}]$ in order to predict their effect on over-squashing. To maintain analytical tractability, we assume the use of an asymmetrically normalized adjacency matrix for message-passing, $\hat{\boldsymbol{A}} = \hat{\boldsymbol{A}}^{\text{asym}}$.

**Lemma 3.1.** *The expected propagation matrix under DropEdge* (DE) *is given as:*

$$\begin{aligned}
\dot{\boldsymbol{P}}_{ii} &\coloneqq \mathbb{E}_{\text{DE}} \left[ \hat{\boldsymbol{A}}_{ii} \right] = \frac{1 - q^{d_i + 1}}{(1 - q)(d_i + 1)} \\
\dot{\boldsymbol{P}}_{ij} &\coloneqq \mathbb{E}_{\text{DE}} \left[ \hat{\boldsymbol{A}}_{ij} \right] = \frac{1}{d_i} \left( 1 - \frac{1 - q^{d_i + 1}}{(1 - q)(d_i + 1)} \right)
\end{aligned} \tag{3.3}$$

*where $q \in [0, 1)$ is the dropping probability and $(j \to i) \in \mathcal{E}$.[4]*

See Appendix B.1 for a proof, and a similar treatment of DropNode, DropAgg and DropGNN.

**1-Layer Linear GCNs.** $\forall q \in (0, 1)$ and nodes $i, j$ such that $(j \to i) \in \mathcal{E}$, we have

$$\begin{aligned}
\dot{\boldsymbol{P}}_{ii} &= \frac{1}{d_i + 1} \sum_{k=0}^{d_i} q^k > \frac{1}{d_i + 1} \\
\dot{\boldsymbol{P}}_{ij} &= \frac{1}{d_i} \left( 1 - \dot{\boldsymbol{P}}_{ii} \right) < \frac{1}{d_i + 1}
\end{aligned} \tag{3.4}$$

where the right-hand sides of the two inequalities are the corresponding entries in the propagation matrix of a NoDrop model. Equations 3.3, 3.4, B.16 and B.17 together imply the following result:

---

[4]We use $\dot{\boldsymbol{P}}$ to denote the expected propagation matrix under the asymmetric propagation rule. In Appendix D.2, we use $\ddot{\boldsymbol{P}}$ to denote the expected propagation matrix under the symmetric propagation rule.

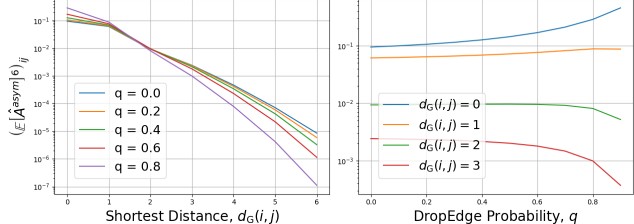 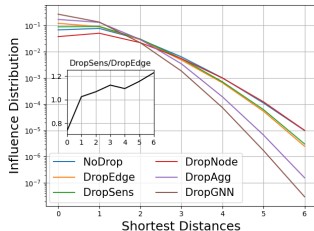

**(a)** Entries of $\dot{\boldsymbol{P}}^6$ decay at exponential rate w.r.t. distance between nodes, and polynomial rate w.r.t. to the DropEdge probability.

**(b)** MC-approximation of influence distribution in ReLU-GCNs.

**Figure 1:** Empirical sensitivity analysis using the Cora dataset. The plot embedded in Figure 1b shows the ratio of influence under DropSens to that under DropEdge, at different distances.

**Lemma 3.2.** *In a 1-layer linear GCN with $\hat{\boldsymbol{A}} = \hat{\boldsymbol{A}}^{asym}$, using DropEdge, DropAgg or DropGNN*

1. *increases the sensitivity of a node's representations to its own input features, and*

2. *decreases the sensitivity to its neighbors' features.*

**L-layer Linear GCNs.** Unfortunately, we cannot draw similar conclusions in L-layer networks, for nodes at arbitrary distances. To see this, view $\dot{\boldsymbol{P}}$ as the transition matrix of a non-uniform random walk. This walk has higher self-transition ($i = j$) probabilities than in a uniform augmented random walk ($\boldsymbol{T} = \hat{\boldsymbol{A}}^{\mathrm{asym}}, q = 0$), but lower inter-node ($i \neq j$) transition probabilities. Note that $\dot{\boldsymbol{P}}^L$ and $\boldsymbol{T}^L$ store the L-step transition probabilities in the corresponding walks. Then, since the paths connecting the nodes $i \in \mathcal{V}$ and $j \in \mathbb{B}^{(L-1)}(i)$ may involve self-loops, $(\dot{\boldsymbol{P}}^L)_{ij}$ may be lower or higher than $(\boldsymbol{T}^L)_{ij}$. Therefore, we cannot conclude how sensitivity between nodes separated by at most $L - 1$ hops changes. For nodes L-hops away, however, we can show that DropEdge always decreases the corresponding entry in $\dot{\boldsymbol{P}}^L$, reducing the effective reachability of GCNs. Using Equations B.16 and B.17, we can show the same for DropAgg and DropGNN, respectively.

**Theorem 3.1.** *In an L-layer linear GCN with $\hat{\boldsymbol{A}} = \hat{\boldsymbol{A}}^{asym}$, using DropEdge, DropAgg or DropGNN decreases the sensitivity of a node $i \in \mathcal{V}$ to another node $j \in \mathbb{S}^{(L)}(i)$, thereby reducing its effective receptive field. Moreover, the sensitivity decreases with increasing dropping probability.*

See Appendix B.2 for a precise quantitative statement and the proof.

**Nodes at Arbitrary Distances.** Although no general statement could be made about the change in sensitivity between nodes up to $L-1$ hops away, we can analyze such pairs empirically. We compute the L-hop transition matrix $\dot{\boldsymbol{P}}^L$ – proportional to expected sensitivity in linear GCNs under DropEdge – for the Cora dataset, and average the entries after binning node pairs by the shortest distance between them. The results are shown in Figure 1a. In the left subfigure, we observe that the expected sensitivity decays at an *exponential rate* with increasing distance between the corresponding nodes. In the middle subfigure, we observe that DropEdge increases the expected sensitivity between nodes close to each other (0-hop and 1-hop neighbors) in the original topology, but reduces it between nodes farther off. Similar conclusions can be made with the symmetrically normalized propagation matrix (see Appendix D.2). Note that the over-squashing effects of DropAgg and DropGNN would, in theory, be even more severe, as suggested by Equations B.16 and B.17.

**Nonlinear MPNNs.** While linear networks are useful in simplifying the theoretical analysis, they are often not practical. In Appendix C.1, we treat the upper bounds on sensitivity established in previous works, and extend them to the DropEdge-variants. Even still, although theoretical bounds offer valuable guarantees, they can be arbitrarily loose in the absence of error quantification, making their practical relevance unclear. To reliably conclude the empirical behaviour of DropEdge- and Dropout-variants, we turn to Monte Carlo simulations with ReLU-GCNs; see Appendix D.1 for a description of the experiment setup. Figure 1b compares the influence of the source nodes [86] at different distances using a dropout probability of $0.5$. We observe that while the effect of DropNode on the sensitivity profile – as compared to the baseline NoDrop – is relatively insignificant, models

using DropEdge, DropAgg and DropGNN have remarkably lower sensitivity to distant nodes, as predicted by our theory.

## 4 Sensitivity-Aware DropEdge

Lemma 3.1 tells us that DropEdge decreases the weight of cross-edges, $(j \rightarrow i)$, in the expected propagation matrix, i.e. the *strength of message passing* over these edges decreases. The fraction of *information preserved* over a cross-edge is dependent only on the dropping probability and the target node's in-degree, $d_i$. We can directly control this quantity using a per-edge dropping probability, $q_i$, dependent only on the receiving node's in-degree:

$$c := \frac{\mathbb{E}_{\mathsf{DE}}[\hat{A}_{ij}]}{\mathbb{E}_{\mathsf{ND}}[\hat{A}_{ij}]} = \frac{d_i + 1}{d_i} \left( 1 - \frac{1 - q_i^{d_i+1}}{(1 - q_i)(d_i + 1)} \right) \implies 1 - c = \frac{q_i - q_i^{d_i+1}}{d_i (1 - q_i)} \qquad (4.1)$$

where $c$ is the fraction of information preserved, e.g. 95%, and ND refers to a NoDrop model. We can solve for $q_i$ and mask the incoming edges to node $i$ accordingly; we name this algorithm *DropSens*. In Appendix E.3, we present a `Python` implementation of the algorithm, as well as a computationally efficient approximation to Equation 4.1. In the plot embedded in Figure 1b, we can observe that DropSens increases the influence of distant nodes, as compared to DropEdge.

Note that higher values of $c$ encourage lower $q_i$, while lower values permit $q_i$ to take higher values; see Figure 7a. Since this can result in abnormally high dropping probabilities, we clip the value of $q_i$ in our experiments using another hyperparameter, $q_{\max}$; see details in Appendix E.2.

**Challenges in Extending to GIN and GAT.** DropSens has been specifically derived for GCN-style architectures, where the edge weights used in message aggregation are simple functions of node degree. Extending it to other architectures raises nontrivial challenges:

1. GIN uses constant edge weights, so the sensitivity of a node to its neighbors simplifies to $(1 - q)\|W\|$ for dropping probability $q$. Enforcing a fixed information preservation ratio $c$ gives $q = 1 - c$, which is equivalent to DropEdge. This limitation arises because GIN's neighbour-weighing scheme is independent of the node degrees, making a principled variant of DropSens unnecessary.

2. GAT, in contrast, uses feature-dependent attention weights that vary at every layer and iteration. This means a DropSens-style approach would require recomputing edge masks in each iteration and each layer, compromising the simplicity we aimed for. Moreover, the presence of softmax attention makes the closed-form of sensitivity complicated.

## 5 Experiments

Our theoretical analysis indicates that random dropping may degrade the performance of GNNs in tasks that depend on capturing LRIs. In this section, we test this hypothesis by evaluating DropEdge- and Dropout-variants on both synthetic and real-world datasets. A complete description of the datasets is provided in Appendix E.1, and the experimental details are in Appendix E.2.

### 5.1 Synthetic Datasets

SyntheticZINC [33] is a synthetic variant of the ZINC dataset [42], designed to study the effect of information mixing in graph learning. Node features are sparsely assigned, and the target requires non-linear mixing of two selected nodes' features, chosen based on their commute time [10]. We vary the mixing level and evaluate an 11-layer GCN, ensuring sufficient message passing. For better readability, we only test DropEdge, Dropout and DropMessage – the three more popular methods used for training deep GNNs.

The results are presented in Figure 2, where we can observe that the mean absolute error (MAE) increases with the commute time percentile used to select the node pairs, as was hypothesized and evidenced in [33]. Additionally, we observe that both train and test performance decline when using dropout with a probability as low as 0.2, and even more so with a higher probability of 0.5. These results provide strong evidence for the detrimental effects of dropout methods in modelling long-range interactions, supporting our theoretical analysis.

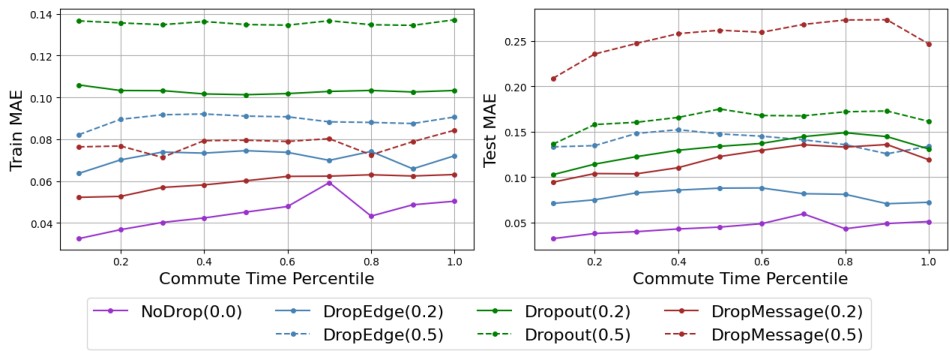

**Figure 2:** Train and test MAE of 11-layer GCNs on the SyntheticZINC dataset, averaged over 10 initializations.

## 5.2 Real-world Datasets

To test the dropping methods on real-world datasets, we use the GCN, GIN [87] and GAT [83] architectures – GCN and GIN satisfy the model assumptions made in all the theoretical results presented in Section 3, while GAT does not satisfy any of them, since the attention scores are computed as a function of *all* the node representations. Therefore, GCN, GIN and GAT together provide a broad representation of different MPNN architectures. We present the results for GCN and GIN in the main text, since these models were used as baselines in a majority of works on alleviating over-squashing [4, 5, 8, 35, 44, 69, 81]; the results for GAT are reported in Table 7.

For each dataset−model−dropout combination, we perform 20 independent runs to find the best performing dropout configuration; results are reported in Table 10. We then perform a t-test to assess whether dropout improves performance, using 50 samples from the NoDrop model ($q = 0$) and 50 samples from the best performing dropout configuration.[5] In this section, we report the p-values of the tests, and in Table 8, we report the effect sizes as Hedges' $g$ statistic [39].

**Node-classification.** Although determining whether a task requires modelling LRIs can be challenging, understanding the structure of the datasets can provide important insight. For example, homophilic datasets have local consistency in node labels, i.e. nodes closely connected to each other have similar labels. On the other hand, in heterophilic datasets, nearby nodes often have dissimilar labels. Since DropEdge-variants increase the sensitivity of a node's representations to its immediate neighbors, and reduce its sensitivity to distant nodes, we expect it to improve performance on homophilic datasets but harm performance on heterophilic ones; such a setup was also used in [81]. In this work, we use Cora [58], CiteSeer [31] and PubMed [63] as representatives of homophilic datasets [53, 100], and Squirrel, Chameleon and TwitchDE [71] to represent heterophilic datasets [53]. The networks' statistics are presented in Table 4, where we can note the remarkably lower homophily measures of heterophilic datasets.

The results are presented in Table 1a. It is clear to see that dropout *can significantly improve* test performance on homophilic datasets – with $31/54 \approx 57\%$ cases performing better than the corresponding NoDrop baseline – indicating that these methods are indeed beneficial in tackling short-range tasks. On the other hand, with the heterophilic datasets, the improvement is *insignificant*. Rather, in most ($46/54 \approx 85\%$) cases, the best dropout configuration performs worse than the NoDrop baseline. This suggests that the dropping methods harm generalization in long-range tasks by forcing models to overfit to short-range signals (see Appendix F.3 for supporting evidence).

**Graph-classification.** Several graph classification datasets have also been identified as long-range tasks, like the molecular networks datasets Mutag [17], Proteins [19] and Enzymes [9], and the social networks datasets Reddit, IMDb and Collab [89]. These datasets have also been used for evaluation in previous works on over-squashing, including [8, 44].

---

[5]The t-test assumes that both samples are drawn from normal distributions – all Shapiro-Wilk tests for non-normality of samples [76] failed at 90% confidence.

**Table 1:** Difference in mean test accuracy (%) between the best performing configuration of each dropout method and the baseline NoDrop model. Cell colors represent p-values from a t-test evaluating whether dropout improves performance: green indicates significance at 90% confidence, while red denotes insignificant results.

**(a)** Node classification tasks.

| GNN | Dropout | Homophilic Networks | | | Heterophilic Networks | | |
|---|---|---|---|---|---|---|---|
| | | Cora | CiteSeer | PubMed | Chameleon | Squirrel | TwitchDE |
| GCN | DropEdge | +0.215 | +0.515 | −0.100 | −1.711 | −0.334 | −0.041 |
| | DropNode | +0.213 | −0.302 | −0.082 | −4.987 | −1.307 | −1.180 |
| | DropAgg | +0.092 | +0.552 | −0.429 | −15.180 | −13.738 | −5.306 |
| | DropGNN | +0.372 | +0.727 | −0.194 | −3.841 | −1.451 | −0.950 |
| | Dropout | +0.418 | −0.262 | +1.111 | −3.672 | −1.298 | −0.516 |
| | DropMessage | +0.140 | +0.092 | +0.820 | −1.501 | −0.549 | +0.270 |
| GIN | DropEdge | −5.370 | −11.911 | −3.977 | −7.814 | −1.503 | +0.724 |
| | DropNode | +3.983 | +4.421 | +1.260 | −2.655 | −0.871 | +0.426 |
| | DropAgg | −2.465 | −11.098 | −2.038 | −7.233 | −4.973 | −0.928 |
| | DropGNN | −0.929 | −10.150 | −2.280 | −9.544 | −5.118 | −0.057 |
| | Dropout | +4.771 | +3.128 | +2.311 | −0.865 | +1.062 | +1.254 |
| | DropMessage | +5.661 | +2.841 | +2.003 | +4.243 | +2.439 | +1.919 |

**(b)** Graph classification tasks.

| GNN | Dropout | Molecular Networks | | | Social Networks | | |
|---|---|---|---|---|---|---|---|
| | | Mutag | Proteins | Enzymes | Reddit | IMDb | Collab |
| GCN | DropEdge | −1.200 | +0.482 | −3.032 | −6.280 | +0.680 | −3.353 |
| | DropNode | +0.000 | +0.875 | −3.083 | −10.930 | −0.320 | −4.511 |
| | DropAgg | +1.100 | −0.286 | −5.424 | −17.450 | +0.540 | −20.940 |
| | DropGNN | +2.600 | +0.482 | −2.900 | −12.940 | −1.180 | +0.880 |
| | Dropout | −2.700 | −0.089 | −4.485 | −0.920 | +0.580 | −3.299 |
| | DropMessage | −0.200 | +0.393 | −3.584 | −3.770 | +3.420 | +2.607 |
| GIN | DropEdge | −2.900 | −1.143 | −8.954 | −0.430 | −0.240 | −0.558 |
| | DropNode | −8.500 | +0.536 | −7.568 | −1.500 | +0.380 | +0.125 |
| | DropAgg | +0.200 | +0.143 | −0.640 | −1.560 | −0.800 | −0.988 |
| | DropGNN | −3.700 | −0.661 | −11.745 | −4.150 | −2.740 | −3.456 |
| | Dropout | +0.600 | +0.661 | −2.042 | +0.390 | +0.080 | +0.383 |
| | DropMessage | +1.100 | +1.536 | −2.406 | −0.130 | +1.340 | +1.506 |

The results are shown in Table 1b, where we observe that dropout methods are usually detrimental to model performance, often recording a non-positive effect ($62/102 \approx 61\%$ cases).[6] Notably, the p-values are lower as compared to those recorded for heterophilic datasets in Table 1a, i.e. higher evidence for efficacy of dropping methods. We conjecture that over-squashing may have limited impact on model performance in graph-level tasks since the aggregation module eventually mixes information from distant nodes for computing graph-level representations.

## 5.3 Evaluating DropSens

We start by comparing DropSens with DropEdge on real-world datasets from Section 5.2, illustrating how algorithms can be readily adapted for better suitability at modelling LRIs. In Figure 3, we present the relative change in error rate ($1 - \text{Acc}$) of DropSens w.r.t. DropEdge, with GCN as the base model. It is clear to observe a uniform improvement in the performance on long-range tasks, suggesting that addressing over-squashing using DropSens can enhance the effectiveness of GCNs.

Finally, we benchmark DropSens against state-of-the-art graph-rewiring techniques designed specifically to tackle over-squashing (see Appendix A.3 for their descriptions). We train a GCN on node classification tasks, following the setup in [81], and both GCN and GIN on graph classification tasks, following [8, 44]. The results with GCN are reported in Table 2, where we find that DropSens

---

[6]We exclude Collab×GAT due to unsuccessful runs.

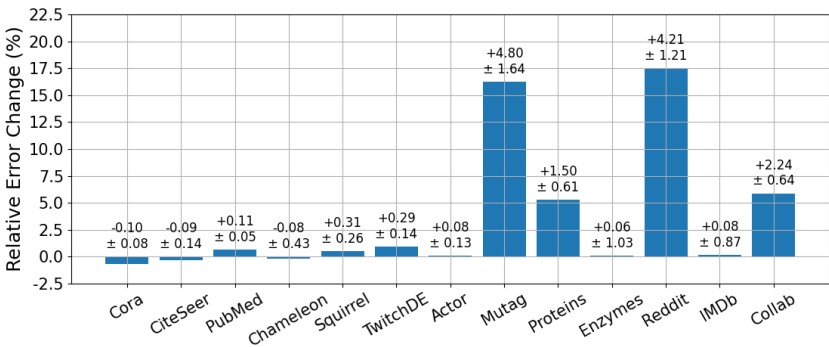

**Figure 3:** Relative change in test-time performance of a GCN using DropSens, compared to the baseline DropEdge, on real-world datasets from Section 5.2. Text over each bar indicates the difference between mean performance of DropSens and DropEdge, and corresponding standard errors.

**Table 2:** Performance of GCN with graph rewiring methods. First, second, and third best results are coloured.

(a) Node-classification tasks – results for other methods taken from [81, Table 2].

| Rewiring | Cora | CiteSeer | PubMed | Chameleon | Squirrel | Actor |
|---|---|---|---|---|---|---|
| None | 81.89 | 72.31 | 78.16 | 41.33 | 30.32 | 23.84 |
| Undirected | - | - | - | 42.02 | 35.53 | 21.45 |
| +FA | 81.65 | 70.47 | 79.48 | 42.67 | 36.86 | 24.14 |
| DIGL (PPR) | 83.21 | 73.29 | 78.84 | 42.02 | 33.22 | 24.77 |
| DIGL + Undirected | - | - | - | 42.68 | 32.48 | 25.45 |
| SDRF | 82.76 | 72.58 | 79.10 | 42.73 | 37.05 | 28.42 |
| SDRF + Undirected | - | - | - | 44.46 | 37.67 | 28.35 |
| DropSens | 85.17 | 73.81 | 83.76 | 53.45 | 39.72 | 22.89 |

(b) Graph-classification tasks – results for other methods from [8, Table 1].

| Rewiring | Mutag | Proteins | Enzymes | Reddit | IMDb | Collab |
|---|---|---|---|---|---|---|
| None | 72.15 | 70.98 | 27.67 | 68.26 | 49.77 | 33.78 |
| Last FA | 70.05 | 71.02 | 26.47 | 68.49 | 48.98 | 33.32 |
| Every FA | 70.45 | 60.04 | 18.33 | 48.49 | 48.17 | 51.80 |
| DIGL | 79.70 | 70.76 | 35.72 | 76.04 | 64.39 | 54.50 |
| SDRF | 71.05 | 70.92 | 28.37 | 68.62 | 49.40 | 33.45 |
| FoSR | 80.00 | 73.42 | 25.07 | 70.33 | 49.66 | 33.84 |
| GTR | 79.10 | 72.59 | 27.52 | 68.99 | 49.92 | 33.05 |
| DropSens | 75.30 | 73.27 | 28.42 | 80.16 | 49.34 | 64.16 |

outperforms other methods in node classification tasks, and performs competitively in graph classification tasks. In addition to superior performance, another advantage of DropSens over the other methods is that it significantly reduces the number of messages being propagated, thereby tackling the problem of over-smoothing and increasing training speed.

The results with GIN are presented in Appendix G.4, where we observe that DropSens does not perform competitively – unsurprising, since DropSens was specifically designed to work with GCN's message-passing scheme.

## 6 Conclusion

There exists an important gap in our understanding of several algorithms designed for training deep GNNs – while their positive effects on model performance have been well-studied, making them popular choices for training deep GNNs, their evaluation has been limited to short-range tasks.

This is rooted in a key assumption: that if a deep GNN is trainable, it must also be capable of modelling LRIs. As a result, potential adverse effects of these algorithms on capturing LRIs have been overlooked. Our results challenge this assumption – we theoretically and empirically show that DropEdge- and Dropout-variants exacerbate the over-squashing problem in deep GNNs, and degrade performance on long-range tasks. This highlights the need for a more comprehensive evaluation of common training practices for deep GNNs, with special emphasis on their capacity to capture LRIs. This is crucial for building confidence in their use beyond controlled benchmarks.

**Limitations.** While our theoretical analysis successfully predicts how DropEdge-variants affect test performance on short-range and long-range tasks, it is based on several simplifying assumptions on the message-passing scheme. These assumptions, although standard in the literature, limit the generalizability of our conclusions to other architectures, including ResGCNs [51], GATs [83], and Graph Transformers [85]. Additionally, an important limitation of DropSens is that it requires an architecture-specific alteration to the edge-dropping strategy, which is not practical in general. As also mentioned in Section 4, we did not intend to introduce DropSens as a benchmark, but rather to demonstrate how methods designed for alleviating over-smoothing can be readily adapted to simultaneously control over-squashing.

**Future Directions.** Currently, real-world datasets are classified as short- or long-range tasks based on extensive model training [4] or weak proxy measures like node homophily [81]. Developing a reliable measure of information mixing in the ground-truth data could greatly benefit the research community. Such a measure would enable more precise identification of short-, intermediate- and long-range tasks, improving evaluation and benchmarking. Another interesting direction is to investigate the significance of over-squashing in graph-level tasks, where the aggregation module of MPNNs enables some mixing of information from distant nodes. To the best of our knowledge, [33] is the only work that directly addresses this question, offering strong theoretical insights. However, empirical validation of these effects remains limited.

# Acknowledgement

Keyue Jiang was supported by the UKRI Engineering and Physical Sciences Research Council (EP-SRC) [grant number EP/R513143/1]. The authors would like to thank the anonymous reviewers for their helpful comments and insights.

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

# Appendix

## Table of Contents

## A  Related Works

### A.1  Methods for Alleviating Over-smoothing

A popular choice for reducing over-smoothing in GNNs is to regularize the model. Recall that DropEdge [70] implicitly regularizes the model by adding noise to it (Section 2.2). A similarly regularization effect is observed with the methods discussed in the main text – DropNode [23], DropAgg [43], DropGNN [66], Dropout [79] and DropMessage [21]. Graph Drop Connect (GDC) [37] combines DropEdge and DropMessage together, resulting in a layer-wise sampling scheme that uses a different subgraph for message-aggregation over each feature dimension. These methods successfully addressed the over-smoothing problem, enabling the training of deep GNNs, and performed competitively on several benchmarking datasets.

**Table 3:** Statistics of node-classification datasets. Homophily measures as defined in [53].

| Dataset | Nodes | Edges | Features | Classes | Homophily |
|---|---|---|---|---|---|
| **Homophilic Networks** | | | | | |
| Reddit | 232,965 | 114,615,892 | 602 | 41 | 0.653 |
| OGBN-ArXiv | 169,343 | 1,166,243 | 128 | 40 | 0.416 |
| Coauthor-CS | 18,333 | 163,788 | 6,805 | 15 | 0.755 |
| Coauthor-Physics | 34,493 | 495,924 | 8,415 | 5 | 0.847 |
| Wiki-CS | 11,701 | 216,123 | 300 | 10 | 0.568 |
| Amazon-Computers | 13,752 | 491,722 | 767 | 10 | 0.700 |
| Amazon-Photo | 7,650 | 238,162 | 745 | 8 | 0.772 |
| **Heterophilic Networks** | | | | | |
| Flickr | 89,250 | 899,756 | 500 | 7 | 0.070 |
| Cornell | 183 | 298 | 1,703 | 5 | 0.031 |
| Texas | 183 | 325 | 1,703 | 5 | 0.001 |
| Wisconsin | 251 | 515 | 1,703 | 5 | 0.094 |

Another powerful form of implicit regularization is feature normalization, which has proven crucial in enhancing the performance and stability of several types of neural networks [41]. Exploiting the inductive bias in graph-structured data, normalization techniques like PairNorm [95], Differentiable Group Normalization (DGN) [97] and NodeNorm [99] have been proposed to reduce oversmoothing in GNNs. On the other hand, Energetic Graph Neural Networks (EGNNs) [98] explicitly regularize the optimization by constraining the layer-wise Dirichlet energy to a predefined range.

In a different vein, motivated by the success of residual networks (ResNets) [38] in computer vision, [49] proposed the use of residual connections to prevent the smoothing of representations. Residual connections successfully improved the performance of GCN on a range of graph-learning tasks. [13] introduced GCN-II, which uses skip connections from the input to all hidden layers. This layer wise propagation rule has allowed for training of ultra-deep networks – up to 64 layers. Some other architectures, like the Jumping Knowledge Network (JKNet) [86] and the Deep Adaptive GNN (DAGNN) [54], aggregate the representations from *all* layers, $\{z_i^{(\ell)}\}_{\ell=1}^{L}$, before processing them through a readout layer.

## A.2 Homophily Bias in Evaluation of Techniques for Deep GNN

We examine the evaluation protocols commonly used for assessing methods aimed at alleviating over-smoothing in deep GNNs – many of which are also widely adopted for training deep architectures. Notably, we highlight a misalignment between the intended goal of these methods – to improve the trainability of deep GNNs – and their evaluation, which is often restricted to short-range tasks.

For example, DropEdge [70] was evaluated on Cora [58], CiteSeer [31], PubMed [63], and a version of Reddit [36] distinct from the one used in our experiments. The first three exhibit high label homophily (see Table 4) and are known to be better modelled by shallower networks [97]. Reddit also displays strong homophily, as can be seen in Table 3. Similarly, DropNode [23] was evaluated on Cora, CiteSeer, and PubMed; DropAgg [43] on Cora ML, CiteSeer, and OGBN-ArXiv [40], which has moderate homophily; DropMessage [70] was evaluated on Cora, CiteSeer, PubMed, OGBN-ArXiv, and Flickr, with only the latter having low homophily; GDC [37] was evaluated on Cora, Cora ML and CiteSeer.

A similar trend can be observed in the evaluation of feature normalization techniques used to regularize GNNs. PairNorm [95] and DGN [97] were evaluated on Cora, CiteSeer, PubMed, and Coauthor-CS [77]; NodeNorm [99] on Cora, CiteSeer, PubMed, Coauthor-CS, Wiki-CS [59], and Amazon-Photo [77]; and EGNNs [98] on Cora, PubMed, Coauthor-Physics [77], and OGBN-ArXiv – all of these datasets are highly homophilic.

A similar trend is observed in the evaluation of architectural modifications designed to enable deeper GNNs. GCN-II [13] on Cora, CiteSeer, PubMed, and Chameleon; JKNet [86] on Cora, CiteSeer, and Reddit; and DAGNN [54] on Cora, CiteSeer, PubMed, Coauthor-CS, Coauthor-Physics, Amazon-Computers [77], and Amazon-Photo – many of these datasets are highly homophilic as well.

This pattern indicates that an overwhelming proportion of evaluations have been restricted to short-range, homophilic tasks. Such a narrow focus risks overstating the general effectiveness of these methods and masking their potential limitations in long-range scenarios.

A few exceptions stand out. DropGNN [66], which was evaluated on graph-classification from the TUDataset [62], aligning more closely with evaluations of rewiring methods targeting over-squashing [8, 44]. NodeNorm, while primarily evaluated on homophilic datasets, was also tested on three heterophilic graphs: Cornell, Texas, and Wisconsin [68]. GCN-II saw broader evaluation, including on several long-range tasks such as Chameleon, Cornell, Texas, Wisconsin, and the Protein-Protein Interaction (PPI) networks [36]. Lastly, JKNet was also evaluated on the PPI networks.

### A.3 Methods for Alleviating Over-squashing

In this section, we will review some of the graph rewiring methods proposed to address the problem of over-squashing. Particularly, we wish to emphasize a commonality among these methods – edge addition is necessary. As a reminder, *graph rewiring* refers to modifying the edge set of a graph by adding and/or removing edges in a systematic manner. In a special case, which includes many of the rewiring techniques we will discuss, the original topology is completely discarded, and only the rewired graph is used for message-passing.

*Spatial* rewiring methods use the topological relationships between the nodes in order to come up with a rewiring strategy. That is the graph rewiring is guided by the objective of optimizing some chosen topological properties. For instance, [4] introduced a fully-adjacent (FA) layer, wherein messages are passed between all nodes. GNNs using a FA layer in the final message-passing step were shown to outperform the baselines on a variety of long-range tasks, revealing the importance of information exchange between far-off nodes which standard message-passing cannot facilitate. [81] proposed a curvature-based rewiring strategy, called the Stochastic Discrete Ricci Flow (SDRF), which aims to reduce the "bottleneckedness" of a graph by adding suitable edges, while simultaneously removing edges in an effort to preserve the statistical properties of the original topology. [8] proposed the Greedy Total Resistance (GTR) technique, which optimizes the graph's total resistance by greedily adding edges to achieve the greatest improvement. One concern with graph rewiring methods is that unmoderated densification of the graph, e.g. using a fully connected graph for propagating messages, can result in a loss of the inductive bias the topology provides, potentially leading to over-fitting. Accordingly, [35] propose a Dynamically Rewired (DRew) message-passing framework that gradually *densifies* the graph. Specifically, in a given layer $\ell$, node $i$ aggregates messages from its entire $\ell$-hop receptive field instead of just the immediate neighbors. This results in an improved communication over long distances while also retaining the inductive bias of the shortest distance between nodes.

*Spectral* methods, on the other hand, use the spectral properties of the matrices encoding the graph topology, e.g. the adjacency or the Laplacian matrix, to design rewiring algorithms. For example, [5] proposed a differentiable graph rewiring layer based on the Lovász bound [56, Corollary 3.3]. Similarly, [6] introduced the Random Local Edge Flip (RLEF) algorithm, which draws inspiration from the "Flip Markov Chain" [22, 57] – a sequence of such steps can convert a connected graph into an *expander graph* – a sparse graph with good connectivity (in terms of Cheeger's constant) – with high probability [1, 15, 22, 30, 57], thereby enabling effective information propagation across the graph.

Some other rewiring techniques don't exactly classify as spatial or spectral methods. For instance, Probabilistically Rewired MPNN (PR-MPNN) [69] learns to probabilistically rewire a graph, effectively mitigating under-reaching as well as over-squashing. Finally, [25] proposed connecting all nodes at most $r$-hops away, for some $r \in \mathbb{N}$, and introducing positional embeddings to allow for distance-aware aggregation of messages.

## A.4 Towards a Unified Treatment

Several studies have shown that an inevitable trade-off exists between the problems of over-smoothing and over-squashing, meaning that optimizing for one will compromise the other. For instance, [64, 81] showed that negatively curved edges create bottlenecks in the graph resulting in over-squashing of information. On the other hand, [64, Proposition 4.3] showed that positively curved edges in a graph contribute towards the over-smoothing problem. To address this trade-off, they proposed Batch Ollivier-Ricci Flow (BORF), which adds new edges adjacent to the negatively curved ones, and simultaneously removes positively curved ones. In a similar vein, [34] demonstrated that the minimum number of message-passing steps required to reach a given level of over-smoothing is inversely related to the Cheeger's constant, $h_\mathcal{G}$. This again implies an inverse relationship between over-smoothing and over-squashing. To effectively alleviate the two issues together, they proposed the Stochastic Jost and Liu Curvature Rewiring (SJLR) algorithm, which adds edges that result in high improvement in the curvature of existing edges, while simultaneously removing those that have low curvature.

Despite the well-established trade-off between over-smoothing and over-squashing, some works have successfully tackled them together despite *only* adding or removing edges. One such work is [44], which proposed a rewiring algorithm that adds edges to the graph but does not remove any. The First-order Spectral Rewiring (FoSR) algorithm computes, as the name suggests, a first-order approximation to the spectral gap of the symmetric Laplacian matrix ($L^{\text{sym}} = I_N - (D^\dagger)^{1/2} A (D^\dagger)^{1/2}$), and adds edges with the aim of maximizing it. Since the spectral gap directly relates to Cheeger's constant – a measure of bottleneck-edness in the graph – through Cheeger's inequality [2, 3, 78], this directly decreases the over-squashing levels. Moreover, [44, Figure 5] empirically demonstrated that addition of (up to a small number of) edges selected by FoSR can lower the Dirichlet energy of the representations, suggesting the method's potential to simultaneously tackle over-smoothing. Taking a somewhat opposite approach, [55] adapted DropEdge to *remove* negatively curved edges sampled from a distribution proportional to edge curvatures. Their method, called CurvDrop, directly reduces over-squashing and, as a side benefit of operating on a sparser subgraph, also mitigates over-smoothing.

# B  Proofs

## B.1  Expected Propagation Matrix under DropEdge-variants

**Lemma.** *When using DropEdge, the expected propagation matrix is given as:*

$$\mathbb{E}_{\text{DE}}\left[\hat{A}_{ii}^{(1)}\right] = \frac{1 - q^{d_i+1}}{(1-q)\,(d_i+1)}$$

$$\mathbb{E}_{\text{DE}}\left[\hat{A}_{ij}^{(1)}\right] = \frac{1}{d_i}\left(1 - \frac{1 - q^{d_i+1}}{(1-q)\,(d_i+1)}\right)$$

*where $(j \to i) \in \mathcal{E}$; $\dot{P}_{ij} = 0$ otherwise.*

*Proof.* Recall that under DropEdge, a self-loop is added to the graph *after* the edges are dropped, and then the normalization is performed. In other words, the self-loop is never dropped. Therefore, given the i.i.d. masks, $m_1, \ldots, m_{d_i} \sim \text{Bern}\,(1-q)$, on incoming edges to node $i$, the total number of messages is given by

$$1 + \sum_{k=1}^{d_i} m_k = 1 + M_i \tag{B.1}$$

where $M_i \sim \text{Binom}\,(d_i, 1 - q)$. Under asymmetric normalization (see Section 2.1), the expected weight of the message along the self-loop is computed as follows:

$$\mathbb{E}_{\mathsf{DE}}\left[\hat{\boldsymbol{A}}_{ii}^{(1)}\right] = \mathbb{E}_{m_1,\ldots,m_{d_i}}\left[\frac{1}{1 + \sum_{k=1}^{d_i} m_k}\right] \tag{B.2}$$

$$= \mathbb{E}_{M_i}\left[\frac{1}{1 + M_i}\right] \tag{B.3}$$

$$= \sum_{k=0}^{d_i} \binom{d_i}{k} (1 - q)^k (q)^{d_i - k}\left(\frac{1}{1 + k}\right) \tag{B.4}$$

$$= \frac{1}{(1 - q)(d_i + 1)} \sum_{k=0}^{d_i} \binom{d_i + 1}{k + 1} (1 - q)^{k+1} (q)^{d_i - k} \tag{B.5}$$

$$= \frac{1}{(1 - q)(d_i + 1)} \sum_{k=1}^{d_i + 1} \binom{d_i + 1}{k} (1 - q)^k (q)^{d_i + 1 - k} \tag{B.6}$$

$$= \frac{1 - q^{d_i + 1}}{(1 - q)(d_i + 1)} \tag{B.7}$$

Similarly, if the Bernoulli mask corresponding to $j \to i$ is 1, then the total number of incoming messages to node $i$ is given by

$$2 + \sum_{k=1}^{d_i - 1} m_k$$

including one self-loop, which is never dropped, as noted earlier. On the other hand, the weight of the edge is simply 0 if the corresponding Bernoulli mask is 0. Using the Law of Total Expectation, the expected weight of the edge $j \to i$ can be computed as follows:

$$\mathbb{E}_{\mathsf{DE}}\left[\hat{\boldsymbol{A}}_{ij}^{(1)}\right] = q \cdot 0 + (1 - q)\,\mathbb{E}_{m_1,\ldots,m_{d_i - 1}}\left[\frac{1}{2 + \sum_{k=1}^{d_i - 1} m_k}\right] \tag{B.8}$$

$$= (1 - q) \sum_{k=0}^{d_i - 1} \binom{d_i - 1}{k} (1 - q)^k (q)^{d_i - 1 - k}\left(\frac{1}{2 + k}\right) \tag{B.9}$$

$$= \sum_{k=0}^{d_i - 1} \frac{(d_i - 1)!}{(k + 2)!\,(d_i - 1 - k)!} (1 - q)^{k+1} (q)^{d_i - 1 - k} (k + 1) \tag{B.10}$$

$$= \sum_{k=2}^{d_i + 1} \frac{(d_i - 1)!}{(k)!\,(d_i + 1 - k)!} (1 - q)^{k-1} (q)^{d_i + 1 - k} (k - 1) \tag{B.11}$$

$$= \frac{1}{d_i (d_i + 1)(1 - q)} \sum_{k=2}^{d_i + 1} \binom{d_i + 1}{k} (1 - q)^k (q)^{d_i + 1 - k} (k - 1) \tag{B.12}$$

$$= \frac{1}{d_i (d_i + 1)(1 - q)}\left[(d_i + 1)(1 - q) - 1 + q^{d_i + 1}\right] \tag{B.13}$$

$$= \frac{1}{d_i}\left(1 - \mathbb{E}_{\mathsf{DE}}\left[\hat{\boldsymbol{A}}_{ii}^{(1)}\right]\right) \tag{B.14}$$

$$\square$$

**Analysis of DropEdge-variants.** We will similarly derive the expected propagation matrix for other random edge-dropping algorithms. First off, DropNode [23] samples nodes and drops corresponding columns from the aggregation matrix directly, followed by rescaling of its entries:

$$\mathbb{E}_{\mathsf{DN}}\left[\frac{1}{1 - q}\hat{\boldsymbol{A}}\right] = \frac{1}{1 - q} \times (1 - q)\,\hat{\boldsymbol{A}} = \hat{\boldsymbol{A}} \tag{B.15}$$

That is, the expected propagation matrix is the same as in a NoDrop model ($q = 0$).

Nodes sampled by DropAgg [43] don't aggregate messages. Therefore, if $\hat{A} = \hat{A}^{\text{asym}}$, then the expected propagation matrix is given by

$$
\begin{aligned}
\mathbb{E}_{\text{DA}}\left[\hat{A}_{ii}\right] &= q + \frac{1-q}{d_i + 1} = \frac{1 + d_i q}{d_i + 1} > \mathbb{E}_{\text{DE}}\left[\hat{A}_{ii}\right] \\
\mathbb{E}_{\text{DA}}\left[\hat{A}_{ij}\right] &= \frac{1}{d_i}\left(1 - \mathbb{E}_{\text{DA}}\left[\hat{A}_{ii}\right]\right) < \mathbb{E}_{\text{DE}}\left[\hat{A}_{ij}\right]
\end{aligned}
\tag{B.16}
$$

Finally, DropGNN [66] samples nodes which neither propagate nor aggregate messages. From any node's perspective, if it is not sampled, then its aggregation weights are computed as for DropEdge:

$$
\begin{aligned}
\mathbb{E}_{\text{DG}}\left[\hat{A}_{ii}\right] &= q + (1-q)\,\mathbb{E}_{\text{DE}}\left[\hat{A}_{ii}\right] = q + \frac{1 - q^{d_i+1}}{d_i + 1} > \mathbb{E}_{\text{DA}}\left[\hat{A}_{ii}\right] \\
\mathbb{E}_{\text{DG}}\left[\hat{A}_{ij}\right] &= \frac{1}{d_i}\left(1 - \mathbb{E}_{\text{DG}}\left[\hat{A}_{ii}\right]\right) < \mathbb{E}_{\text{DA}}\left[\hat{A}_{ij}\right]
\end{aligned}
\tag{B.17}
$$

## B.2 Sensitivity in L-layer Linear GCNs

**Theorem.** *In an L-layer linear GCN with $\hat{A} = \hat{A}^{asym}$, using DropEdge, DropAgg or DropGNN decreases the sensitivity of a node $i \in \mathcal{V}$ to another node $j \in \mathbb{S}^{(L)}(i)$, thereby reducing its effective receptive field.*

$$
\mathbb{E}_{\cdots}\left[\left(\hat{A}^L\right)_{ij}\right] = \sum_{(u_0,\ldots,u_L)\in\text{Paths}(j\to i)} \prod_{\ell=1}^{L} \mathbb{E}_{\cdots}\left[\hat{A}_{u_\ell u_{\ell-1}}\right] < \mathbb{E}_{\text{ND}}\left[\left(\hat{A}^L\right)_{ij}\right]
\tag{B.18}
$$

*where* ND *refers to a NoDrop model ($q = 0$), the placeholder $\cdots$ can be replaced with one of the edge-dropping methods* DE, DA *or* DG, *and the corresponding entries of $\mathbb{E}_{\cdots}[\hat{A}]$ can be plugged in from Equation 3.3, Equation B.16 and Equation B.17, respectively. Moreover, the sensitivity monotonically decreases as the dropping probability is increased.*

*Proof.* Recall that $\dot{P}$ can be viewed as the transition matrix of a non-uniform random walk, such that $\dot{P}_{uv} = \mathbb{P}(u \to v)$. Intuitively, since there is no self-loop on any given L-length path connecting nodes $i$ and $j$ (which are assumed to be L-hops away), the probability of each transition on any path connecting these nodes is reduced. Therefore, so is the total probability of transitioning from $i$ to $j$ in exactly L hops.

More formally, denote the set of paths connecting the two nodes by

$$
\text{Paths}(j \to i) = \{(u_0,\ldots,u_L) : u_0 = j; u_L = i; (u_{\ell-1} \to u_\ell) \in \mathcal{E}, \forall \ell \in [L]\}
\tag{B.19}
$$

The $(i, j)$-entry in the propagation matrix is given by

$$
\left(\dot{P}^L\right)_{ij} = \sum_{(u_0,\ldots,u_L)\in\text{Paths}(j\to i)} \prod_{\ell=1}^{L} \dot{P}_{u_\ell u_{\ell-1}}
\tag{B.20}
$$

Since there is no self-loop on any of these paths,

$$
\left(\dot{P}^L\right)_{ij} = \sum_{(u_0,\ldots,u_L)\in\text{Paths}(j\to i)} \prod_{\ell=1}^{L} \frac{1}{d_{u_\ell}}\left(1 - \frac{1 - q^{d_{u_\ell}+1}}{(1-q)(d_{u_\ell}+1)}\right)
\tag{B.21}
$$

$$
< \sum_{(u_0,\ldots,u_L)\in\text{Paths}(j\to i)} \prod_{\ell=1}^{L} \left(\frac{1}{d_{u_\ell}+1}\right)
\tag{B.22}
$$

The right hand side of the inequality is the $(i, j)$-entry in the L[th] power of the propagation matrix of a NoDrop model. From Equation B.16 and Equation B.17, we know that Equation B.22 is true for

DropAgg and DropGNN as well. We conclude the first part of the proof using Equation 3.2 – the sensitivity of node $i$ to node $j$ is proportional to $(\dot{\boldsymbol{P}}^L)_{ij}$.

Next, we recall the geometric series for any $q$:

$$1 + q + \ldots + q^d = \frac{1 - q^{d+1}}{1 - q} \tag{B.23}$$

Each of the terms on the right are increasing in $q$, hence, all the $\dot{\boldsymbol{P}}_{u_\ell u_{\ell-1}}$ factors are decreasing in $q$. Similarly, $\mathbb{E}_{\mathsf{DA}}[\hat{\boldsymbol{A}}_{ij}]$ and $\mathbb{E}_{\mathsf{DG}}[\hat{\boldsymbol{A}}_{ij}]$ decrease with increasing $q$. Using these results with Equation B.20, we conclude the second part of the theorem.

$\square$

## C   Theoretical Extensions

### C.1   Sensitivity in Nonlinear MPNNs

While linear networks are useful in simplifying the theoretical analysis, they are often not practical. In this subsection, we will consider the upper bounds on sensitivity established in previous works, and extend them to the DropEdge setting.

**ReLU GCNs.** [86] considered the case of ReLU nonlinearity, so that the update rule is $\boldsymbol{Z}^{(\ell)} = \mathrm{ReLU}(\hat{\boldsymbol{A}}\boldsymbol{Z}^{(\ell-1)}\boldsymbol{W}^{(\ell)})$. Additionally, it makes the simplifying assumption that each path in the computational graph is *active* with a fixed probability, $\rho$ [45, Assumption A1p-m]. Accordingly, the sensitivity (in expectation) between any two nodes is given as

$$\left\| \mathbb{E}_{\mathsf{ReLU}}\left[\frac{\partial \boldsymbol{z}_i^{(L)}}{\partial \boldsymbol{x}_j}\right] \right\|_1 = \left[\rho \left\|\prod_{\ell=1}^{L}\boldsymbol{W}^{(\ell)}\right\|_1\right]\left(\hat{\boldsymbol{A}}^L\right)_{ij} = \zeta_1^{(L)}\left(\hat{\boldsymbol{A}}^L\right)_{ij} \tag{C.1}$$

where $\zeta_1^{(L)}$ depends only on the depth $L$, and is independent of the choice of nodes $i, j \in \mathcal{V}$. Taking an expectation w.r.t. the random edge masks, we get

$$\mathbb{E}_{\boldsymbol{M}^{(1)},\ldots,\boldsymbol{M}^{(L)}}\left[\left\|\mathbb{E}_{\mathsf{ReLU}}\left[\frac{\partial \boldsymbol{z}_i^{(L)}}{\partial \boldsymbol{x}_j}\right]\right\|_1\right] = \zeta_1^{(L)}\left(\mathbb{E}\left[\prod_{\ell=1}^{L}\hat{\boldsymbol{A}}^{(\ell)}\right]\right)_{ij} = \zeta_1^{(L)}\left(\mathbb{E}\left[\hat{\boldsymbol{A}}\right]^L\right)_{ij} \tag{C.2}$$

Using Theorem 3.1, we conclude that in a ReLU-GCN, DropEdge, DropAgg and DropGNN will reduce the expected sensitivity between nodes L-hops away. Empirical observations in Figures 1a and 4 suggest that we may expect an increase in sensitivity to neighboring nodes, but a significant decrease in sensitivity to those farther away.

**Source-only Message Functions.** [8, Lemma 3.2] considers MPNNs with aggregation functions of the form

$$\mathsf{Agg}^{(\ell)}\left(\boldsymbol{z}_i^{(\ell-1)}, \left\{\boldsymbol{z}_j^{(\ell-1)} : j \in \mathbb{S}^{(1)}(i)\right\}\right) = \sum_{j \in \mathbb{B}^{(1)}(i)} \hat{\boldsymbol{A}}_{ij}\mathsf{Msg}^{(\ell)}\left(\boldsymbol{z}_j^{(\ell-1)}\right) \tag{C.3}$$

and Upd and Msg functions with bounded gradients. In this case, the sensitivity between two nodes $i, j \in \mathcal{V}$ can be bounded as

$$\left\|\frac{\partial \boldsymbol{z}_i^{(L)}}{\partial \boldsymbol{x}_j}\right\|_1 \leq \zeta_2^{(L)}\left(\sum_{\ell=0}^{L}\hat{\boldsymbol{A}}^\ell\right)_{ij} \tag{C.4}$$

As before, we can use the independence of edge masks to get an upper bound on the expected sensitivity:

$$\mathbb{E}_{\boldsymbol{M}^{(1)},\ldots,\boldsymbol{M}^{(L)}}\left[\left\|\frac{\partial \boldsymbol{z}_i^{(L)}}{\partial \boldsymbol{x}_j}\right\|_1\right] \leq \zeta_2^{(L)}\left(\mathbb{E}\left[\boldsymbol{I}_N + \sum_{\ell=1}^{L}\prod_{k=1}^{\ell}\hat{\boldsymbol{A}}^{(k)}\right]\right)_{ij} = \zeta_2^{(L)}\left(\sum_{\ell=0}^{L}\mathbb{E}\left[\hat{\boldsymbol{A}}\right]^\ell\right)_{ij} \tag{C.5}$$

Figure 5 shows the plot of the entries of $\sum_{\ell=0}^{6} \dot{\boldsymbol{P}}^\ell$ (i.e. for DropEdge), as in the upper bound above, with $\hat{\boldsymbol{A}} = \hat{\boldsymbol{A}}^{\text{asym}}$. We observe that the sensitivity between nearby nodes marginally increases, while that between distant nodes notably decreases (similar to Figure 1a), suggesting significant over-squashing. Similar observations can be made with $\hat{\boldsymbol{A}} = \hat{\boldsymbol{A}}^{\text{sym}}$, and for other DropEdge-variants.

**Source-and-Target Message Functions.** [81, Lemma 1] showed that if the aggregation function is instead given by

$$\text{Agg}^{(\ell)}\left(\boldsymbol{z}_i^{(\ell-1)}, \left\{\boldsymbol{z}_j^{(\ell-1)} : j \in \mathbb{S}^{(1)}(i)\right\}\right) = \sum_{j \in \mathbb{B}^{(1)}(i)} \hat{\boldsymbol{A}}_{ij} \text{Msg}^{(\ell)}\left(\boldsymbol{z}_i^{(\ell-1)}, \boldsymbol{z}_j^{(\ell-1)}\right) \tag{C.6}$$

then the sensitivity between nodes $i \in \mathcal{V}$ and $j \in \mathbb{S}^{(L)}(i)$ can be bounded as

$$\left\|\frac{\partial \boldsymbol{z}_i^{(L)}}{\partial \boldsymbol{x}_j}\right\|_1 \leq \zeta_3^{(L)}\left(\hat{\boldsymbol{A}}^L\right)_{ij} \tag{C.7}$$

With random edge-dropping, this bound can be adapted as follows:

$$\mathbb{E}_{\mathbf{M}^{(1)},\ldots,\mathbf{M}^{(L)}}\left[\left\|\frac{\partial \boldsymbol{z}_i^{(L)}}{\partial \boldsymbol{x}_j}\right\|_1\right] \leq \zeta_3^{(L)}\left(\mathbb{E}\left[\hat{\boldsymbol{A}}\right]^L\right)_{ij} \tag{C.8}$$

which is similar to Equation C.2, only with a different proportionality constant, that is anyway independent of the choice of nodes. Here, again, we invoke Theorem 3.1 to conclude that $(\mathbb{E}[\hat{\boldsymbol{A}}]^L)_{ij}$ decreases monotonically with increasing DropEdge probability $q$. This implies that, in a non-linear MPNN with $\hat{\boldsymbol{A}} = \hat{\boldsymbol{A}}^{\text{asym}}$, DropEdge lowers the sensitivity bound given above. Empirical results in Figure 4 support the same conclusion for $\hat{\boldsymbol{A}} = \hat{\boldsymbol{A}}^{\text{sym}}$.

## C.2 Test-time Monte-Carlo Dropout

Up until now, we have focused on the expected sensitivity of the stochastic representations in models using DropEdge-variants. This corresponds to their training-time behavior, wherein the activations are random. At test-time, the standard practice is to turn these methods off by setting $q = 0$. However, this raises the over-smoothing levels back up [88]. Another way of making predictions is to perform multiple stochastic forward passes, as during training, and then averaging the model outputs. This is similar to Monte-Carlo Dropout, which is an efficient way of ensemble averaging in MLPs [27], CNNs [26] and RNNs [28]. In addition to alleviating over-smoothing, this approach also outperforms the standard implementation in practical settings [88]. We can study the effect of random edge-dropping in this setting by examining the sensitivity of the *expected representations*:

$$\left\|\frac{\partial}{\partial \boldsymbol{x}_j}\mathbb{E}\left[\boldsymbol{z}_i^{(L)}\right]\right\|_1 = \left\|\mathbb{E}\left[\frac{\partial \boldsymbol{z}_i^{(L)}}{\partial \boldsymbol{x}_j}\right]\right\|_1 \tag{C.9}$$

In linear models, the order of the two operations – expectation and 1-norm – is irrelevant:

$$\left\|\mathbb{E}\left[\frac{\partial \boldsymbol{z}_i^{(L)}}{\partial \boldsymbol{x}_j}\right]\right\|_1 = \left\|\mathbb{E}\left[\left(\hat{\boldsymbol{A}}^L\right)_{ij}\right]\boldsymbol{W}\right\|_1 = \mathbb{E}\left[\left(\hat{\boldsymbol{A}}^L\right)_{ij}\|\boldsymbol{W}\|_1\right] = \mathbb{E}\left[\left\|\frac{\partial \boldsymbol{z}_i^{(L)}}{\partial \boldsymbol{x}_j}\right\|_1\right] \tag{C.10}$$

In general, the two quantities can be related using the convexity of norms and Jensen's inequality:

$$\left\|\frac{\partial}{\partial \boldsymbol{x}_j}\mathbb{E}\left[\boldsymbol{z}_i^{(L)}\right]\right\|_1 \leq \mathbb{E}\left[\left\|\frac{\partial \boldsymbol{z}_i^{(L)}}{\partial \boldsymbol{x}_j}\right\|_1\right] \leq \ldots \tag{C.11}$$

Therefore, the upper bound results in Appendix C.1 trivially extend to the MC-averaged representations. Although tighter bounds may be derived for this setting, we leave that for future works.

# D Empirical Sensitivity Analysis

In this section, we elaborate on the setup used for the empirical sensitivity analysis in Section 3. We also present some supplemental figures demonstrating the negative effects of random edge-dropping, particularly focusing on scenarios not covered by the theory.

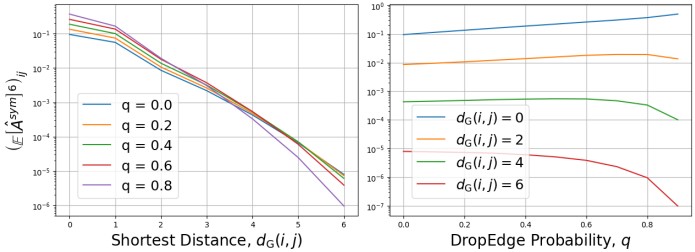

**Figure 4:** Entries of $\ddot{\boldsymbol{P}}^6$, averaged after binning node-pairs by their shortest distance.

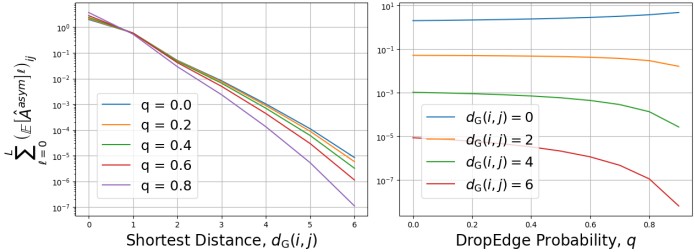

**Figure 5:** Entries of $\sum_{\ell=0}^{6} \dot{\boldsymbol{P}}^\ell$, averaged after binning node-pairs by their shortest distance.

### D.1 MC-Approximation of Sensitivity in Nonlinear MPNNs

Given a target node from the Cora dataset [58], we computed the sensitivity of its representation to source nodes up to $L = 6$ hops away in ReLU-GCNs of width 32. The raw sensitivities were normalized to obtain *influence scores* [86]. This was repeated for 25 target nodes, and 25 model$-$dropout samples were used for each of them. The source nodes were binned by the shortest distance from the corresponding target node, and the influence scores were averaged over each bin to obtain an *average influence* from nodes $\ell$-hops away.

### D.2 Symmetrically Normalized Propagation Matrix

The results in Section 3 correspond to the use of $\hat{\boldsymbol{A}} = \hat{\boldsymbol{A}}^{\text{asym}}$ for aggregating messages – in each message passing step, only the in-degree of node $i$ is used to compute the aggregation weights of the incoming messages. In practice, however, it is more common to use the symmetrically normalized propagation matrix, $\hat{\boldsymbol{A}} = \hat{\boldsymbol{A}}^{\text{sym}}$, which ensures that nodes with high out-degree do not dominate the information flow in the graph [47]. As in Equation 3.2, we are looking for

$$\ddot{\boldsymbol{P}}^L := \mathbb{E}_{\mathbf{M}^{(1)},\ldots,\mathbf{M}^{(L)}}\left[\prod_{\ell=1}^{L}\hat{\boldsymbol{A}}^{(\ell)}\right] \tag{D.1}$$

where $\ddot{\boldsymbol{P}} := \mathbb{E}_{\text{DE}}[\hat{\boldsymbol{A}}^{\text{sym}}]$. While $\ddot{\boldsymbol{P}}$ is analytically intractable, we can approximate it using Monte-Carlo sampling. Accordingly, we use the Cora dataset, and sample 20 DropEdge masks to compute an approximation of $\ddot{\boldsymbol{P}}$, and plot out the entries of $\ddot{\boldsymbol{P}}^L$, as we did for $\dot{\boldsymbol{P}}^L$ in Figure 1a. The results are presented in Figure 4, which shows that while the sensitivity between nodes up to 3 hops away is increased, that between nodes farther off is significantly reduced, same as in Figure 1a.

### D.3 Upper Bound on Expected Sensitivity

[8] showed that the sensitivity between any two nodes in a graph can be bounded using the sum of the powers of the propagation matrix. In Appendix C.1, we extended this bound to random edge-dropping methods with independent edge masks sampled in each layer:

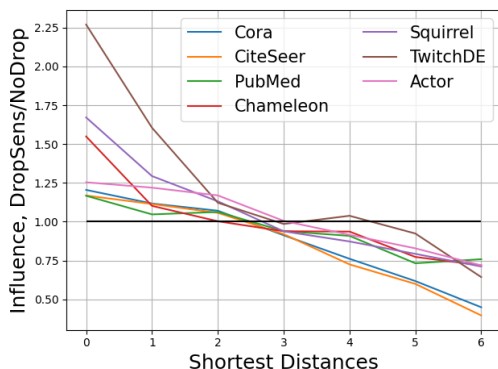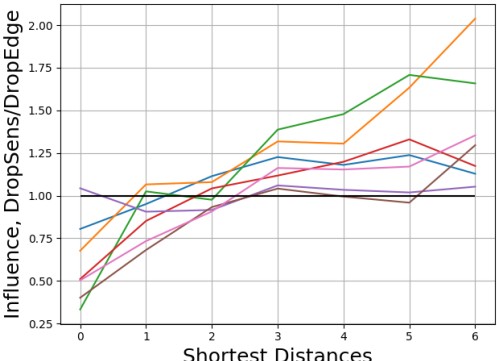

**Figure 6:** Average sensitivity under DropSens, normalized by average sensitivity under NoDrop (*left*) and DropEdge (*right*). We note that DropSens is more sensitive to nearby nodes ($d \leq 2$) than NoDrop, and more sensitive to far off nodes ($d \geq 3$) than DropEdge.

$$\mathbb{E}_{\mathbf{M}^{(1)},...,\mathbf{M}^{(L)}} \left[ \left\| \frac{\partial \boldsymbol{z}_i^{(L)}}{\partial \boldsymbol{x}_j} \right\|_1 \right] \leq \zeta_3^{(L)} \left( \sum_{\ell=0}^{L} \mathbb{E} \left[ \hat{\boldsymbol{A}} \right]^{\ell} \right)_{ij}$$

Although this bound does not have a closed form, we can again use the Cora network to study its entries. We plot the entries of $\sum_{\ell=0}^{6} \dot{\boldsymbol{P}}^{\ell}$, corresponding to DropEdge, against the shortest distance between node-pairs. The results are presented in Figure 5. We observe an exponential decrease in the sensitivity bound as the distance between nodes increases, suggesting that DropEdge is not suitable for capturing LRIs.

### D.4 Closer Look at DropSens

Lastly, we evaluate DropSens on node-classification tasks, comparing its effects with NoDrop and DropEdge. In Figure 6, we compare the sensitivity profiles of these methods. It is clear to observe that DropSens allows node representations to be more sensitive to distant nodes' ($d \geq 3$) features, than DropEdge, thereby improving performance on heterophilic datasets. What is interesting is that for homophilic datasets, DropSens is more sensitive to nearby nodes ($d \geq 1$) than DropEdge, allowing the model to effectively capture short-range interactions. Although it is poorer at capturing short-range interactions for heterophilic datasets, DropSens improves over NoDrop on that front, similar to DropEdge.

This set of experiments suggests that DropSens strikes a perfect balance between NoDrop and DropEdge:

1. improves sensitivity to nearby nodes ($d \leq 2$), over NoDrop,
2. improves sensitivity to distant nodes ($d \geq 3$), over DropEdge, and
3. mitigates over-fitting (by stochastic regularization) and over-smoothing (by reducing message-passing), just as DropEdge.

## E    Experiments Details

In this section, we expand on the details of the experiments in Section 5. All experiments were run on a server equipped with an Intel(R) Xeon(R) E5-2620 v3 CPU, 62 GB of RAM, 4 × NVIDIA GeForce GTX TITAN X GPU (12 GB VRAM each), and CUDA version 12.4.

### E.1    Descriptions of the Datasets

**Synthetic Datasets.** The SyntheticZINC dataset [33], as the name suggests, is a synthetic dataset derived from the ZINC chemical dataset [42], with the dataset size constrained to 12K molecular

**Table 4:** Statistics of node-classification datasets. Homophily measures from [53].

| Dataset | Nodes | Edges | Features | Classes | Homophily |
|---------|-------|-------|----------|---------|-----------|
| **Homophilic Networks** | | | | | |
| Cora | 2,708 | 10,556 | 1,433 | 7 | 0.766 |
| CiteSeer | 3,327 | 9,104 | 3,703 | 6 | 0.627 |
| PubMed | 19,717 | 88,648 | 500 | 3 | 0.664 |
| **Heterophilic Networks** | | | | | |
| Chameleon | 2,277 | 36,051 | 2,325 | 5 | 0.062 |
| Squirrel | 5,201 | 216,933 | 2,089 | 5 | 0.025 |
| Actor | 7,600 | 29,926 | 931 | 5 | 0.011 |
| TwitchDE | 9,498 | 306,276 | 128 | 2 | 0.142 |

**Table 5:** Statistics of graph-classification datasets.

| Dataset | Graphs | Nodes | Edge | Features | Classes |
|---------|--------|-------|------|----------|---------|
| Mutag | 188 | 17.9 | 39.6 | 7 | 2 |
| Proteins | 1,113 | 39.1 | 145.6 | 3 | 2 |
| Enzymes | 600 | 32.6 | 124.3 | 3 | 6 |
| Collab | 5,000 | 74.5 | 4914.4 | 0 | 3 |
| IMDb | 1,000 | 19.8 | 193.1 | 0 | 2 |
| Reddit | 2,000 | 429.6 | 995.5 | 0 | 2 |

graphs [20]. Specifically, given a molecular graph $\mathcal{G}$, we set all its nodes' features to 0, except for two nodes, $i$ and $j \neq i$, whose features are sampled as $x_i, x_j \in \mathcal{U}(0, 1)$. The graph-level target is computed as $y = \tanh(x_i + x_j)$, i.e. learning the task requires a non-linear mixing between the features of nodes $i$ and $j$. These nodes are chosen to induce the desired level of underlying mixing – given $\alpha \in [0, 1]$, the node-pair $(i, j)$ is chosen such that the commute time [10] between them is the $\alpha^{\text{th}}$ quantile of the distribution of commute times over $\mathcal{G}$. We analyze the effect of underlying mixing on model performance by varying $\alpha$ as $0.1, 0.2, \ldots, 1.0$. The MPNN is chosen to be an L-layer GCN with a MAX-pooling readout, which encourages the model to learn the mixing by effectively passing messages [33, Theorem 3.2]. The model depth is set at $L = \max_{\mathcal{G}} \lceil \text{diam}(\mathcal{G})/2 \rceil = 11$ to ensure that the GCN does not suffer from under-reaching [4, 7].

**Node-classification Tasks.** Cora [58], CiteSeer [31] and PubMed [63] are citation networks – their nodes represent scientific publications and an edge between two nodes indicates that one of them has cited the other. The features of each publication are represented by a binary vector, where each index indicates whether a specific word from a dictionary is present or absent. Several studies have showed that these datasets have high homophily in node labels [53, 100] and that they are modelled much better by shallower networks than by deeper ones [97]. Chameleon and Squirrel [71] are networks of English Wikipedia web pages on the respective topics, and the edges between web pages indicate links between them. The task is to predict the average-monthly traffic on each of the web pages. The Actor dataset is induced from a larger film-director-actor-writer network [68]. It is a network of film actors, with edges between those that occur on the same Wikipedia page, and node features are binary vectors denoting the presence of specific keywords in the corresponding Wikipedia entries. The task is to classify actors into five categories based on the content of their Wikipedia pages. Finally, TwitchDE [71] is a network of Twitch users in Germany, with the edges between them representing their mutual follower relationships. The node features are embeddings of the games played by the users, and the task is to predict whether the users use explicit language.

**Graph-classification Tasks.** Following [8, 44], we use the graph-classification datasets introduced in [62], which were hypothesized to be long-range tasks. On one hand we have the molecular datasets Mutag, Proteins and Enzymes, and on another we have the social networks Reddit-Binary, IMDb-Binary and Collab. Their statistics are presented in Table 5.

Mutag [17] consists of nitroaromatic compounds, with the objective of predicting their mutagenicity, i.e. the ability to cause genetic mutations, in cells in *Salmonella typhimurium*. Each compound is represented as a graph, where nodes correspond to atoms, represented by their type using one-hot encoding, and edges denote the bonds between them. Proteins [9, 19] is a collection of proteins classified as enzymes or not. The molecules are represented as a graph of amino acids, with edges between those separated up to 6Å apart. The Enzymes [9, 75] dataset is represented similarly, and the task is to classify the proteins into one of 6 Enzyme Commission (EC) numbers – a system to classify enzymes based on the reactions they catalyze.

Collab [89] is a scientific collaboration dataset where each graph represents a researcher's ego network. Nodes correspond to researchers and their collaborators, with edges indicating co-authorship. Each network is labeled based on the researcher's field, which can be High Energy Physics, Condensed Matter Physics, or Astrophysics. IMDb-Binary [89] is a movie collaboration dataset containing the ego-networks of 1,000 actors and actresses from IMDB. In each graph, nodes represent actors, with edges connecting those who have co-starred in the same film. Each graph is derived from either the Action genre or Romance. Finally, Reddit-Binary [89] comprises graphs representing online discussions on Reddit, where nodes correspond to users and edges indicate interactions through comment responses. Each graph is labeled based on whether it originates from a Q&A or discussion-based subreddit.

For the molecular datasets, we use the node features supplied by PyG [24], but since they are unavailable for the social networks, we set scalar features $x_i = (1)$ for all nodes in these datasets, following [44].

## E.2 Training Configurations

**Dataset Splits.** For the SyntheticZINC task, we use the train-val-test splits provided in PyG. For the homophilic (citation) networks, we use the 'full' split [12], as provided in PyG, and for the heterophilic networks, we randomly sample 60% of the nodes for training, 16% for validation, and 24% for testing. On the other hand, for the graph classification tasks, we sample 80% of the graphs for training, and 10% each for validation and testing, following [8, 44].

**Model Architecture.** We standardize most of the hyperparameters across all experiments to isolate the effect of random dropping. Specifically, we use symmetric normalization of the adjacency matrix to compute the edge weights for GCN, and we set the number of attentions heads for GAT to 2 in order to keep the computational load manageable, while at the same time harnessing the expressiveness of the multi-headed self-attention mechanism. For the SyntheticZINC dataset, we fix the size of the hidden representations at 16, while we fix them to 64 for all the real-world datasets. In all settings, a linear transformation is applied to the node features before message-passing. Afterwards, a bias term is added and then the ReLU nonlinearity is applied. Finally, a linear readout layer is used to compute the regressand (for regression tasks) or logits (for classification).

**Dropping Probability.** For the synthetic datasets, we experiment with a NoDrop baseline, and DropEdge, Dropout and DropMessage, each with $q = 0.2$ and $q = 0.5$. For the real-world datasets, the dropping probabilities are varied as $q = 0.1, 0.2, \ldots, 0.9$, so as to reliably find the best performing configuration. We adopt the common practice of turning the dropping methods off at test-time ($q = 0$), isolating the effects on optimization and generalization, which our theory does not address.

**DropSens Configurations.** For DropSens, we use 4 possible values for proportion of information preserved over corss-edges, $c = 0.5, 0.8, 0.9, 0.95$. Since the dropping probability, $q_i$, increases with the in-degree of the target node, $d_i$, the proportion of all edges dropped could become very high, especially with small $c$. Therefore, we clip the value of $q_i$ by 4 possible choices, $q_i \leq q_{\max} \in \{0.2, 0.3, 0.5, 0.8\}$. We exclude the following configurations: $(c, q_{\max}) \in \{(0.5, 0.2), (0.5, 0.3), (0.5, 0.5), (0.8, 0.2)\}$, since they use the same dropping probability ($= q_{\max}$) for each edge, and are therefore equivalent to DropEdge. In summary, we test with a total of 12 configurations for DropSens to find the best one for each task.

**Optimization Algorithm.** The models are trained using the Adam optimizer [46]. On the SyntheticZINC dataset, the models are trained with a learning rate of $2 \times 10^{-3}$ and a weight decay of $1 \times 10^{-4}$, for a total of 200 epochs. On the real-world datasets, we use a learning rate of $1 \times 10^{-3}$ and no weight decay, following [8, 44]. Here, we cap the maximum number of epochs at 300. In

both cases, the learning rate is reduced by a factor of $1 \times 10^{-1}$ if the validation loss fails to improve beyond a relative threshold of $1 \times 10^{-4}$ for 10 epochs, again following [8, 44].

**Number of Independent Runs.** We perform only 10 independent runs on the SyntheticZINC dataset due to its consistently low variance in performance, as also observed by [33]. For real-world datasets, we conduct 20 runs to identify the best-performing dropping configurations. We then perform a one-sided t-test to assess whether dropout improves performance, using a 90% confidence-level ($\alpha = 0.1$) and targeting a statistical power of 0.9 ($\beta = 0.1$). Under the assumption that the dropping method offers superior performance, detecting a medium effect size of 0.5 [14] requires approximately 53 samples per group according to standard power analysis. We round this to 50, and accordingly perform 30 additional runs for the final comparison of the best-performing dropping configuration with the NoDrop baseline.

### E.3 DropSens Implementation

```python
import warnings
import sympy
from sympy.abc import x as q
import torch
from torch_geometric.utils import degree, contains_self_loops

def drop_sens(
    edge_index: torch.Tensor,
    c: float,
    max_drop_prob: float = None
):

    if max_drop_prob is None:
        max_drop_prob = 1.

    # Assuming edge index does not have self loops
    if contains_self_loops(edge_index):
        warnings.warn("Degree computation in DropSens assumes absence"
            " of self-loops, but the edge_index passed contains some.")
    degrees = degree(edge_index[1]).int()        # Node index -> in-degree

    ds = torch.unique(degrees).tolist()          # Sorted array
    mapper = torch.nan * torch.ones(ds[-1]+1)
    mapper[ds] = max_drop_prob                   # Node in-degree -> dropping prob

    for d_i in ds:
        q_i = float(sympy.N(sympy.real_roots(
            (1-c)*d_i*(1-q) - q + q**(d_i+1))[-2]    # Following Equation 4.1
        )) if d_i > 0 else 0.
        if q_i > max_drop_prob:
            # Because q monotonic wrt d, and ds is sorted
            break
        mapper[d_i] = q_i

    in_degrees = degrees[edge_index[1]]          # Edge index -> in-degree of target node
    qs = mapper[in_degrees]                      # Edge index -> dropping probability
    edge_mask = qs <= torch.rand(edge_index.size(1))
    edge_index = edge_index[:, edge_mask]

    return edge_index, edge_mask
```

**Listing 1:** DropSens Implementation

In Listing 1, we present the DropSens implementation used in our experiments, relying mainly on SymPy [60].

Unfortunately, computing the roots of Equation 4.1 becomes slow when the in-degree $d_i$ is large – a common scenario in large networks. This issue is especially pronounced when the proportion of information preserved $c$ is large, as the dropping threshold is only met at a higher value of $d_i$. To address this computational challenge, we propose an approximation:

$$1 - c = \frac{q_i - q_i^{d_i+1}}{d_i \left(1 - q_i\right)} \approx \frac{q_i}{d_i \left(1 - q_i\right)} \implies q_i \approx \frac{(1-c)\, d_i}{1 + (1-c)\, d_i} \tag{E.1}$$

This approximation becomes increasingly accurate as $c$ increases – since more information needs to be preserved, $q_i$ needs to be small, and hence, $q_i^{d_i+1} \to 0$.

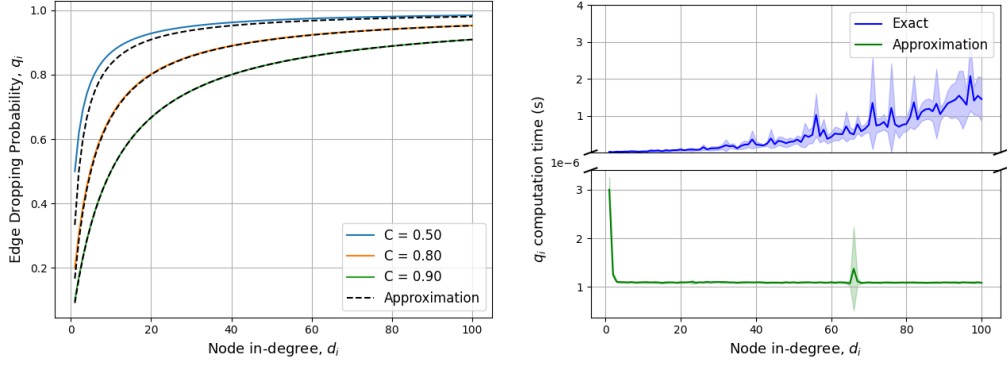

**(a)** Edge-wise dropping probabilities      **(b)** Compute time, averaged over $c = 0.1, \ldots, 0.9$

**Figure 7:** Edge-wise dropping probabilities under DropSens for varying values of $c$, along with the approximation as in Equation E.1. Compute times are measured using only 1 execution thread.

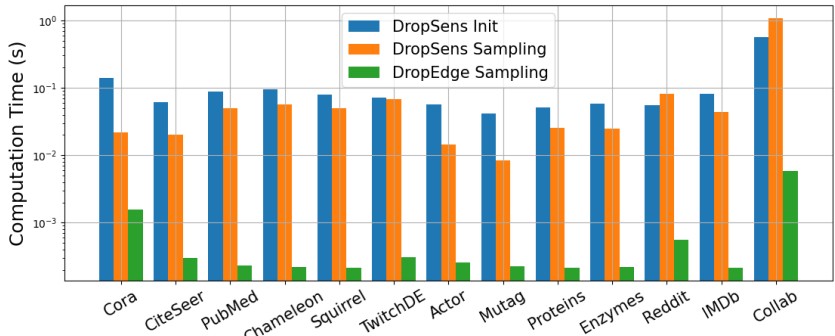

**Figure 8:** Initialization and sampling time of DropSens ($c = 0.8$, $q_{\max} = 0.5$) compared to sampling time of DropEdge, averaged over 10 runs. For graph classification tasks, edge masks are computed in one go (instead of one mini-batch at a time, as in practice).

Figure 7a shows the DropSens probabilities for masking incoming messages based on the in-degree of the target nodes, along with the corresponding approximations. It is clear to see that the approximation gets increasingly more accurate for higher proportions of information preserved. Figure 7b shows that the approximation is several orders of magnitude cheaper than the exact computation, and also does not scale with the in-degree $d_i$, as expected.[7]

Figure 8 shows the initialization time of DropSens, and compares the sampling times of DropSens and DropEdge. It is clear to see that DropSens sampling is more expensive than DropEdge, primarily because the edge-wise dropping probabilities need to be computed for new graphs,[8] before sampling edge masks. Moreover, in our experiments, sampling consumed a very small fraction of the total training time.

## F    Supplementary Experiments

### F.1    Test Accuracy versus DropEdge Probability

In Section 3, we studied the effect of edge-dropping probability on sensitivity between nodes at different distances. However, this analysis may be insufficient to precisely predict the impact on model performance since DropEdge-variants significantly affect the optimization trajectory as well.

---

[7]Note that this comparison is only valid for our `SymPy`-based implementation, and can be significantly different for other implementations.

[8]This step can be sped up for node-level tasks by caching edge-wise dropping probabilities.

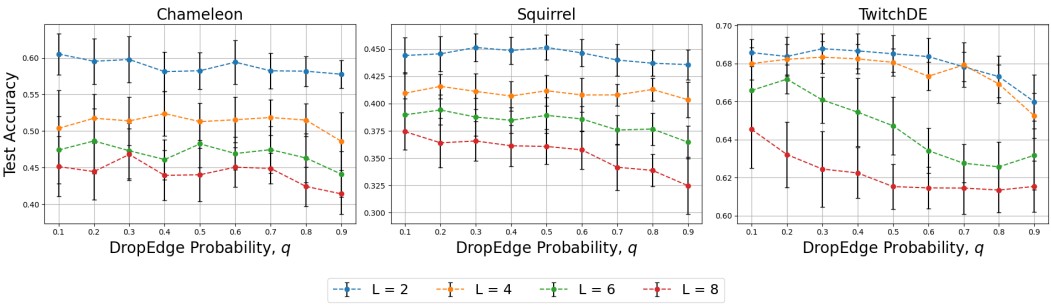

**Figure 9:** Dropping probability versus test accuracy of DropEdge-GCN. The theory the explains the contrasting trends as follows: random edge-dropping pushes models to fit to local information during training, which is suitable for short-range tasks, but harms test-time performance in long-range ones.

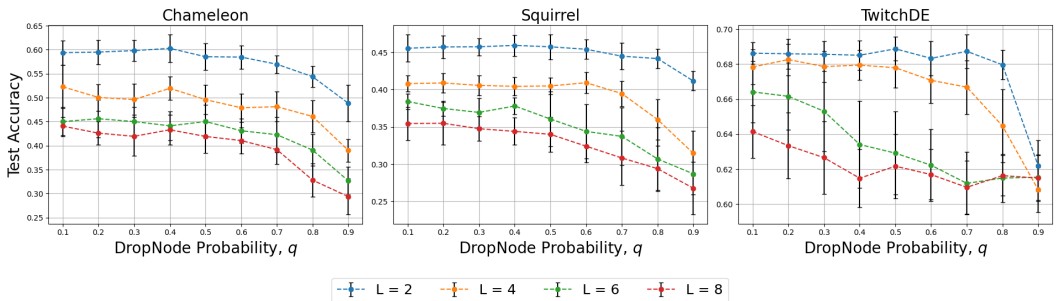

**Figure 10:** Dropping probability versus test accuracy of DropNode-GCN.

To learn more about the relationship between test-time performance and dropping probability, we evaluate DropEdge-GCNs on the heterophilic datasets; the results are shown in Figure 9. Clearly, on Chameleon, Squirrel and TwitchDE, the performance degrades with increasing dropping probability, as was suggested by Theorem 3.1 and Figure 1a. Surprisingly, the trends are significantly monotonic with GCNs of all depths, $L = 2, 4, 6, 8$.

### F.2    Remark on DropNode

In Equation B.15, we noted that DropNode does not suffer from loss in sensitivity. However, those results were in expectation. Moreover, our analysis did not account for the effects on the learning trajectory. In practice, a high DropNode probability would make it hard for information in the node features to reach distant nodes. This would prevent the model from learning to effectively combine information from large neighborhoods, harming generalization. In Figure 10, we visualize the relationship between test-time performance and DropNode probability. The performance monotonically decreases with increasing dropping probability, as was observed with DropEdge.

### F.3    Over-squashing or Under-fitting?

The results in the previous subsection suggest that using random edge-dropping to regularize model training leads to poor test-time performance. We hypothesize that this occurs because the models struggle to propagate information over long distances, causing node representations to overfit to local neighborhoods. However, a confounding effect is at play: DropEdge variants reduce the generalization gap by preventing overfitting to the training set. If this regularization is too strong, the model could underfit, which could also explain the poor test-time performance on heterophilic datasets. This concern is particularly relevant because the heterophilic networks are much larger than homophilic ones (see Table 4), making them more prone to underfitting. To investigate this, we plot the training accuracies of DropEdge-GCNs on the heterophilic datasets; Figure 11 shows the results. It is clear that the models do not underfit as the dropping probability increases. In fact, somewhat unexpectedly, the training metrics improve. Together with the results in Figure 9, we con-

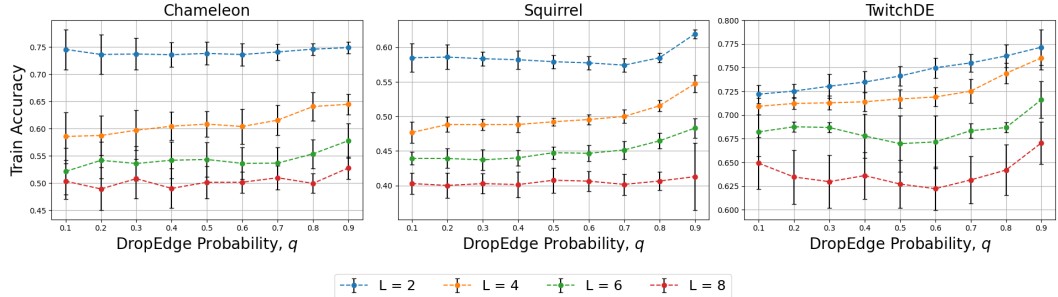

**Figure 11:** DropEdge probability versus training accuracy of GCNs. The training performance improves with $q$, suggesting that the models are not underfitting. Instead, the reason for poor test-time performance (Figure 9) is that models are over-fitting to short-range signals during training, resulting in poor generalization.

clude that DropEdge-like methods are detrimental in long-range tasks since they cause overfitting to short-range artifacts in the training data, resulting in poor generalization at test-time.

# G  Supplementary Experimental Results

## G.1  Ranking Dropping Methods

In Table 1, we reported the mean improvement in test accuracy (%) of each dropout method over the baseline NoDrop model, using the best-performing configuration for each. Table 6 presents the corresponding raw accuracies and ranks the methods accordingly. Notably, DropSens ranks first in $7/38 \approx 18\%$ of dataset×model combinations, and places within the top 3 in $21/38 \approx 55\%$ cases, highlighting its consistent efficacy across a broad range of settings.[9]

## G.2  Performance of GAT with Dropping Methods

In Table 7, we present the results of experiments in Section 5.2, but with the GAT architecture. For node classification tasks, we see the same dichotomy as in Table 1a, with dropping methods significantly improving performance on homophilic networks, while being detrimental to performance on heterophilic networks. On graph classification tasks, the dropping methods improve performance in $13/30 \approx 43\%$ cases, but the improvement, if any, is not statistically significant ($24/30 \approx 80\%$ cases). Note that the GAT architecture was unable to learn the Collab dataset, i.e. the performance in all cases was as good as a random classifier's.

## G.3  Effect Size in Statistical Tests

The reliance on p-values as a measure of statistical significance has been widely criticized due to its limitations in conveying the magnitude of an effect. Although a low p-value indicates that an observed difference is unlikely to have occurred under the null hypothesis, it does not provide information about the practical significance of the result. A statistically significant effect may be too small to be meaningful in real-world applications, while a non-significant result does not necessarily imply the absence of a meaningful effect, particularly when sample sizes are small. These concerns have led to an increased emphasis on effect size measures, which quantify the magnitude of differences independently of sample size.

One widely used measure of effect size is Cohen's $d$ statistic [14], which standardizes the difference between two group means by dividing by the pooled standard deviation:

$$d = \frac{\bar{x}_1 - \bar{x}_2}{s_p} \tag{G.1}$$

$$s_p = \sqrt{\frac{(n_1 - 1)\, s_1^2 + (n_2 - 1)\, s_2^2}{n_1 + n_2 - 2}} \tag{G.2}$$

---

[9]We exclude Collab×GAT due to unsuccessful runs.

**Table 6:** Accuracy of dropping methods and their rankings.

| GNN | Dataset | Nodrop | DropSens | DropEdge | DropNode | DropAgg | DropGNN | Dropout | DropMessage |
|---|---|---|---|---|---|---|---|---|---|
| | Cora | 85.06 | 85.17 | 85.27 | **85.27** | 85.15 | **85.43** | **85.48** | 85.20 |
| | CiteSeer | 73.39 | 73.81 | **73.90** | 73.09 | **73.94** | **74.11** | 73.13 | 73.48 |
| | PubMed | 83.75 | **83.76** | 83.65 | 83.67 | 83.32 | 83.56 | **84.86** | **84.57** |
| | Chameleon | **55.25** | 53.45 | **53.54** | 50.26 | 40.07 | 51.41 | 51.58 | **53.75** |
| | Squirrel | **39.74** | **39.72** | **39.40** | 38.43 | 26.00 | 38.29 | 38.44 | 39.19 |
| | TwitchDE | **68.01** | **68.26** | 67.97 | 66.83 | 62.71 | 67.06 | 67.50 | **68.28** |
| GCN | Actor | 22.54 | **22.89** | **22.80** | 22.42 | 22.08 | **23.15** | 22.71 | 22.25 |
| | Mutag | 71.70 | **75.30** | 70.50 | 71.70 | **72.80** | **74.30** | 69.00 | 71.50 |
| | Proteins | 71.29 | **73.27** | 71.77 | **72.16** | 71.00 | **71.77** | 71.20 | 71.68 |
| | Enzymes | **31.39** | **28.42** | 28.36 | 28.31 | 25.97 | **28.49** | 26.91 | 27.81 |
| | Reddit | **82.23** | **80.16** | 75.95 | 71.30 | 64.78 | 69.29 | **81.31** | 78.46 |
| | IMDb | 48.58 | **49.34** | **49.26** | 48.26 | 49.12 | 47.40 | 49.16 | **52.00** |
| | Collab | **65.27** | 64.16 | 61.92 | 60.76 | 44.33 | **66.15** | 61.97 | **67.88** |
| | Cora | 76.12 | 75.15 | 70.75 | **80.10** | 73.65 | 75.19 | **80.89** | **81.78** |
| | CiteSeer | 63.69 | 63.80 | 51.78 | **68.11** | 52.59 | 53.54 | **66.82** | **66.53** |
| | PubMed | 81.78 | 81.16 | 77.81 | **83.05** | 79.75 | 79.51 | **84.10** | **83.79** |
| | Chameleon | **44.28** | 34.23 | 36.47 | 41.63 | 37.05 | 34.74 | **43.42** | **48.53** |
| | Squirrel | **33.30** | 31.49 | 31.80 | 32.43 | 28.33 | 28.18 | **34.36** | **35.74** |
| | TwitchDE | 63.48 | **65.37** | 64.20 | 63.90 | 62.55 | 63.42 | **64.73** | **65.40** |
| GIN | Actor | 20.24 | 20.39 | **22.56** | 20.16 | **23.18** | **20.74** | 20.27 | 20.22 |
| | Mutag | 79.50 | **80.90** | 76.60 | 71.00 | 79.70 | 75.80 | **80.10** | **80.60** |
| | Proteins | 69.02 | 68.20 | 67.87 | **69.55** | 69.16 | 68.36 | **69.68** | **70.55** |
| | Enzymes | **44.80** | 30.40 | 35.85 | 37.24 | **44.16** | 33.06 | **42.76** | 42.40 |
| | Reddit | **89.72** | **90.32** | 89.29 | 88.22 | 88.16 | 85.57 | **90.11** | 89.59 |
| | IMDb | 69.96 | 67.50 | 69.72 | **70.34** | 69.16 | 67.22 | **70.04** | **71.30** |
| | Collab | 73.47 | 70.68 | 72.91 | **73.59** | 72.48 | 70.01 | **73.85** | **74.98** |
| | Cora | 83.89 | **84.49** | 84.47 | 84.37 | 84.19 | 84.43 | **84.95** | **84.88** |
| | CiteSeer | 72.87 | **73.38** | **73.53** | 72.44 | 73.23 | **73.88** | 72.73 | 72.44 |
| | PubMed | 83.36 | **83.45** | 83.28 | 83.00 | 83.02 | 83.05 | **83.44** | **83.54** |
| | Chameleon | **62.87** | **60.28** | 59.81 | 49.74 | 48.97 | 48.02 | 57.02 | **60.53** |
| | Squirrel | **44.94** | 41.70 | **41.77** | 36.63 | 36.27 | 35.07 | 38.55 | **42.40** |
| | TwitchDE | **67.20** | **67.03** | 66.83 | 64.40 | 66.13 | 65.45 | 66.22 | **67.08** |
| GAT | Actor | 21.74 | 22.57 | **22.69** | 21.67 | **22.75** | **22.75** | 21.91 | 21.97 |
| | Mutag | 74.30 | **75.90** | 71.10 | 72.60 | 73.20 | 68.70 | **73.50** | 72.10 |
| | Proteins | 71.32 | **72.96** | 71.64 | 70.89 | **72.02** | **71.86** | 70.61 | 71.09 |
| | Enzymes | 28.42 | **29.70** | **28.67** | 25.51 | 26.79 | 25.08 | 26.38 | **29.11** |
| | Reddit | 48.71 | 49.31 | 47.96 | **50.15** | 48.87 | 48.35 | **50.23** | **49.90** |
| | IMDb | 48.08 | **49.48** | 47.56 | 49.36 | 48.36 | 47.56 | **49.50** | **50.20** |
| | Collab | 33.33 | 33.33 | 33.33 | 33.33 | 33.33 | **33.33** | 33.33 | **33.33** |

where $\bar{x}_1$ and $\bar{x}_2$ are sample means, $s_1^2$ and $s_2^2$ are unbiased sample variance, and $n_1$ and $n_2$ are the sizes of the samples. However, Cohen's $d$ assumes that the sample standard deviation is an unbiased estimator of the population standard deviation. In small samples, this assumption does not hold, as the sample standard deviation tends to underestimate the true population variability. To address this bias, Hedges' $g$ statistic [39] introduces a correction factor that adjusts Cohen's $d$ statistic for small sample sizes:

$$g \approx \left( 1 - \frac{3}{4\left(n_1 + n_2\right) - 9} \right) d \tag{G.3}$$

[14] suggested that an effect size of $0.2$ be considered small, $0.5$ be considered medium, and $0.8$ be considered large. In Table 8, we present Hedges' $g$ statistic for the statistical tests in Section 5.2. We can clearly see that for homophilic datasets, there is a strong *positive effect* of using the dropping methods, but for heterophilic datasets and graph classification datasets, there is *at most* a small positive effect of using the dropping methods; rather, in most cases there is a *negative effect*.

### G.4 Performance of GIN with DropSens

In Section 5.3, we showed that when modelling long-range graph-classification tasks using GCNs, DropSens outperforms state-of-the-art graph-rewiring techniques designed for alleviating over-squashing. However, it does not perform as well with GIN, as can be seen in Table 9 – unsurprising, since DropSens was specifically designed to work with GCN's message-passing scheme.

**Table 7:** Difference in mean test accuracy (%) between the best performing configuration of each dropout method and the baseline NoDrop model, with GAT as the base model. Cell colors represent p-values from a t-test evaluating whether dropout improves performance.

**(a)** Node classification tasks.

| Dropout | Homophilic Networks | | | Heterophilic Networks | | |
|---|---|---|---|---|---|---|
| | Cora | CiteSeer | PubMed | Chameleon | Squirrel | TwitchDE |
| DropEdge | +0.584 | +0.668 | −0.075 | −3.059 | −3.164 | −0.363 |
| DropNode | +0.476 | −0.426 | −0.358 | −13.128 | −8.303 | −2.801 |
| DropAgg | +0.303 | +0.368 | −0.340 | −13.897 | −8.669 | −1.071 |
| DropGNN | +0.543 | +1.012 | −0.313 | −14.850 | −9.869 | −1.744 |
| Dropout | +1.061 | −0.140 | +0.081 | −5.851 | −6.390 | −0.981 |
| DropMessage | +0.987 | −0.425 | +0.183 | −2.344 | −2.534 | −0.114 |

**(b)** Graph classification tasks.

| Dropout | Molecular Networks | | | Social Networks | | |
|---|---|---|---|---|---|---|
| | Mutag | Proteins | Enzymes | Reddit | IMDb | Collab |
| DropEdge | −3.200 | +0.321 | +0.246 | −0.750 | −0.520 | +0.000 |
| DropNode | −1.700 | −0.429 | −2.914 | +1.440 | +1.280 | +0.000 |
| DropAgg | −1.100 | +0.696 | −1.632 | +0.160 | +0.280 | +0.000 |
| DropGNN | −5.600 | +0.536 | −3.340 | −0.360 | −0.520 | +0.000 |
| Dropout | −0.800 | −0.714 | −2.047 | +1.520 | +1.420 | +0.000 |
| DropMessage | −2.200 | −0.232 | +0.687 | +1.190 | +2.120 | +0.000 |

This highlights the main limitation of DropSens, necessitating architecture-specific alteration to the edge-dropping strategy, which is not practical in general.

## G.5 Best-performing Dropping Probabilities

For the real-world datasets in Section 5.2, we report the best performing dropping probability for different dataset−model−dropout combinations in Table 10.

**Table 8:** Hedges' $g$ statistic for different dataset−model−dropout combinations. Color-coding for effect size according to [14]; red denotes negative effect, and green denotes positive effect. Medium to large positive effect sizes in bold.

| | No effect | | Small effect | | Medium effect | | Large effect |

| Dataset | GNN | DropSens | DropEdge | DropNode | DropAgg | DropGNN | Dropout | DropMessage |
|---|---|---|---|---|---|---|---|---|
| GCN | Cora | **0.355** | **0.486** | **0.528** | 0.204 | **0.736** | **0.95** | 0.308 |
| | CiteSeer | **0.653** | **0.621** | −0.31 | **0.679** | **0.906** | −0.29 | 0.121 |
| | PubMed | 0.031 | −0.298 | −0.194 | −1.082 | −0.601 | **2.337** | **1.586** |
| | Chameleon | −0.782 | −0.638 | −1.97 | −4.462 | −1.465 | −1.352 | −0.59 |
| | Squirrel | −0.016 | −0.222 | −0.894 | −8.94 | −0.994 | −0.952 | −0.339 |
| | TwitchDE | 0.324 | −0.045 | −1.275 | −3.844 | −0.944 | −0.568 | 0.3 |
| | Actor | **0.499** | 0.301 | −0.143 | −0.507 | **0.693** | 0.202 | −0.322 |
| | Mutag | **0.524** | −0.126 | 0.0 | 0.121 | 0.292 | −0.281 | −0.022 |
| | Proteins | **0.532** | 0.11 | 0.194 | −0.061 | 0.104 | −0.02 | 0.089 |
| | Enzymes | −0.57 | −0.486 | −0.479 | −0.97 | −0.462 | −0.73 | −0.56 |
| | Reddit | −0.352 | −0.946 | −1.826 | −2.493 | −2.377 | −0.131 | −0.588 |
| | IMDb | 0.277 | 0.147 | −0.081 | 0.121 | −0.305 | 0.147 | **0.603** |
| | Collab | −0.314 | −0.742 | −1.158 | −3.612 | 0.194 | −0.775 | **0.658** |
| GIN | Cora | −0.735 | −2.983 | **2.424** | −1.474 | −0.525 | **3.556** | **3.296** |
| | CiteSeer | 0.09 | −7.019 | **2.523** | −7.432 | −6.995 | **1.962** | **1.732** |
| | PubMed | −0.486 | −1.558 | **1.031** | −1.185 | −1.447 | **1.961** | **1.71** |
| | Chameleon | −2.912 | −1.794 | −0.654 | −1.635 | −2.312 | −0.246 | **0.902** |
| | Squirrel | −0.837 | −0.499 | −0.318 | −1.823 | −1.998 | **0.391** | **0.92** |
| | TwitchDE | **1.056** | **0.35** | 0.17 | −0.426 | −0.025 | **0.475** | **0.775** |
| | Actor | 0.258 | **2.094** | −0.1 | **2.67** | **0.484** | 0.043 | −0.032 |
| | Mutag | 0.142 | −0.242 | −0.717 | 0.017 | −0.347 | 0.049 | 0.097 |
| | Proteins | −0.196 | −0.24 | 0.121 | 0.032 | −0.136 | 0.153 | 0.331 |
| | Enzymes | −2.424 | −1.272 | −1.068 | −0.086 | −1.646 | −0.293 | −0.336 |
| | Reddit | 0.313 | −0.183 | −0.641 | −0.608 | −1.563 | 0.17 | −0.053 |
| | IMDb | −0.626 | −0.048 | 0.078 | −0.159 | −0.527 | 0.015 | 0.252 |
| | Collab | −1.26 | −0.203 | 0.048 | −0.374 | −1.241 | 0.147 | **0.562** |
| GAT | Cora | **1.233** | **0.992** | **0.829** | **0.496** | **0.894** | **1.842** | **1.788** |
| | CiteSeer | **0.521** | **0.674** | −0.334 | **0.355** | **1.015** | −0.118 | −0.319 |
| | PubMed | 0.163 | −0.12 | −0.63 | −0.629 | −0.593 | 0.114 | 0.247 |
| | Chameleon | −0.811 | −0.782 | −3.488 | −3.824 | −4.049 | −1.334 | −0.563 |
| | Squirrel | −1.269 | −1.076 | −2.964 | −3.363 | −3.722 | −2.28 | −0.799 |
| | TwitchDE | −0.188 | −0.354 | −1.741 | −0.859 | −0.793 | −0.633 | −0.093 |
| | Actor | **1.138** | **1.143** | −0.079 | **1.202** | **1.074** | 0.201 | 0.265 |
| | Mutag | 0.206 | −0.333 | −0.184 | −0.116 | −0.556 | −0.09 | −0.234 |
| | Proteins | **0.507** | 0.081 | −0.109 | 0.162 | 0.136 | −0.193 | −0.061 |
| | Enzymes | 0.261 | 0.04 | −0.494 | −0.26 | −0.591 | −0.356 | 0.109 |
| | Reddit | 0.234 | −0.234 | **0.468** | 0.057 | −0.123 | **0.491** | 0.328 |
| | IMDb | **0.408** | −0.124 | 0.292 | 0.064 | −0.134 | 0.318 | **0.474** |
| | Collab | 0.0 | 0.0 | 0.0 | 0.0 | 0.0 | 0.0 | 0.0 |

**Table 9:** Performance of GIN with different rewiring methods in graph-classification tasks, following [8, 44]. First, second, and third best results are colored.

| Rewiring | Mutag | Proteins | Enzymes | Reddit | IMDb | Collab |
|---|---|---|---|---|---|---|
| None | 77.70 | 70.80 | 33.80 | 86.79 | 70.18 | **72.99** |
| Last FA | **83.45** | **72.30** | **47.40** | **90.22** | **70.91** | **75.06** |
| Every FA | 72.55 | 70.38 | 28.38 | 50.36 | 49.16 | 32.89 |
| DIGL | **79.70** | 70.76 | **35.72** | 76.04 | 64.39 | 54.50 |
| SDRF | 78.40 | 69.81 | **35.82** | 86.44 | 69.72 | 72.96 |
| FoSR | 78.00 | **75.11** | 29.20 | **87.35** | **71.21** | **73.28** |
| GTR | 77.60 | **73.13** | 30.57 | 86.98 | **71.28** | 72.93 |
| DropSens | **80.90** | 68.20 | 30.40 | **90.32** | 67.50 | 70.68 |

**Table 10:** Best performing dropout configuration – $q_{\max}$ and $c$ for DropSens, and $q$ for other dropping methods.

| GNN | Dataset | DropSens | DropEdge | DropNode | DropAgg | DropGNN | Dropout | DropMessage |
|-----|---------|----------|----------|----------|---------|---------|---------|-------------|
| GCN | Cora | 0.3, 0.8 | 0.9 | 0.4 | 0.9 | 0.9 | 0.6 | 0.4 |
| | CiteSeer | 0.8, 0.9 | 0.7 | 0.2 | 0.8 | 0.9 | 0.4 | 0.1 |
| | PubMed | 0.3, 0.9 | 0.1 | 0.1 | 0.1 | 0.3 | 0.6 | 0.8 |
| | Chameleon | 0.2, 0.9 | 0.1 | 0.2 | 0.1 | 0.1 | 0.1 | 0.2 |
| | Squirrel | 0.3, 0.9 | 0.1 | 0.1 | 0.1 | 0.1 | 0.3 | 0.2 |
| | TwitchDE | 0.2, 0.95 | 0.2 | 0.2 | 0.1 | 0.1 | 0.1 | 0.2 |
| | Actor | 0.3, 0.9 | 0.7 | 0.1 | 0.7 | 0.9 | 0.2 | 0.1 |
| | Mutag | 0.2, 0.9 | 0.4 | 0.2 | 0.5 | 0.1 | 0.1 | 0.7 |
| | Proteins | 0.2, 0.95 | 0.2 | 0.3 | 0.5 | 0.5 | 0.2 | 0.3 |
| | Enzymes | 0.2, 0.95 | 0.1 | 0.1 | 0.1 | 0.1 | 0.2 | 0.3 |
| | Reddit | 0.2, 0.95 | 0.1 | 0.1 | 0.1 | 0.1 | 0.1 | 0.1 |
| | IMDb | 0.5, 0.9 | 0.9 | 0.2 | 0.5 | 0.9 | 0.4 | 0.3 |
| | Collab | 0.5, 0.95 | 0.2 | 0.1 | 0.1 | 0.1 | 0.1 | 0.1 |
| GIN | Cora | 0.2, 0.9 | 0.1 | 0.4 | 0.1 | 0.1 | 0.5 | 0.8 |
| | CiteSeer | 0.2, 0.95 | 0.1 | 0.4 | 0.1 | 0.1 | 0.5 | 0.6 |
| | PubMed | 0.2, 0.95 | 0.1 | 0.3 | 0.1 | 0.1 | 0.7 | 0.8 |
| | Chameleon | 0.2, 0.9 | 0.1 | 0.1 | 0.1 | 0.1 | 0.2 | 0.4 |
| | Squirrel | 0.2, 0.9 | 0.1 | 0.2 | 0.1 | 0.1 | 0.1 | 0.8 |
| | TwitchDE | 0.8, 0.8 | 0.5 | 0.1 | 0.3 | 0.1 | 0.1 | 0.7 |
| | Actor | 0.5, 0.8 | 0.9 | 0.4 | 0.9 | 0.9 | 0.7 | 0.2 |
| | Mutag | 0.8, 0.95 | 0.1 | 0.4 | 0.2 | 0.1 | 0.1 | 0.1 |
| | Proteins | 0.2, 0.9 | 0.1 | 0.3 | 0.1 | 0.1 | 0.1 | 0.1 |
| | Enzymes | 0.8, 0.95 | 0.1 | 0.1 | 0.1 | 0.1 | 0.1 | 0.1 |
| | Reddit | 0.2, 0.95 | 0.1 | 0.1 | 0.1 | 0.1 | 0.1 | 0.3 |
| | IMDb | 0.2, 0.9 | 0.1 | 0.3 | 0.3 | 0.1 | 0.1 | 0.6 |
| | Collab | 0.2, 0.9 | 0.1 | 0.1 | 0.1 | 0.1 | 0.1 | 0.6 |
| GAT | Cora | 0.8, 0.5 | 0.7 | 0.2 | 0.8 | 0.5 | 0.6 | 0.7 |
| | CiteSeer | 0.5, 0.95 | 0.9 | 0.1 | 0.9 | 0.9 | 0.1 | 0.1 |
| | PubMed | 0.2, 0.95 | 0.1 | 0.1 | 0.1 | 0.1 | 0.6 | 0.8 |
| | Chameleon | 0.2, 0.95 | 0.1 | 0.1 | 0.1 | 0.1 | 0.1 | 0.1 |
| | Squirrel | 0.2, 0.95 | 0.1 | 0.1 | 0.1 | 0.1 | 0.1 | 0.1 |
| | TwitchDE | 0.2, 0.95 | 0.1 | 0.1 | 0.1 | 0.6 | 0.1 | 0.1 |
| | Actor | 0.8, 0.5 | 0.8 | 0.1 | 0.9 | 0.8 | 0.1 | 0.1 |
| | Mutag | 0.5, 0.95 | 0.7 | 0.2 | 0.1 | 0.6 | 0.2 | 0.2 |
| | Proteins | 0.5, 0.8 | 0.7 | 0.1 | 0.8 | 0.5 | 0.1 | 0.4 |
| | Enzymes | 0.8, 0.95 | 0.1 | 0.1 | 0.1 | 0.3 | 0.2 | 0.2 |
| | Reddit | 0.8, 0.9 | 0.8 | 0.6 | 0.1 | 0.8 | 0.9 | 0.8 |
| | IMDb | 0.3, 0.95 | 0.8 | 0.9 | 0.3 | 0.3 | 0.4 | 0.6 |
| | Collab | 0.2, 0.9 | 0.1 | 0.1 | 0.1 | 0.1 | 0.1 | 0.1 |

