# OpenReview forum: "Effects of Dropout on Performance in Long-range Graph Learning Tasks"
_NeurIPS.cc/2025/Conference — NeurIPS 2025 poster_

### Official Review · Reviewer_RHiq · 2025-06-14

**Clarity:** 3
**Significance:** 2
**Originality:** 2
**Rating:** 4
**Confidence:** 4

**Summary:**

Graph neural networks are known to suffer from issues like over-squashing (due to topological bottlenecks) which affect long-range interactions and over-smoothing, wherein repeated rounds of information aggregation results in node representations losing their discriminative power. This work explores the connections between Dropout style algorithms such DropEdge, DropNode, DropAgg and DropGNN and over-squashing. While Drop-out style algorithms have been shown to mitigate over-smoothing and improve downstream performance, this current work highlights the limitations of such methods by showing how these methods desensitize distant nodes to each other consequently making over-squashing worse. As a resolution DropSens is proposed which is a sensitivity-aware variant of DropEdge that can control the proportion of information lost due to edge-dropping, ensuring long-range interactions between distant nodes is enhanced. Extensive experiments on different datasets and GNN (GCN and GIN) models confirm their hypothesis.

**Questions:**

**See weaknesses**

**Ethical Concerns:**

["NO or VERY MINOR ethics concerns only"]

**Final Justification:**

The authors have done a good job of providing with extra experiments. However, I maintain my position on the fundamental motivation: demonstrating that random dropout-style algorithms perform poorly on long-range tasks—when they were designed for entirely different purposes—feels like evaluating the wrong tool for the wrong job. Further, results on LRGB which actually have long range interactions suggest the proposed DropSens (and Drop-out variants like DropEdge) perform poorly compared to NoDrop also reinforces my initial concern about the applicability of such methods for modeling long range interactions.
I have increased my score from 2 to 4 in recognition of the authors' responses and the technical merit of their approach, but I am not entirely convinced to champion this paper for acceptance.

**Limitations:**

Yes

**Paper Formatting Concerns:**

L904 in the appendix has missing subsection content only a heading called - "Why influence scores?".

**Quality:**

2

**Strengths And Weaknesses:**

**Strengths**

1. The characterization of interaction between random dropout style algorithms and over-squashing via the sensitivity analysis is really interesting.

2. The proposed DropSens algorithm seems to increase the sensitivity of long-range nodes and thus possibly help with over-squashing.

3. The paper is well-written and well-structured.

**Weaknesses**

1. While I understand the whole dichotomy between over-squashing and over-smoothing trade-off, I am not really sure if the current motivation of the work is strong enough. The sensitivity analysis has already been proposed in [1] and [2] and making use of this analysis to show that dropout style algorithms might make over-squashing worse while is an interesting observation but its perhaps not entirely unexpected, since the dropout style algorithms were originally proposed with different design goals.

2. The discussion on sensitivity analysis (Figure 1) could use a little more nuance. How would the sensitivity analysis look like for a heterophilic dataset?  Does random edge dropping affect **different-class neighbors** more than **same-class neighbors** (which would be good)? Or does it reduce both? And thus could potentially explain why we see no significant results on heterophilic datasets?

3. The paper argues that random edge dropping harms long-range tasks, and the results on heterophilic datasets are presented as evidence. However, one might intuitively expect dropping a small number of edges might be beneficial in heterophilic settings, since its more likely we prune connections between nodes of different labels? For instance, the work talks about absolute sensitivity reduction between distant nodes through random dropout style algorithms, but it is possible deleting a small number of edges in a heterophilic setting might reduce the influence of noisy neighbours and thus might improve relative sensitvity between same-class distant nodes?

4. Experiments on Long Range Graph Benchmark datasets (LRGB) [3] [4] are missing. I think this is an important benchmark to include if the theme is on long range interactions. It would be interesting to see how these Dropout style algorithms including the proposed DropSens perform when it comes to LRGB datasets. I understand the datasets in LRGB have different tasks but PASCAL-VOC is a node classification task and datasets such as peptides-func and peptides-struct are graph classification and regression tasks respectively and ideally should have no problem including them in the experiments (since results on graph classification is already included in the work anyway).

5. I think the work currently assumes somewhat pessimistic view - **dropout style algorithms which remove edges are detrimental**, but what about strategic edge deletions? For instance [5] shows that specific edge deletions can increase the spectral gap via the Braess paradox and thus can mitigate over-squashing while not making over-smoothing worse?



**References**

1. Understanding over-squashing and bottlenecks on graphs via curvature. Topping et al, ICLR 2022.

2. On Over-Squashing in Message Passing Neural Networks: The Impact of Width, Depth, and Topology. Di Giovanni et al, ICML 2023.

3. Long Range Graph Benchmark, Dwivedi et al 2022.

4. Where Did the Gap Go? Reassessing the Long-Range Graph Benchmark, Toenshoff et al, TMLR 2023.

5. Spectral Graph Pruning Against Over-squashing and Over-smoothing, Jamadandi et al, NeurIPS, 2024.

---

> ### Author Rebuttal · Authors · 2025-07-26
>
> Thank you very much for the insightful summary, for recognizing the strengths of our work, and for the overall feedback!
>
> Before we address your concerns, we wanted to bring to your attention that we had accidentally omitted the DropSens configuration \$(c, q\_{\max}) = (0.8, 0.3)\$ from our original hyperparameter sweep. This has now been included, and the results have been updated accordingly. The overall findings remain unchanged, with the exception that **DropSens now ranks 3rd on the Proteins dataset**.
>
> ---
>
> Kindly find our responses to the concerns raised:
>
> > [...] making use of this analysis to show that dropout style algorithms might make over-squashing worse [...] perhaps not entirely unexpected, since the dropout style algorithms were originally proposed with different design goals.
>
> Thank you for sharing your perspective. We agree that it may seem intuitive that dropout-style methods reduce sensitivity between distant nodes. However, **our theoretical analysis serves a deeper purpose** than simply confirming this behavior: it provides a concrete understanding of ***why*** and ***how*** these methods fall short in the context of long-range tasks, and it **motivates the design of DropSens as a principled alternative**.
>
> While it may not be surprising that these methods diminish long-range sensitivity, the key concern we highlight is that **this fundamentally undermines their utility for the very tasks that deeper GNNs aim to solve**:
>
> *The original motivation behind alleviating over-smoothing was to enable training of deeper MPNNs, which are only necessary when long-range dependencies matter. But if dropout-based methods worsen over-squashing, they negate this benefit &ndash; deep models may remain trainable (i.e., node representations don't collapse), but the model is still unable to leverage information from distant nodes for prediction.*
>
> This, in our view, is a **critical and underappreciated limitation**. We are not claiming that these methods are ineffective in general, but that **they should *not* be applied in isolation when long-range interactions are important**. Our hope is this analysis will motivate practitioners to thoroughly assess the effects of methods designed for alleviating over-smoothing and training deep GNNs, with regards to their effects on over-squashing.
>
> > How would the sensitivity analysis look like for a heterophilic dataset? [...] could potentially explain why we see no significant results on heterophilic datasets?
>
> Thank you for these thoughtful questions. First, we’d like to clarify a possible misconception: **our sensitivity analysis (as shown in Figure 1) does not depend on node labels**. It measures the norm of the Jacobian of the final node representations (e.g., logits or regressands) with respect to the input node features. As such, **it is agnostic to whether a dataset is homophilic or heterophilic** &ndash; the analysis captures how much a node’s output depends on the input features of other nodes, regardless of class.
>
> Second, **random edge-dropping methods like DropEdge are entirely blind to node or edge attributes**, including class membership. They drop edges uniformly at random, and thus **reduce influence from both same-class and different-class neighbors indiscriminately**. This uniformity could indeed be suboptimal, and it points toward a promising direction, as you hint: *incorporating structural or label-aware heuristics to guide edge dropping* may help preserve task-relevant information more effectively, especially in settings where homophily cannot be assumed.
>
> As for why dropout methods perform much worse on heterophilic datasets than on homophilic datasets, we refer to Section 5.2:
>
> "*[...] homophilic datasets have local consistency in node labels, i.e. nodes closely connected to each other have similar labels. On the other hand, in heterophilic datasets, nearby nodes often have dissimilar labels. Since DropEdge-variants increase the sensitivity of a node’s representations to its immediate neighbors, and reduce its sensitivity to distant nodes, we expect it to improve performance on homophilic datasets but harm performance on heterophilic ones […]*"
>
> > [...] one might intuitively expect dropping a small number of edges might be beneficial in heterophilic settings [...] might reduce the influence of noisy neighbours and thus might improve relative sensitvity between same-class distant nodes?
>
> Thank you for this thoughtful observation. We understand the intuition that dropping a small number of edges might reduce the influence of noisy (i.e., different-class) neighbors in heterophilic settings. However, we believe this intuition overlooks a key limitation of random edge-dropping methods like DropEdge: **they do not use any information from node features, class labels, or graph structure when sampling edge masks**. Edges are dropped independently and uniformly at random, with no consideration for whether an edge connects same-class or different-class nodes.
>
> This means that while it is possible that a given sampled mask might coincidentally drop some "noisy" edges, **the probability of consistently retaining the correct paths for information flow &ndash; especially between distant same-class nodes &ndash; is extremely low**. This is because successful long-range propagation typically requires *entire paths* of edges to be active in the forward pass. Dropping even a single edge on such a path breaks the flow of information. Hence, random dropout is more likely to disrupt useful long-range communication than improve it, particularly in heterophilic graphs where same-class nodes tend to be farther apart.
>
> This is precisely the phenomenon our sensitivity analysis and experiments aim to highlight: dropout methods, while helpful in alleviating over-smoothing, can severely exacerbate over-squashing by indiscriminately pruning edges &ndash; thereby reducing a node’s sensitivity to informative distant features.
>
> Ultimately, **if the goal is to reduce influence from different-class neighbors while preserving important long-range same-class connections, this requires structured or informed edge-dropping, not random sampling**. Our work points toward this need by showing the limitations of uniform random edge-drop methods in heterophilic and long-range settings.
>
> > Experiments on Long Range Graph Benchmark are missing [...]
>
> Thank you for this thoughtful suggestion. We fully agree that the LRGB benchmark is highly relevant for evaluating long-range reasoning capabilities, and we appreciate the opportunity to clarify our decision.
>
> First, we would like to highlight the computational cost of our experiments. For each *dataset $\times$ GNN $\times$ dropping method* combination, we conducted 20 runs across 9 different dropping probabilities, followed by 30 additional runs for the best configuration (see "Number of Independent Runs" under Appendix E.2 to learn why we take 50 runs for evaluation), totaling **210 runs per setting**. Extending this setup to large-scale LRGB datasets, which involve significantly larger graphs and longer training times, was unfortunately beyond our available compute budget.
>
> Second, **our current choice of datasets was motivated by the desire to make direct and meaningful comparisons with prior works on graph rewiring for alleviating over-squashing** (Black et al., 2023; Karhadkar et al., 2023; Topping et al., 2022). These prior works have widely adopted the benchmark suite used in our study, so evaluating on these datasets allowed us to assess DropSens in the context of established baselines.
>
> That said, we recognize the value of testing dropout-style methods on LRGB tasks, and we agree that it is an exciting direction, especially since the LRGB datasets include both node- and graph-level tasks, making them a natural complement to our current benchmark. For that reason, **we are aiming to share a comparison between NoDrop, DropEdge and DropSens on the PeptidesStruct graph-regression dataset** by the end of discussion period. We would appreciate your patience as we work on it.
>
> > I think the work currently assumes somewhat pessimistic view [...] can mitigate over-squashing while not making over-smoothing worse?
>
> Thank you for sharing this reference &ndash; we weren’t previously aware of it, and we found it very insightful. We fully agree that *strategic* edge deletion offers a promising approach to addressing both over-smoothing and over-squashing. In fact, this aligns with our broader message: **it’s not the act of edge deletion itself that is harmful, but *how* it is done**.
>
> **Our work critiques dropout-style methods like DropEdge not because they remove edges per se, but because they do so uniformly at random**, without regard to the underlying graph topology or feature space. By contrast, **methods that perform *informed* or *topology-aware* edge deletion** &ndash; including those leveraging phenomena like the Braess paradox to enhance spectral properties &ndash; **can potentially mitigate over-squashing without exacerbating over-smoothing**. We view such methods as complementary to our analysis, and have discussed a few works on unified treatment of over-smoothing and over-squashing in Appendix A.4; we have also added the reference you shared in this section.
>
> ---
>
> **References**:
>
> Mitchell Black, Zhengchao Wan, Amir Nayyeri, and Yusu Wang. *Understanding oversquashing in GNNs through the lens of effective resistance*. In International Conference on Machine Learning, 2023.
>
> Kedar Karhadkar, Pradeep Kr. Banerjee, and Guido Montufar. *FoSR: First-order spectral rewiring for addressing oversquashing in GNNs*. In International Conference on Learning Representations, 2023.
>
> Jake Topping, Francesco Di Giovanni, Benjamin Paul Chamberlain, Xiaowen Dong, and Michael M. Bronstein. *Understanding over-squashing and bottlenecks on graphs via curvature*. In International Conference on Learning Representations, 2022.

---

> > ### Comment · Reviewer_RHiq · 2025-08-02
> > **Thank you for the rebuttal**
> >
> > I would like to thank the authors for their responses. Some of my concerns still remain
> >
> > 1. As I said earlier in my review, my main concern is that the paper is highlighting an observation for a method that was not intended to be used in the long range context in the first place as a *fundamental flaw of Drop-out like methods*. To elaborate, all the dropout-style methods have primarily been evaluated on homophilic graphs [1],[2],[3] (while [3] also shows results on graph classification which have LRIs, but there I think the idea of model expressivity is also under consideration) where the label/feature of nodes are homophilic and LRIs might not be a problem and Drop-out style methods *work* and they also *meet their intended design goal of training deeper GNNs*. By showing they perform poorly on heterophilic graphs, the work essentially confirms an expected outcome rather than uncovering a surprising flaw. This makes the motivation feel less like a novel insight and more like an application of a known tool to the wrong problem.
> >
> > 2. I understand that the sensitivity analysis does not depend on the node labels and relies on the node features, but my motivation to ask that question was about how the concept of heterophily is not limited to class labels; it can also refer to feature heterophily, where connected nodes have dissimilar feature distributions [4]. In many real-world heterophilic graphs, the *heterophily-ness* is a combined effect of both label and feature dissimilarity. I believe presenting a sensitivity analysis on a heterophilic graph would be a valuable addition. It would help us understand whether the observed reduction in sensitivity just harms all long-range connections equally, or if it has a more nuanced, feature-dependent effect that could better explain the performance on these graphs.
> >
> > 3. I also think the paper could benefit from analysing what kind of edges are more likely dropped when applying Drop-out style algorithms to heterophilic datasets (same-class vs different class), i understand that the method is label agnostic, but what I am failing to understand is - in a heterophilic graph the node/feature heterophily patterns are more complex than the current theoretical analysis suggests. The analysis only identifies two regimes - immediate neighbors vs. long-distance interactions - but both of these categories can contain beneficial or harmful connections depending on label relationships. For instance, in a heterophilic graph dropping an edge increases the sensitvity to nearby neighbours which might be bad but if the long range interaction we are trying to capture also has a different label then it could be beneficial vs if that node had a same label, then its bad. Which brings me back to my original doubt about applying random drop-out style algorithms for heterophilic graphs in the first place. If the method was optimizing some objective for instance spectral gap then it wouldn't matter if we dropped same-label/different-label edges because we are concerned with optimizing a global objective of information flow (which the spectral gap characterizes) and which random-style algorithms lack.
> >
> >
> >
> >
> >
> >
> >
> > References:
> >
> > 1. DropEdge: Towards Deep Graph Convolutional Networks on Node Classification. Rong et al. ICLR 2020.
> >
> > 2. Graph Random Neural Networks for Semi-Supervised Learning on Graphs. Feng et al. NeurIPS 2020.
> >
> > 3. DropGNN: Random Dropouts Increase the Expressiveness of Graph Neural Networks. Papp et al. NeurIPS 2021.
> >
> > 4. What Is Missing In Homophily? Disentangling Graph Homophily For Graph Neural Networks. Zheng et al NeurIPS 2024.

---

> ### Author Response · Authors · 2025-08-02
>
> Thank you very much for the response!
>
> > [...] all the dropout-style methods have primarily been evaluated on homophilic graphs [...] where the label/feature of nodes are homophilic and LRIs might not be a problem and Drop-out style methods *work* and they also *meet their intended design goal of training deeper GNNs*. [...] This makes the motivation feel less like a novel insight and more like an application of a known tool to the wrong problem.
>
> Thank you for raising this concern. We believe the disagreement stems from a different understanding of the **purpose behind dropout-style methods and deep MPNNs**.
>
> Our work focuses on message-passing neural networks (MPNNs), where **depth is synonymous with reach**, since the number of layers directly governs how far information can propagate in the graph. Therefore, we believe **the motivation for training deep MPNNs is *precisely* to model long-range interactions**. In fact, using deep MPNNs on tasks with only short-range dependencies may result in *dataset-model misalignment*. This is why we argue that **evaluating such methods solely on homophilic datasets** (where long-range signals are not necessary) **paints an incomplete picture**. As we discuss in Appendix A.2, this has led to relatively little attention being paid to how these methods affect performance on tasks that *do* require long-range modelling.
>
> Since dropout-based methods *compromise sensitivity to distant nodes*, as we show both theoretically and empirically, **the very reason for going deeper is undermined**. The model may remain trainable by mitigating over-smoothing, but it cannot leverage the long-range signals, defeating the very purpose of going deeper.
>
> Does this clarify the motivation behind our work, and the significance of our findings?
>
> > [...] it can also refer to feature heterophily, where connected nodes have dissimilar feature distributions [...] It would help us understand whether the observed reduction in sensitivity just harms all long-range connections equally, or if it has a more nuanced, feature-dependent effect that could better explain the performance on these graphs.
>
> Ah, I see &ndash; that makes total sense! Just to make sure we are on the same page, you're suggesting that comparing the sensitivity profiles of homophilic and heterophilic datasets is helpful because homophily extends beyond node labels, so despite being label-agnostic, sensitivity analysis may offer an explanation for the difference in performances on these datasets &ndash; correct?
>
> Would a **comparison between the sensitivity profiles of PubMed (edge- and feature-homophilic, as suggested by Zheng et al. (2024, Table 3)) and Chameleon (edge- and feature-heterophilic)** help address your concern?
>
> > I also think the paper could benefit from analysing what kind of edges are more likely dropped when applying Drop-out style algorithms to heterophilic datasets (same-class vs different class) [...] For instance, in a heterophilic graph dropping an edge increases the sensitvity to nearby neighbours which might be bad but if the long range interaction we are trying to capture also has a different label then it could be beneficial vs if that node had a same label, then its bad.
>
> If I understand correctly, the question here is whether random edge-dropping affects same-class and different-class node pairs differently, particularly in the context of heterophilic graphs. In other words, you're pointing to how the topology of heterophilic datasets may influence which types of connections are disproportionately impacted by dropout.
>
> Would it be helpful if we conducted **an analysis similar to what we did for Cora, but using Chameleon instead, and with same-class and different-class node pairs grouped and analyzed separately**? This would allow us to see whether dropout has different effects depending on the label relationship between nodes.
>
> ---
>
> Yilun Zheng, Sitao Luan, and Lihui Chen. *What Is Missing In Homophily? Disentangling Graph Homophily For Graph Neural Networks*. In Annual Conference on Neural Information Processing Systems, 2024.

---

> > ### Comment · Reviewer_RHiq · 2025-08-02
> > **Response**
> >
> > Thank you for your response. Some clarifications
> >
> > 1. I think the discussion here is a little nuanced. Lets break this down into 2 parts
> >
> > a) Firstly, increasing model depth does not help with long-range interactions, as authors in [1] show it quickly leads to vanishing gradients and of course there is the problem of over-smoothing. On the other hand, graph rewiring has largely been proposed to mitigate over-squashing/help model long-range interactions. All of these techniques modify the graph in a principled way either based on Ricci curvature, spectral gap, effective resistance etc. which optimize for certain topological properties of the graph, for instance, spectral gap is tied to how well information can flow in a graph and is guaranteed to help with long-range interactions. Random edge modifications do not have such guarantees.
> >
> >
> > b) Consequently, methods that talk about training deeper GNN models do not have the objective of *modeling long-range interactions*. For instance in [2] the DropEdge acts as a graph augmentor and is supposed to help with over-fitting and over-smoothing and thus allows for training deeper GNNs, similarly in [3] DropGNN helps with increasing model expressivity (Weisfeler-Leman tests etc) they dont even train a deeper GNN model, in [4] the goal is to reduce memory consumption and the OGB datasets they use are all largely homophilic.
> >
> > This is precisely why I believe the motivation is somewhat limited. Random dropout-style methods have different objectives. While it would be an interesting experiment to see what happens when such methods are applied to heterophilic datasets, this does not necessarily mean *it fundamentally undermines their utility for the very tasks that deeper GNNs aim to solve.*
> >
> > 2. Yes exactly! I think having a figure like Figure 1 for a heterophilic dataset should provide more insights.
> >
> > 3. Yes, I think that would provide valuable insight into if Random drop-out style algorithms affect all types of edges indiscriminately or are more likely to affect certain types of edges (meaning connecting same-label nodes vs different label nodes), if for instance DropSens affected different label nodes more often than same-labels then it can also act as a message-passing reducer effectively allowing you to also mitigate over-smoothing and possibly show training deeper GNNs is possible? I'm not asking the authors to conduct this experiment - this is just another line of reasoning that might strengthen the paper's claims.
> >
> > I maintain my concerns about the fundamental motivation and methodology.
> >
> > References :
> >
> > [1] On Over-Squashing in Message Passing Neural Networks: The Impact of Width, Depth, and Topology. DiGiovanni et al. ICML 2023.
> >
> > [2] DropEdge: Towards Deep Graph Convolutional Networks on Node Classification. Rong et al. ICLR 2020.
> >
> > [3] DropGNN: Random Dropouts Increase the Expressiveness of Graph Neural Networks. Papp et al. NeurIPS 2021.
> >
> > [4] Training Graph Neural Networks with 1000 Layers. Li et al. ICML 2021.

---

> ### Author Response · Authors · 2025-08-02
> **Rebuttal**
>
> First off, thank you very much for actively engaging in the discussion &ndash; we really appreciate the thoughtful feedback!
>
> We're currently working on adding **experimental results for heterophilic datasets**, but in the meantime, we wanted to briefly respond to your first point.
>
> > Firstly, increasing model depth does not help with long-range interactions.
>
> We mostly agree &ndash; **increasing depth *does* improve the reach of MPNNs, *but* problems like over-smoothing and over-squashing limit their potential in practice**. That’s precisely why much of the community, including us, is interested in mitigating these issues.
>
> > Graph rewiring has largely been proposed to mitigate over-squashing [...] these techniques modify the graph in a principled way [...] Random edge modifications do not have such guarantees.
>
> We agree that dropout-based techniques don't provide formal guarantees for information flow. Therefore, **their detrimental effects on over-squashing doesn’t invalidate the claims made by those works** &ndash; namely, that they alleviate over-smoothing and improve the trainability of deeper MPNNs.
>
> > Consequently, methods that talk about training deeper GNN models do not have the objective of modeling long-range interactions. For instance in [2] the DropEdge acts as a graph augmentor and is supposed to help with over-fitting and over-smoothing and thus allows for training deeper GNNs [...]
>
> Respectfully, we’d disagree. The motivation for deep MPNNs largely stems from the goal of capturing long-range interactions (LRIs). As noted in Li et al. (2018), a foundational work on over-smoothing:
>
> "*Since the graph convolution is a localized filter [...] a shallow GCN cannot sufficiently propagate the label information to the entire graph with only a few labels. [...] the accuracy of GCNs decreases much faster than the accuracy of label propagation. [...] it reflects the inability of the GCN model in exploring the global graph structure.*"
>
> **If the task were inherently short-range**, there would be little reason to train deep MPNNs, i.e. execute multiple message-passing steps, at all. In such cases, **improving the capacity of the message, aggregation or update functions** (e.g., using deeper MLPs as update functions in GIN), **while keeping the number of message-passing steps low**, would be a more effective and efficient alternative, and **would avoid the challenge of over-smoothing altogether**.
>
> Alon & Yahav (2021) make a similar point when distinguishing between short- and long-range tasks:
>
> "*Over-smoothing was mostly demonstrated in short-range tasks [...] &ndash; tasks that have small problem radii, where a node’s correct prediction mostly depends on its local neighborhood. [...] Since the learning problems depend mostly on short-range information in these datasets, it makes sense why more layers than the problem radius might be extraneous.*"
>
> This framing further supports our view that the primary motivation for alleviating over-smoothing and training deeper GNNs is, indeed, to capture LRIs.
>
> Finally, regarding the point that random dropout-like algorithms target overfitting: while we acknowledge their effectiveness on short-range tasks, **in the context of long-range tasks, their benefits are often overshadowed by increased over-squashing and a tendency to overfit to short-range signals**. We explore this in Appendix F, and request you to check Figures 7 and 9 for supporting evidence.
>
> > [...] similarly in [3] DropGNN helps with increasing model expressivity (Weisfeler-Leman tests etc) they dont even train a deeper GNN model [...]
>
> We agree, and thank you for pointing this out &ndash; we apologize for the oversight. We've now **added a clarification noting that while DropGNN falls under the broader class of edge-dropping algorithms, its design was not motivated by the goal of alleviating over-smoothing** or enabling the training of deeper GNNs.
>
> ---
>
> Uri Alon and Eran Yahav. *On the bottleneck of graph neural networks and its practical implications*. In International Conference on Learning Representations, 2021.
>
> Qimai Li, Zhichao Han, and Xiao-ming Wu. *Deeper insights into graph convolutional networks for semi-supervised learning*. Proceedings of the AAAI Conference on Artificial Intelligence, 2018.

---

> ### Author Response · Authors · 2025-08-02
> **Rebuttal (Continued)**
>
> > [...] this does not necessarily mean it fundamentally undermines their utility for the very tasks that deeper GNNs aim to solve.
>
> Kindly see our remark above. To reiterate, we are not suggesting that these algorithms are without merit &ndash; they are effective at preventing overfitting and can help alleviate over-smoothing, thereby improving the trainability of deeper MPNNs. However, this often comes at the cost of exacerbating over-squashing, which limits their suitability for long-range tasks &ndash; the very setting where deeper MPNNs are most needed.
>
> Instead, **we view these techniques primarily as different regularizers that can be complementary to methods specifically aimed at alleviating over-squashing**. For instance, Dropout was used alongside graph rewiring approaches in works such as Black et al. (2023); Karhadkar et al. (2023); and Topping et al. (2022).
>
> We hope this makes the our motivation clear.
>
> ---
>
> **References**
>
> Mitchell Black, Zhengchao Wan, Amir Nayyeri, and Yusu Wang. *Understanding oversquashing in GNNs through the lens of effective resistance*. In International Conference on Machine Learning, 2023.
>
> Kedar Karhadkar, Pradeep Kr. Banerjee, and Guido Montufar. *FoSR: First-order spectral rewiring for addressing oversquashing in GNNs*. In International Conference on Learning Representations, 2023.
>
> Jake Topping, Francesco Di Giovanni, Benjamin Paul Chamberlain, Xiaowen Dong, and Michael M. Bronstein. *Understanding over-squashing and bottlenecks on graphs via curvature*. In International Conference on Learning Representations, 2022.

---

> ### Author Response · Authors · 2025-08-03
> **Sensitivity Analysis -- Homophilic vs Heterophilic Datasets**
>
> Below, we present the average sensitivity under DropEdge, normalized by average sensitivity with NoDrop, to understand how the relative sensitivities vary for homophilic and heterophilic datasets:
>
> | Dataset | $d_G(j, i) =$ 0 | 1 | 2 | 3 | 4 | 5 | 6 |
> | :-: | :-: | :-: | :-: | :-: | :-: | :-: | :-: |
> Cora | **1.783e+00** | **1.165e+00** | 9.752e-01 | 7.570e-01 | 6.473e-01 | 4.837e-01 | 2.519e-01 |
> CiteSeer | **1.727e+00** | **1.045e+00** | 9.798e-01 | 6.984e-01 | 5.559e-01 | 3.672e-01 | 1.946e-01 |
> PubMed | **3.511e+00** | **1.022e+00** | **1.089e+00** | 6.777e-01 | 6.151e-01 | 4.290e-01 | 4.576e-01 |
> Chameleon | **3.034e+00** | **1.295e+00** | 9.625e-01 | 8.401e-01 | 7.816e-01 | 5.815e-01 | 6.126e-01 |
> Squirrel | **1.602e+00** | **1.429e+00** | **1.235e+00** | 8.854e-01 | 8.444e-01 | 7.787e-01 | 6.763e-01 |
> TwitchDE | **5.657e+00** | **2.357e+00** | **1.203e+00** | 9.466e-01 | 1.044e+00 | 9.640e-01 | 4.968e-01 |
> Actor | **2.493e+00** | **1.663e+00** | **1.289e+00** | 8.653e-01 | 7.941e-01 | 7.081e-01 | 5.314e-01 |
>
> It is interesting to note that the **relative sensitivity is higher for heterophilic datasets than for homophilic datasets**. That is, the graph topology and node features enable LRIs to be modelled better for heterophilic datasets than for homophilic datasets.
>
> **Yet, relative sensitivity falls below 1** (i.e., sensitivity under DropEdge is lower than under NoDrop) **for nodes more than 2-hops away**. This is not much of an issue for homophilic datasets, which are inherently short-range tasks, but it *explains the poor performance on heterophilic datasets*, which require modelling LRIs.
>
> ---
>
> Next, we observe how DropSens improves on DropEdge by comparing their sensitivity profiles:
>
> | Dataset | $d_G(j, i) =$ 0 | 1 | 2 | 3 | 4 | 5 | 6 |
> | :-: | :-: | :-: | :-: | :-: | :-: | :-: | :-: |
> Cora | 7.312e-01 | **1.025e+00** | **1.067e+00** | **1.123e+00** | **1.094e+00** | **1.155e+00** | **1.228e+00** |
> CiteSeer | 6.768e-01 | **1.066e+00** | **1.079e+00** | **1.318e+00** | **1.305e+00** | **1.633e+00** | **2.038e+00** |
> PubMed | 3.325e-01 | **1.025e+00** | 9.755e-01 | **1.387e+00** | **1.478e+00** | **1.708e+00** | **1.658e+00** |
> Chameleon | 5.106e-01 | 8.513e-01 | **1.042e+00** | **1.117e+00** | **1.198e+00** | **1.330e+00** | **1.174e+00** |
> Squirrel | **1.043e+00** | 9.056e-01 | 9.158e-01 | **1.060e+00** | **1.034e+00** | **1.019e+00** | **1.052e+00** |
> TwitchDE | 4.012e-01 | 6.809e-01 | 9.326e-01 | **1.041e+00** | 9.943e-01 | 9.586e-01 | **1.295e+00** |
> Actor | 5.030e-01 | 7.333e-01 | 9.069e-01 | **1.162e+00** | **1.153e+00** | **1.170e+00** | **1.353e+00** |
>
> It is clear to observe that **DropSens allows node representations to be more sensitive to distant ($d \geq 3$) nodes' features, than DropEdge**, thereby improving performance on heterophilic datasets.
>
> What is interesting is that **for homophilic datasets, DropSens is more sensitive to *nearby nodes* than DropEdge**, allowing the model to effectively capture short-range interactions. Although it is poorer at capturing short-range interactions for heterophilic datasets, the table below shows that DropSens improves over NoDrop on that front, similar to DropEdge:
>
> | Dataset | $d_G(j, i) =$ 0 | 1 | 2 | 3 | 4 | 5 | 6 |
> | :-: | :-: | :-: | :-: | :-: | :-: | :-: | :-: |
> Cora | **1.304e+00** | **1.193e+00** | **1.040e+00** | 8.503e-01 | 7.081e-01 | 5.586e-01 | 3.094e-01 |
> CiteSeer | **1.169e+00** | **1.114e+00** | **1.057e+00** | 9.202e-01 | 7.254e-01 | 5.997e-01 | 3.966e-01 |
> PubMed | **1.167e+00** | **1.047e+00** | **1.063e+00** | 9.399e-01 | 9.090e-01 | 7.325e-01 | 7.586e-01 |
> Chameleon | **1.549e+00** | **1.102e+00** | **1.003e+00** | 9.387e-01 | 9.367e-01 | 7.732e-01 | 7.189e-01 |
> Squirrel | **1.671e+00** | **1.294e+00** | **1.131e+00** | 9.383e-01 | 8.729e-01 | 7.931e-01 | 7.118e-01 |
> TwitchDE | **2.269e+00** | **1.605e+00** | **1.122e+00** | 9.856e-01 | 1.038e+00 | 9.241e-01 | 6.436e-01 |
> Actor | **1.254e+00** | **1.219e+00** | **1.169e+00** | **1.005e+00** | 9.157e-01 | 8.286e-01 | 7.191e-01 |
>
> ---
>
> This set of experiments suggests that **DropSens strikes a perfect balance between NoDrop and DropEdge**:
> 1. improves sensitivity to nearby nodes ($d \leq 2$), over NoDrop
> 2. improves sensitivity to distant nodes ($d \geq 3$), over DropEdge
> 3. mitigates over-fitting (by stochastic regularization) and over-smoothing (by reducing message-passing), just as DropEdge

---

> > ### Comment · Reviewer_RHiq · 2025-08-04
> > **Thanks for the response**
> >
> > Thank you for your responses. I appreciate the authors taking time to conduct extra experiments. I am satisfied with most of these answers and the new experiments are also interesting. I will be increasing my score.

---

> > > ### Author Response · Authors · 2025-08-04
> > > **New Results**
> > >
> > > Thank you very much for acknowledging the new results!
> > >
> > > We also request you to take a look at our official comment, where we have updated DropSens rankings compared to graph-rewiring methods, and also added results for the PeptidesStruct (LRGB) dataset.

---

### Official Review · Reviewer_5aYk · 2025-06-26

**Clarity:** 3
**Significance:** 2
**Originality:** 3
**Rating:** 4
**Confidence:** 2

**Summary:**

The paper looks at various dropout methods for graph NNs. These method have been
introduced to try and alleviate the over-smoothing problem. The paper postulates
that these methods might actually make another problem, over-squashing, worse,
because they introduce more bottlenecks in the graph. The authors show this is
in fact true when the model is a linear GCN.
The analysis is used to design a new dropout method, DropSens, which makes the
dropout node-dependent in such a way that it reduces over-squashing. DropSens is
shown to be more effective in enabling long-range interactions in GCNs than
alternative graph rewiring methods.

**Questions:**

- Do you mean to include a kronecker delta ($\delta_{ij}$) in Eq. (3.3)?
- Can you provide more experiments with a wider variety of datasets? Only 3
  homophilous and 3 heterphilous datasets is lacking.
- Can you provide an explanation as to why you believe DropSens has an advantage
  over graph rewiring methods?
- Can you speculate (or even better, evaluate) on a sensible version of DropSens for GIN/GAT?

**Ethical Concerns:**

["NO or VERY MINOR ethics concerns only"]

**Final Justification:**

I believe this is a clearly written paper, with the proposed DropSens method well motivated and producing benefits in practice regarding long-range interactions. I thank the authors for running more benchmarks

While the reviewers addressed the majority of my points, I keep my rating because the method is weaker for the other common architectures (GAT/GIN) and there doesn't seem to be any obvious recourse here.

**Limitations:**

Authors should mention a small number of 'real world' datasets as a limitation

**Paper Formatting Concerns:**

No concerns

**Quality:**

3

**Strengths And Weaknesses:**

Strengths:

- The paper provides a clear and sufficient background for understanding the
  problem of over-squashing.
- The proposed method (GCN+DropSens) is clear and targeted, and provides obvious improvements
  over the GCN+Rewiring in the node-classification setting.
- The work provides a fairly convincing explanation as to why dropout methods
  damage long-range abilities for GCNs.

Weaknesses:

- While performance improvements are clear for node-classification tasks,
  improvements are less clear on the graph-classification tasks
  (Table 2 (b)). For example, DropSens underperforms by $\sim$10 points on
  Mutag, $\sim 15$ points on IMDb
- DropSens only performs well on GCN, and not on GAT or GIN.
- The paper does not propose an alternative for these GAT/GIN.
- The empirical analysis should include more datasets.
- Empirical results from the paper are not clear from the abstract (i.e. that DropSens only works well on GCN).

Overall the results are lacking, and the paper could be drastically improved by adding more datasets and/or an improvement for GIN.

Minor issues:

- Some notations are not clearly defined. For example, ND/DE in the
  expectations (line 195, line 145), difference between P and \dot{P} could be
  clearer, and $E_{M^i}$ is not properly defined.
- A comparison between DropSens and the other dropout variants would be helpful;
  perhaps this could be done by include the best dropout method in Table 2.
- Missing error bars in Table 2, Figure 3

---

> ### Author Rebuttal · Authors · 2025-07-26
>
> Thank you very much for the insightful summary, for recognizing the strengths of our work, and for the overall feedback!
>
> Before we address your concerns, we wanted to bring to your attention that we had accidentally omitted the DropSens configuration \$(c, q\_{\max}) = (0.8, 0.3)\$ from our original hyperparameter sweep. This has now been included, and the results have been updated accordingly. The overall findings remain unchanged, with the exception that **DropSens now ranks 3rd on the Proteins dataset**.
>
> ---
>
> Kindly find our responses to the concerns raised:
>
> > While performance improvements are clear for node-classification tasks, improvements are less clear on the graph-classification tasks (Table 2 (b)). For example, DropSens underperforms by $\sim 10$ points on Mutag, $\sim 15$ points on IMDb
>
> Thank you for highlighting this. We agree that DropSens underperforms the best rewiring method on the two datasets you mention. That said, we would like to point out that **DropSens achieves the best performance on 3 out of the 6 graph classification tasks, and ranks 3rd on another**. Note that in the case of the IMDb dataset, *DIGL exhibits unusually strong performance*, even outperforming several more recent graph rewiring methods such as SDRF, FoSR, and GTR. While we are unsure of the exact reason for this, we do observe that **DropSens remains competitive with the rest of the baselines on this dataset**.
>
> > DropSens only performs well on GCN, and not on GAT or GIN.
>
> > The paper does not propose an alternative for these GAT/GIN.
>
> > Can you speculate (or even better, evaluate) on a sensible version of DropSens for GIN/GAT?
>
> Thank you &ndash; we agree that this is an important limitation, and we've acknowledged it in our conclusion.
>
> DropSens was specifically derived for GCN-style architectures, where the edge weights used in message aggregation are simple functions of node degree. **Extending it to other architectures raises nontrivial challenges**:
> 1. GIN uses constant edge weights, so the sensitivity of a node to its neighbors simplifies to $(1 - q)||\mathbf{W}||$ for dropping probability $q$. **Enforcing a fixed information preservation ratio $c$ leads to $q = 1 - c$, which is equivalent to DropEdge**. This limitation arises because GIN’s aggregation is insensitive to the local graph structure, making a principled variant of DropSens unnecessary.
> 2. GAT, in contrast, uses feature-dependent attention weights that vary at every layer and iteration. This means **a DropSens-style approach would require recomputing edge masks dynamically in each iteration and each layer**, negating the simplicity we aim for. Moreover, the presence of softmax attention makes a closed-form derivation of sensitivity intractable (if at all possible).
>
> In short, while DropSens is not directly applicable to GIN or GAT, **this is a consequence of their architectural design, not a shortcoming of the method per se**. We do not propose alternatives for these architectures in this paper, but see this as an opportunity for future work &ndash; developing principled dropout strategies tailored to different message-passing schemes.
>
> Does this address your concern regarding the architectural specificity of DropSens and the lack of proposed variants for GAT and GIN?
>
> > The empirical analysis should include more datasets.
>
> > Can you provide more experiments with a wider variety of datasets? Only 3 homophilous and 3 heterphilous datasets is lacking.
>
> Thank you for the suggestion. We used **a total of 14 datasets in our empirical evaluation &ndash; one synthetic graph regression dataset, 7 real-world node classification datasets and 6 graph-classification ones**. The node classification datasets were carefully chosen to span both short-range and long-range tasks, in line with prior work studying over-squashing effects. The graph classification datasets follow the benchmark selection used in recent influential work on rewiring strategies. While we agree that broader empirical coverage is always desirable, we believe our current set strikes a strong balance between diversity and relevance for the problem setting.
>
> That said, **we are aiming to share a comparison between NoDrop, DropEdge and DropSens on the PeptidesStruct graph-regression dataset from the Long Range Graph Benchmark** by the end of discussion period. We would appreciate your patience as we work on it.
>
> > Empirical results from the paper are not clear from the abstract (i.e. that DropSens only works well on GCN).
>
> Thank you for this feedback &ndash; **we’ve updated the abstract to clarify that DropSens is specifically designed for GCNs, and that its empirical effectiveness is primarily demonstrated in that setting**:
>
> "*To address this, we introduce DropSens, a sensitivity-aware variant of DropEdge, which is developed following the message-passing scheme of GCN. [...] DropSens with GCN consistently outperforms graph rewiring techniques designed to mitigate over-squashing, suggesting that simple, targeted modifications can substantially improve a model's ability to capture long-range interactions.*"
>
> > Some notations are not clearly defined. For example, ND/DE in the expectations (line 195, line 145), difference between P and \dot{P} could be clearer, and $E_{M_i}$ is not properly defined.
>
> Thank you for pointing this out &ndash; **we’ve made the following clarifications to address these notational issues**:
> 1. The subscripts ND and DE now explicitly refer to NoDrop and DropEdge models, respectively.
> 2. We’ve added a footnote clarifying that $\dot{\mathbf{P}}$ and $\ddot{\mathbf{P}}$ denote the expected propagation matrices under the asymmetric and symmetric propagation rules, respectively.
> 3. To reduce ambiguity, we’ve renamed $\mathbf{P}$ &ndash; previously used for the transition matrix of a uniform random walk &ndash; to $\mathbf{T}$.
> 4. We’ve also defined the expectation $\mathbb{E}_{\mathbf{M}_i}$ over edge masks, right before Equation 3.2.
>
> Does this address your concerns regarding notation clarity?
>
> > A comparison between DropSens and the other dropout variants would be helpful; perhaps this could be done by include the best dropout method in Table 2.
>
> Thank you for the suggestion. We agree that a broader comparison with other dropout variants could be valuable, especially to situate DropSens in the wider space of dropping methods. Accordingly, we have added the performance rankings for dropout methods in Appendix G.1 (Table 6). Notably, **DropSens ranks 1stin $11/38 \approx 29\\%$ of dataset$\times$model combinations, and places within the top-3 in $21/38 \approx 55\\%$ cases**, highlighting its efficacy across a broad range of settings.
>
> Just to clarify our scope and intent: our goal in introducing DropSens was not to establish a new state-of-the-art dropout method, but rather to demonstrate that *techniques originally designed for mitigating over-smoothing can be adapted to also address over-squashing in a principled way*. For this reason, our comparisons focus on DropEdge and NoDrop as representative baselines.
>
> > Missing error bars in Table 2, Figure 3
>
> Thank you for the observation. We omitted error bars from Table 2 primarily due to space constraints. Additionally, since **Table 2 already reports p-values, which directly address statistical significance**, we felt this conveyed the relevant information more concisely. For Figure 3, error bars are non-trivial to compute since we report relative improvement. Instead, **we have included mean difference and the standard error** as text above the bars, in the revised version to reflect variability across runs:
>
> | Cora | CiteSeer | PubMed | Chameleon | Squirrel | TwitchDE | Mutag | Proteins | Enzymes | Reddit | IMDb | Collab |
> | :-: | :-: | :-: | :-: | :-: | :-: | :-: | :-: | :-: | :-: | :-: | :-: |
> | $-0.11 \pm 0.07$ | $-0.45 \pm 0.13$ | $-0.00 \pm 0.06$ | $1.70 \pm 0.57$ | $0.36 \pm 0.21$ | $0.08 \pm 0.15$ | $2.80 \pm 1.98$ | $1.41 \pm 0.82$ | $1.31 \pm 1.19$ | $4.71 \pm 0.99$ | $0.04 \pm 1.03$ | $-0.67 \pm 0.84$ |
>
> Does this address your concern?
>
> > Do you mean to include a kronecker delta ($\delta_{ij}$) in Eq. (3.3)?
>
> Yes, thank you for catching that. We have specified that Equations 3.3 and 3.4 hold if and only if $(j \to i) \in \mathcal{E}$.
>
> > Can you provide an explanation as to why you believe DropSens has an advantage over graph rewiring methods
>
> Thank you for this question. We want to clarify that we do not claim DropSens to have an advantage over graph rewiring methods. Our goal was to demonstrate that a simple modification to DropEdge can lead to consistent performance gains, particularly by taking sensitivity to distant nodes into account.
>
> That said, we believe DropSens performs competitively in practice for two reasons:
> 1. It **addresses both over-smoothing and over-squashing simultaneously**, rather than focusing on just one of the problems, and
> 2. As a stochastic method, it **introduces a regularization effect** that improves generalization.

---

> > ### Author Response · Authors · 2025-08-03
> > **Request for Feedback**
> >
> > We humbly ask if our rebuttal has helped clarify your concerns. If any points remain unclear or unaddressed, please don’t hesitate to let us know &ndash; we’d be glad to clarify them further before the discussion period ends on August 6th.

---

> > > ### Comment · Reviewer_5aYk · 2025-08-05
> > >
> > > Thank you for taking the time to respond to my review. My points have been mostly addressed.
> > >
> > > >Does this address your concern regarding the architectural specificity of DropSens and the lack of proposed variants for GAT and GIN?
> > >
> > > Is it possible to document
> > > these insights in the limitations section? These are helpful observations for understanding how DropSens fits in the broader context.
> > >
> > > > we believe our current set strikes a strong balance between diversity and relevance for the problem setting.
> > >
> > > I disagree --- I do not think benchmarking more datasets is an unreasonable ask. Doing more than prior work would be a big improvement. I look forward to the benchmarking of
> > > PeptidesStruct graph-regression dataset. For this reason I won't be increasing my score.

---

> ### Author Response · Authors · 2025-08-05
>
> Thank you again for your thoughtful feedback.
>
> Of course, we will make sure to document the current limitations of DropSens in the revised manuscript, including why adapting it to architectures like GIN and GAT is non-trivial. We hope these observations can help guide future work toward extending DropSens-style approaches to a broader range of models.
>
> We didn’t mean to imply that benchmarking additional datasets is an unreasonable request &ndash; rather, **we aimed to strike a balance between coverage and relevance *given space and resource constraints***. That said, we understand if concerns about dataset diversity remain. In this regard, we’d like to point you to the new results for NoDrop, DropEdge, and DropSens on PeptidesStruct, shared in the comment titled "***Updated DropSens Rankings and Results on the LRGB Dataset***".
>
> We truly appreciate your engagement and hope our responses have clarified our position.

---

> > ### Comment · Reviewer_5aYk · 2025-08-05
> >
> > Thank you for providing the further results, these will improve the completeness of the paper.
> >
> > The issue of experimental setup is slightly worrying
> > >We realized that our experimental setup was slightly different compared to the one used by Black et al. (2023); Karhadkar et al. (2023),
> >
> > Are you able to confirm that you can reproduce figures in the referenced works (e.g. for vanilla or the other Drop- methods)?

---

> > > ### Author Response · Authors · 2025-08-06
> > >
> > > Thank you for pointing this out! We understand how differences in experimental setup can raise concerns.
> > >
> > > To clarify, **the only difference lies in the training duration and stopping criterion**:
> > >
> > > * In our original setup, we trained for a fixed 300 epochs and reported the test accuracy corresponding to the best validation accuracy.
> > > * In contrast, Black et al. (2023) and Karhadkar et al. (2023) trained for up to 100,000 epochs with early stopping, halting when there was less than 1% improvement in validation accuracy over the last 100 epochs, and then reported the final test accuracy.
> > >
> > > We believe **this early stopping criterion is problematic, as it implicitly demands greater accuracy gains as training progresses**. For instance, with 10% accuracy, a 0.1% gain is sufficient; at 90%, a 0.9% gain is required to continue training. This penalizes progress at later stages, when improvements naturally plateau, and results in premature convergence. In fact, we observed that in many cases, training stopped well before 300 epochs under their setup.
> > >
> > > We have updated our implementation to match their stopping protocol, so the comparison is now aligned and fair. That said, we respectfully believe their criterion leads to suboptimal training in many cases.
> > >
> > > If the concern is about selective reporting, we want to emphasize that while DropSens improved its rank on 5 datasets and dropped only on CiteSeer, the **raw accuracy actually decreased on Cora, CiteSeer, PubMed, Chameleon, Squirrel, IMDb, and Collab** with the updated setup. This reflects our commitment to transparency and fair comparison, even when it doesn't favor our method.
> > >
> > > ---
> > >
> > > Since Black et al. (2023); Karhadkar et al. (2023) do not evaluate dropout-based methods, a direct comparison is unfortunately not possible. However, we can still compare against their baseline, which corresponds to the NoDrop setting in our work. We are in the process of collecting these results under the updated experimental setup and would appreciate your patience in the meantime.

---

> > > > ### Author Response · Authors · 2025-08-07
> > > >
> > > > Hi again, and thank you for your continued engagement.
> > > >
> > > > We have run the experiments on graph classification tasks using the baseline model setup from Karhadkar et al. (2023) &ndash; that is, no rewiring and feature dropout with probability 0.5, which they apply across all rewiring methods. Below, we present a comparison between their reported results and ours under this setup:
> > > >
> > > > | | Mutag | Proteins | Enzymes | Reddit | IMDb | Collab |
> > > > | :-: | :-: | :-: | :-: | :-: | :-: | :-: |
> > > > | **Theirs** | $72.15$ | $70.98$ | $27.67$ | $68.26$ | $49.77$ | $33.78$ |
> > > > | **Ours** | $72.00$ | $66.52$ | $25.08$ | $67.77$ | $50.95$ | $32.99$ |
> > > >
> > > > **The results are broadly consistent across datasets, suggesting that our implementation aligns well with theirs**. We note a few minor differences:
> > > > 1. Their results are averaged over 100 independent runs, while ours are based on 20 runs due to the limited time available during the rebuttal period.
> > > > 2. For binary classification tasks, their implementation uses a two-dimensional output layer, while we use a single output logit. While both approaches are theoretically equivalent, they may lead to slight differences in optimization trajectories.
> > > > 3. They apply feature dropout after message passing, whereas we apply it to the layer input, which is the more standard approach. While this distinction is not directly relevant to DropSens (which does not use feature dropout), it may affect the comparison presented above. To the best of our knowledge, however, it does not significantly alter the model.
> > > >
> > > > We hope this comparison helps clarify the alignment between our implementation and theirs. Please let us know if further details would be helpful.

---

> > > > > ### Comment · Reviewer_5aYk · 2025-08-08
> > > > >
> > > > > Indeed, selective reporting would be a concern; your methodology sounds reasonable, however. Do you have error bars to show that these figures are/are not significantly different? I assume the differences are really mostly due to your point #2 and/or #3 above though.
> > > > >
> > > > > Are you able to document the methodology you used, and how they are different to Karhadkar et al. (2023) in the appendix of your paper? A reference to show that applying feature dropout to the layer input is the more standard approach would help too.

---

> > > > > > ### Author Response · Authors · 2025-08-08
> > > > > >
> > > > > > Thank you for your reply!
> > > > > >
> > > > > > > Indeed, selective reporting would be a concern
> > > > > >
> > > > > > Could you please clarify what you are referring to here? In the main paper, we have reported results on all datasets used in Topping et al. (2022) and Karhadkar et al. (2023). For comparing our implementation with Karhadkar et al. (2023), we evaluate the single method common to both work, i.e. feature dropout with probability 0.5, across all the datasets they test.
> > > > > >
> > > > > > > Do you have error bars to show that these figures are/are not significantly different?
> > > > > >
> > > > > > Yes, our apologies for not including them earlier. Here you go:
> > > > > >
> > > > > > | | Mutag | Proteins | Enzymes | Reddit | IMDb | Collab |
> > > > > > | :-: | :-: | :-: | :-: | :-: | :-: | :-: |
> > > > > > | **Theirs** | $72.15 \pm 2.44$ | $70.98 \pm 0.74$ | $27.67 \pm 1.16$ | $68.26 \pm 1.10$ | $49.77 \pm 0.82$ | $33.78 \pm 0.49$ |
> > > > > > | **Ours** | $72.00 \pm 9.80$ | $66.52 \pm 4.43$ | $25.08 \pm 5.25$ | $67.77 \pm 3.52$ | $50.95 \pm 5.33$ | $32.99 \pm 0.46$ |
> > > > > >
> > > > > > > Are you able to document the methodology you used, and how they are different to Karhadkar et al. (2023) in the appendix of your paper? A reference to show that applying feature dropout to the layer input is the more standard approach would help too.
> > > > > >
> > > > > > Yes, of course, we will make sure to include it in the camera-ready version!

---

> > > > > > > ### Comment · Reviewer_5aYk · 2025-08-08
> > > > > > >
> > > > > > > I was simply referring to your early comment ("If the concern is about selective reporting, we want to..."). I am not raising it as an issue! I agree that your work has a clear methodology.

---

> > > > > > > > ### Author Response · Authors · 2025-08-08
> > > > > > > >
> > > > > > > > Oh, understood! Apologies for the misunderstanding!

---

### Official Review · Reviewer_SqpF · 2025-06-26

**Clarity:** 4
**Significance:** 3
**Originality:** 3
**Rating:** 4
**Confidence:** 3

**Summary:**

This paper analyses how popular random‐dropping techniques for deep graph neural networks (GNNs) --- DropEdge, DropNode, DropAgg, DropGNN and DropMessage---affect over-squashing, a bottleneck that blocks long-range information flow. By computing the expected Jacobian of linear and nonlinear GCNs under random edge masks, the authors prove that DropEdge-style methods shrink sensitivity between distant nodes at an exponential rate, while marginally increasing sensitivity to immediate neighbours. To mitigate this, they introduce DropSens, which chooses a per-edge drop probability that preserves a user-specified fraction c of message strength, thus keeping long-range sensitivity constant while dropping the same number of edges. Experiments on SyntheticZINC, six node-classification benchmarks (homophilic vs heterophilic) and six graph-classification datasets show: (i) random dropping indeed helps short-range tasks but hurts or fails to help long-range ones; (ii) DropSens consistently outperforms or matches recent graph-rewiring baselines on long-range node tasks and is competitive on graph-level tasks while remaining projection-free.

**Questions:**

1. Please report GPU wall-clock time and memory for DropSens vs DropEdge and two rewiring baselines on the Squirrel (heterophilic) dataset with a 32-layer GCN until convergence to a fixed validation loss.

2. Provide an ablation on SyntheticZINC varying the preserved-information parameter $c\in\{0.7,0.8,0.9,0.95\}$; show accuracy and number of edges kept.

3. Extreme degree graphs How does DropSens behave on power-law graphs where a few hubs dominate degrees? Include an experiment or theoretical bound on maximum drop probability in that regime.

**Ethical Concerns:**

["NO or VERY MINOR ethics concerns only"]

**Final Justification:**

Thank you the author(s) for your clarification on my previous misunderstanding, taking my suggestion into account and completing several extra experiments which can be revised into the new version of the paper. I have no more questions, and at this stage I support the acceptance of the paper.

**Limitations:**

The authors candidly state that the theory assumes standard message-passing with known parallel transport and that DropSens must be tailored to the propagation rule; extending it to attention or transformer-style layers is non-trivial . They also acknowledge that experiments are small-scale and limited to a few datasets.

**Paper Formatting Concerns:**

None noted. The paper adheres to NeurIPS formatting guidelines.

**Quality:**

3

**Strengths And Weaknesses:**

Strengths:

1. The theoretical contribution is clear and original: the authors give the first explicit sensitivity decay analysis for edge-dropping schemes and extend it to nonlinear MPNNs, filling a gap in the literature on over-squashing.

2. The proposed DropSens is simple, requires no extra model parameters, and is derived directly from the theory, illustrating good “analysis-to-algorithm” design.

3. Empirical evaluation spans synthetic and real datasets, uses three backbone architectures (GCN, GIN, GAT) and includes statistical significance tests and effect sizes, lending credibility to the claims . The authors also release pseudocode and discuss computational shortcuts, supporting reproducibility.

Weaknesses:

1. Empirical scope is still modest: all large-scale experiments run on CPUs with no wall-clock comparisons, so the claimed training-speed advantage of DropSens remains anecdotal.

2. DropSens relies on in-degree–specific formulas and was tuned for GCN; results for GIN and GAT are weaker, underlining limited architecture generality.

3. The paper does not explore the effect of the preservation hyper-parameter c nor the sensitivity of DropSens to degree distribution outliers.

---

> ### Author Rebuttal · Authors · 2025-07-26
>
> Thank you very much for the insightful summary, for recognizing the strengths of our work, and for the overall feedback!
>
> Before we address your concerns, we wanted to bring to your attention that we had accidentally omitted the DropSens configuration \$(c, q\_{\max}) = (0.8, 0.3)\$ from our original hyperparameter sweep. This has now been included, and the results have been updated accordingly. The overall findings remain unchanged, with the exception that **DropSens now ranks 3rd on the Proteins dataset**.
>
> ---
>
> Kindly find our responses to the concerns raised:
>
> > Empirical scope is still modest: all large-scale experiments run on CPUs with no wall-clock comparisons, so the claimed training-speed advantage of DropSens remains anecdotal.
>
> Thank you for pointing this out. To clarify, the hardware setup used for all large-scale experiments is described in Appendix E: **we use 4$\times$ NVIDIA GeForce GTX TITAN X GPUs (12 GB VRAM each)**.
>
> You're absolutely right that wall-clock runtimes were not reported in the initial submission. We have now addressed this:
>
> **We first show the preprocessing time required to compute edge-dropping probabilities for DropSens as a function of node in-degree**, since the *cost of solving for $q_i$ grows with $d_i$*. We plot results for $d_i = 1, 2, \ldots, 100$, averaged over $c = 0.1, 0.2, \ldots, 0.9$ in Figure 6b; for readability, we report values only for $d_i = 1, 2, 5, 10, 20, 50, 100$ in the table below.
>
> |  | 1 | 2 | 5 | 10 | 20 | 50 | 100 |
> | :-: | :-: | :-: | :-: | :-: | :-: | :-: | :-: |
> | Exact Computation time $\times 10^{-2}$ | $3.316 \pm 5.961$ | $1.904 \pm 0.952$ | $2.713 \pm 0.728$ | $4.011 \pm 1.514$ | $10.181 \pm 6.186$ | $43.803 \pm 23.147$ | $169.897 \pm 52.593$ |
> | Approximation Computation Time $\times 10^{-6}$ | $3.253 \pm 0.568$ | $1.316 \pm 0.131$ | $1.097 \pm 0.021$ | $1.117 \pm 0.025$ | $1.113 \pm 0.013$ | $1.089 \pm 0.062$ | $1.103 \pm 0.011$ |
>
> **We now compare the initialization and sampling time of DropSens ($c=0.8, q_{\max}=0.5$) compared to sampling time of DropEdge**, averaged over 10 runs, isolating the runtime difference since sampling is the only point where the two methods differ computationally$.^1$ We have added this comparison in Figure 7, and also present the results in the table below, with all entries reported in milliseconds:
>
> |  | Cora | CiteSeer | PubMed | Chameleon | Squirrel | TwitchDE | Actor | Mutag | Proteins | Enzymes | Reddit | IMDb | Collab |
> | :-: | :-: | :-: | :-: | :-: | :-: | :-: | :-: | :-: | :-: | :-: | :-: | :-: | :-: |
> | DropSens Initialization | $104$ | $67$ | $74$ | $89$ | $85$ | $83$ | $43$ | $49$ | $68$ | $58$ | $69$ | $132$ | $483$ |
> | DropSens Sampling | $17$ | $13$ | $51$ | $50$ | $61$ | $44$ | $16$ | $13$ | $39$ | $27$ | $92$ | $58$ | $1163$ |
> | DropEdge Sampling | $1$ | $1$ | $0$ | $1$ | $0$ | $0$ | $0$ | $1$ | $0$ | $1$ | $1$ | $1$ | $6$ |
>
> As expected, **DropSens sampling is more expensive** due to the need to compute edge-wise dropping probabilities per graph. That said, **this step accounts for only a small fraction of overall training time and was never the bottleneck in our experiments**
>
> ***Note**: Runtimes might vary form server to server, but we expect the trends to be similar.*
>
> We do want to clarify that **we do not claim a training-speed advantage for DropSens**. On the contrary, due to its preprocessing step, it may be marginally slower than DropEdge. We made this explicit in Appendix E.3, where we also provide a low-cost approximation for solving Equation 4.1.
>
> Does this address your concern about the empirical scope and runtime reporting?
>
> > DropSens relies on in-degree–specific formulas and was tuned for GCN; results for GIN and GAT are weaker, underlining limited architecture generality.
>
> Thank you &ndash; we agree that this is an important limitation, and we've acknowledged it in our conclusion.
>
> DropSens was specifically derived for GCN-style architectures, where the edge weights used in message aggregation are simple functions of node degree. **Extending it to other architectures raises nontrivial challenges**:
> 1. GIN uses constant edge weights, so the sensitivity of a node to its neighbors simplifies to $(1 - q)\|\|\mathbf{W}\|\|$ for dropping probability $q$. **Enforcing a fixed information preservation ratio $c$ leads to $q = 1 - c$, which is equivalent to DropEdge**. This limitation arises because GIN’s aggregation is insensitive to the local graph structure, making a principled variant of DropSens unnecessary.
> 2. GAT, in contrast, uses feature-dependent attention weights that vary at every layer and iteration. This means **a DropSens-style approach would require recomputing edge masks dynamically in each iteration and each layer**, negating the simplicity we aim for. Moreover, the presence of softmax attention makes a closed-form derivation of sensitivity intractable (if at all possible).
>
> In short, while DropSens is not directly applicable to GIN or GAT, **this is a consequence of their architectural design, not a shortcoming of the method per se**. We see this as an opportunity for future work &ndash; developing principled dropout strategies tailored to different message-passing schemes.
>
> Does this address your concern regarding the architectural generality of DropSens?
>
> > The paper does not explore the effect of the preservation hyper-parameter c nor the sensitivity of DropSens to degree distribution outliers.
>
> Thank you for highlighting this. You’re absolutely right that we did not explicitly discuss the role of the preservation hyper-parameter $c$ in the *main text*. While Figure 6a does compare the dropping probabilities for different values of $c$, we have now **added a discussion at the end of Section 4 to clarify how varying $c$ affects the dropping probabilities** for different in-degrees.
>
> Regarding sensitivity to degree outliers, DropSens assigns dropping probabilities based on the degree of the target node, which means that edges connected to extremely high-degree nodes may receive disproportionately high dropping probabilities. **To prevent overly aggressive edge-removal in such cases, we clip the computed probabilities**, as described under "DropSens Configurations" in Appendix E.2. We have also added this implementation detail at the end of Section 4 to make this design choice explicit:
>
> "*Note that higher values of $c$ encourage lower $q_i$, while lower values permit $q_i$ to take higher values; see Figure 6a. Since this can result in abnormally high dropping probabilities, we clip the value of $q_i$ in our experiments using another hyperparameter, $q_{\max}$; see details in Appendix E.2.*"
>
> Does this address your concern about both the role of $c$ and the robustness to degree outliers?
>
> > Provide an ablation on SyntheticZINC varying the preserved-information parameter $c \in 0.7, 0.8, 0.9, 0.95$; show accuracy and number of edges kept.
>
> Thank you for the suggestion &ndash; we agree that this ablation would offer valuable insight. As a first step, **we evaluate DropSens with $c = 0.9, 0.65$**, which preserve approximately the same number of edges in expectation ($\approx 402$k, $248$k, respectively) as DropEdge with $q = 0.2, 0.5$ ($\approx 400$k, $250$k, respectively).
>
> Kindly find the results for DropSens with $c = 0.9$ below, where we observe that it improves on its DropEdge counterpart, $q=0.2$; we will report on $c = 0.65$ soon.
>
> **Train MAE $\times 10^{-2}$** (lower is better)
> | Method | $\alpha=0.1$ | $0.2$ | $0.3$ | $0.4$ | $0.5$ | $0.6$ | $0.7$ | $0.8$ | $0.9$ | $1.0$ |
> | :-:| :-: | :-: | :-: | :-: | :-: | :-: | :-: | :-: | :-: | :-: |
> | DropSens ($c=0.9$) | $6.062$ | $6.831$ | $7.203$ | $7.297$ | $8.204$ | $7.157$ | $6.605$ | $6.552$ | $6.545$ | $6.807$ |
> | DropEdge ($q=0.2$) | $6.369$ | $7.016$ | $7.389$ | $7.339$ | $7.455$ | $7.374$ | $6.993$ | $7.420$ | $6.588$ | $7.213$ |
> | DropSens ($c=0.65$) |  |  |  |  |  |  |  |  |  |  |
> | DropEdge ($q=0.5$) | $8.218$ | $8.954$ | $9.173$ | $9.210$ | $9.109$ | $9.076$ | $8.834$ | $8.807$ | $8.754$ | $9.067$ |
>
> **Test MAE $\times 10^{-2}$** (lower is better)
> | Method | $\alpha=0.1$ | $0.2$ | $0.3$ | $0.4$ | $0.5$ | $0.6$ | $0.7$ | $0.8$ | $0.9$ | $1.0$ |
> | :-:| :-: | :-: | :-: | :-: | :-: | :-: | :-: | :-: | :-: | :-: |
> | DropSens ($c=0.9$) | $6.739$ | $7.221$ | $7.830$ | $8.156$ | $9.134$ | $8.221$ | $7.442$ | $7.118$ | $6.760$ | $6.355$ |
> | DropEdge ($q=0.2$) | $7.096$ | $7.496$ | $8.262$ | $8.567$ | $8.781$ | $8.801$ | $8.168$ | $8.099$ | $7.066$ | $7.221$ |
> | DropSens ($c=0.65$) |  |  |  |  |  |  |  |  |  |  |
> | DropEdge ($q=0.5$) | $13.326$ | $13.459$ | $14.805$ | $15.215$ | $14.753$ | $14.501$ | $14.101$ | $13.567$ | $12.566$ | $13.381$ |
>
> > How does DropSens behave on power-law graphs where a few hubs dominate degrees? Include an experiment or theoretical bound on maximum drop probability in that regime.
>
> Thank you for raising this important point. You're absolutely right that on power-law graphs, **a small number of high-degree hubs can lead DropSens to assign extremely high dropping probabilities, potentially close to $1$**. Since DropSens does not impose a theoretical bound on dropping probability (except the vacuous bound, $1$), this behavior is expected when in-degrees are very large.
>
> To mitigate this, **we cap the dropping probabilities using a hyper-parameter $q_{\max}$, which ensures that even high-degree nodes retain a minimum level of connectivity**. This clipping mechanism is already implemented in our experiments, but we realize this detail was not clearly communicated. **We have now added a clarification at the end of Section 4 to make this design choice explicit**.
>
> Does this address your concern about DropSens on power-law graphs and the lack of a theoretical bound?
>
> ---
>
> **Footnotes**:
>
> 1. For graph classification tasks, edge masks are computed in one go (instead of one mini-batch at a time, as in practice).

---

> > ### Comment · Reviewer_SqpF · 2025-07-31
> > **Thanks**
> >
> > Thank you the author(s) for taking my suggestion into account and completing several extra experiments which can be revised into the new version of the paper.   I have no more questions, and support the acceptance of the paper.

---

> ### Author Response · Authors · 2025-08-01
> **Updating the Rating**
>
> Thank you very much for actively engaging in the review process. If you have no further concerns remaining unresolved, we would appreciate it if you would consider updating your rating!

---

> ### Author Response · Authors · 2025-08-01
> **Updated Results for SyntheticZINC**
>
> We present the full results for DropSens on SyntheticZINC below (bold indicates better performance in the comparisons 1. DropSens ($c = 0.9$) v.s. DropEdge ($q = 0.2$), and 2. DropSens ($c = 0.65$) v.s. DropEdge ($q = 0.5$)):
>
> **Train MAE $\times10^{-2}$** (lower is better)
>
> | Method | $\alpha = 0.1$ | $0.2$ | $0.3$ | $0.4$ | $0.5$ | $0.6$ | $0.7$ | $0.8$ | $0.9$ | $1.0$ |
> | :-: | :-: | :-: | :-: | :-: | :-: | :-: | :-: | :-: | :-: | :-: |
> | DropSens ($c = 0.9$) | $\mathbf{6.062}$ | $\mathbf{6.831}$ | $\mathbf{7.203}$ | $\mathbf{7.297}$ | $8.204$ | $\mathbf{7.157}$ | $\mathbf{6.605}$ | $\mathbf{6.552}$ | $\mathbf{6.545}$ | $\mathbf{6.807}$ |
> | DropEdge ($q = 0.2$) | $6.369$ | $7.016$ | $7.389$ | $7.339$ | $\mathbf{7.455}$ | $7.374$ | $6.993$ | $7.420$ | $6.588$ | $7.213$ |
> | DropSens ($c = 0.65$) | $8.223$ | $\mathbf{8.946}$ | $\mathbf{9.138}$ | $9.311$ | $9.160$ | $\mathbf{9.013}$ | $8.852$ | $\mathbf{8.563}$ | $8.779$ | $\mathbf{8.871}$ |
> | DropEdge ($q = 0.5$) | $\mathbf{8.218}$ | $8.954$ | $9.173$ | $\mathbf{9.210}$ | $\mathbf{9.109}$ | $9.076$ | $\mathbf{8.834}$ | $8.807$ | $\mathbf{8.754}$ | $9.067$ |
>
> **Test MAE $\times10^{-2}$** (lower is better)
>
> | Method | $\alpha = 0.1$ | $0.2$ | $0.3$ | $0.4$ | $0.5$ | $0.6$ | $0.7$ | $0.8$ | $0.9$ | $1.0$ |
> | :-: | :-: | :-: | :-: | :-: | :-: | :-: | :-: | :-: | :-: | :-: |
> | DropSens ($c = 0.9$) | $\mathbf{6.739}$ | $\mathbf{7.221}$ | $\mathbf{7.830}$ | $\mathbf{8.156}$ | $9.134$ | $\mathbf{8.221}$ | $\mathbf{7.442}$ | $\mathbf{7.118}$ | $\mathbf{6.760}$ | $\mathbf{6.355}$ |
> | DropEdge ($q = 0.2$) | $7.096$ | $7.496$ | $8.262$ | $8.567$ | $\mathbf{8.781}$ | $8.801$ | $8.168$ | $8.099$ | $7.066$ | $7.221$ |
> | DropSens ($c = 0.65$) | $14.138$ | $\mathbf{13.223}$ | $\mathbf{13.704}$ | $\mathbf{15.088}$ | $\mathbf{14.699}$ | $\mathbf{14.226}$ | $\mathbf{13.784}$ | $\mathbf{12.240}$ | $\mathbf{12.166}$ | $\mathbf{10.458}$ |
> | DropEdge ($q = 0.5$) | $\mathbf{13.326}$ | $13.459$ | $14.805$ | $15.215$ | $14.753$ | $14.501$ | $14.101$ | $13.567$ | $12.566$ | $13.381$ |
>
> It is clear to see that **DropSens outperforms DropEdge in most cases, demonstrating the benefits brought in by simple sensitivity-aware modifications** to algorithms originally designed for alleviating over-smoothing and training deep GNNs.

---

### Official Review · Reviewer_mWNa · 2025-07-04

**Clarity:** 4
**Significance:** 4
**Originality:** 3
**Rating:** 5
**Confidence:** 4

**Summary:**

The paper investigates GNN-specific dropout methods and their effect on two phenomena preventing high depth in message passing GNNs, namely, oversmoothing and oversquashing. While the positive effect of such dropout methods on oversmoothing has been investigated, their effect on oversquashing and long-range interactions is underexplored. The authors use an existing sensitive analysis quantifying the dependence of a node's embedding on the initial features of distant nodes and generalize it to situations where edges are dropped randomly. This theoretical analysis suggests that higher depth under dropout does not necessarily lead to a better ability to capture long-range interaction. Experiments support the theoretical results.

**Questions:**

**Q1)** Could you clarify my questions regarding the experimental evaluation mentioned in W1?

**Ethical Concerns:**

["NO or VERY MINOR ethics concerns only"]

**Final Justification:**

The authors clarified my question regarding the interpretation of Figure 2 in the rebuttal, and I believe that the promised changes will fix the minor flaws I mentioned in my review. I keep my rating.

**Limitations:**

yes

**Quality:**

3

**Strengths And Weaknesses:**

## Strengths

**S1)** The interplay between dropout, oversmoothing, and oversquashing is highly relevant and gives new insights into the capabilities and predictive performance of GNNs.

**S2)** The paper summarizes the related work well and puts the contribution in the context of state-of-the-art techniques.

**S3)** The paper makes theoretical contributions (albeit under limitations) and shows the relevance of the theoretical results empirically. The limitations are clearly stated and discussed.

## Wekaness

**W1)** The experimental evaluation has several weaknesses and does not provide a consistent picture of the topic:
 * Figure 1(b) is difficult to read since curves overlap. In Section 4, it is claimed that the proposed method, DropSens, improves the sensitivity compared to DropEdge. However, as both lines almost overlap, I cannot follow the interpretation.
 * I do not understand why DropSens is not included in the experiments of Sections 5.1 and 5.2; it would be highly relevant to see how the method compares to other dropout methods in these experiments.
 * Figure 2 shows that NoDrop achieves the best results. Why is this dataset suitable for comparing different dropout methods?

## Minor remarks
 * In Eq. (2), the set should be the argument of the Out function.
 * Eq. (3.2) The index $\mathbf{M}^{(1)},...$ is not clear.

---

> ### Author Rebuttal · Authors · 2025-07-26
>
> Thank you very much for the insightful summary, for recognizing the strengths of our work, and for the overall feedback!
>
> Before we address your concerns, we wanted to bring to your attention that we had accidentally omitted the DropSens configuration \$(c, q\_{\max}) = (0.8, 0.3)\$ from our original hyperparameter sweep. This has now been included, and the results have been updated accordingly. The overall findings remain unchanged, with the exception that **DropSens now ranks 3rd on the Proteins dataset**.
>
> ---
>
> Kindly find our responses to the concerns raised:
>
> > Figure 1(b) is difficult to read since curves overlap. In Section 4, it is claimed that the proposed method, DropSens, improves the sensitivity compared to DropEdge. However, as both lines almost overlap, I cannot follow the interpretation.
>
> Thank you for pointing this out &ndash; we agree that the overlapping curves in Figure 1(b) make the comparison difficult to interpret. To address this, **we have revised Figure 1(b) by embedding a subplot** that explicitly shows the **ratio of influence at different distances between DropSens and DropEdge**. This makes the differences between the two methods much more visible. Kindly find the entries of the plot below, with the columns denoting shortest distance between nodes:
>
> | $d_G(j,i) =$ | 0 | 1 | 2 | 3 | 4 | 5 | 6 |
> | :-: | :-: | :-: | :-: | :-: | :-: | :-: | :-: |
> |  | 0.7312 | 1.0247 | 1.0668 | 1.1233 | 1.0938 | 1.1548 | 1.2283 |
>
> Compared to DropEdge, DropSens decreases sensitivity of a node's final representations to its input features ($d=0$), and increases it to other nodes ($d \geq 1$). **This effect is especially significant for distant nodes, suggesting that DropSens should be better at modeling LRIs than DropEdge**.
>
> We hope this modification will make the comparison and the interpretation in Section 4 clearer.
>
> > I do not understand why DropSens is not included in the experiments of Sections 5.1 and 5.2; it would be highly relevant to see how the method compares to other dropout methods in these experiments.
>
> That makes complete sense! **We have included the results for DropSens, as well as a commentary on the results** in Section 5.3:
>
> |  | Cora | CiteSeer | PubMed | Chameleon | Squirrel | TwitchDE |
> | :-: | :-: | :-: | :-: | :-: | :-: | :-: |
> | GCN | +0.308 | +0.239 | +0.383 | +1.069 | +0.372 | -0.018 |
> | GIN | -0.022 | -0.065 | -0.666 | +0.174 | +0.330 | +0.616 |
> | GAT | +0.577 | +0.663 | -0.003 | -3.859 | -1.583 | -0.090 |
>
> |  | Mutag | Proteins | Enzymes | Reddit | IMDb | Collab |
> | :-: | :-: | :-: | :-: | :-: | :-: | :-: |
> | GCN | +1.700 | +3.161 | +0.716 | -3.670 | +1.340 | -1.818 |
> | GIN | -2.700 | -1.304 | -1.650 | -4.500 | -6.200 | -1.379 |
> | GAT | +4.300 | +1.036 | +0.897 | +0.400 | -0.760 | 0.000 |
>
> "*We now evaluate whether DropSens offers meaningful improvements over a NoDrop baseline, consistent with our analysis for other dropout methods. Results for GCN and GIN are shown in Table 1a, and those for GAT are in Table 7. The performance boost observed with DropSens is much more remarkable for GCN than for other architectures, which is unsurprising since DropSens was specifically designed to work with GCN's message-passing scheme. We also observe that while it offers significant improvement in 5/6 node classification tasks, its effect on graph classification &ndash; while positive in 4/6 datasets &ndash; is not statistically significant.*"
>
> Just to clarify our scope: **We do not claim that DropSens undoes all the negative effects of DropEdge**. Rather, we intend to demonstrate that techniques originally designed for mitigating over-smoothing can severely underperform on long-range tasks, but **they can be readily adapted to perform *better* on long-range tasks**.
>
> Does this address your concerns?
>
> > Figure 2 shows that NoDrop achieves the best results. Why is this dataset suitable for comparing different dropout methods?
>
> Thank you for this question. The SyntheticZINC dataset &ndash; introduced in Giovanni et al. (2024, Section 5.1); described further in our Appendix E.1 &ndash; was specifically designed to study how different levels of information mixing in the underlying ground-truth affect model performance. Here's a brief overview of how it is designed:
>
> 1. A desired mixing level \$\alpha \in \[0, 1]\$ is fixed.
> 2. Commute times between all node pairs in a graph are computed.
> 3. A node pair is selected whose commute time lies at the \$\alpha\$-th percentile of the graph's commute time distribution.
>
> As shown in Giovanni et al. (2024, Theorem 4.4), **over-squashing in GNNs &ndash; especially with MAX pooling for graph classification** (as is also our setup) **&ndash; is directly linked to the commute time between nodes.** Therefore, SyntheticZINC is well-suited for comparing dropout methods in this context: **it provides a controlled environment to isolate and measure the over-squashing effects of different algorithms**.
>
> Does that clarify your concern about the suitability of this dataset?
>
> > In Eq. (2), the set should be the argument of the Out function.
>
> Yes, thank you for the correction!
>
> > Eq. (3.2) The index \$\mathbf{M}^{(1)}, \ldots\$ is not clear.
>
> Fair enough, **we have now defined the variables to be the edge masks** in the preceding line. Kindly let us know in case it still remains unclear.
>
> ---
>
> **References**:
>
> Francesco Di Giovanni, T. Konstantin Rusch, Michael Bronstein, Andreea Deac, Marc Lackenby, Siddhartha Mishra, and Petar Velickovic. *How does over-squashing affect the power of GNNs?* Transactions on Machine Learning Research, 2024.

---

> > ### Author Response · Authors · 2025-08-03
> > **Request for Feedback**
> >
> > We humbly ask if our rebuttal has helped clarify your concerns. If any points remain unclear or unaddressed, please don’t hesitate to let us know &ndash; we’d be glad to clarify them further before the discussion period ends on August 6th.

---

> > > ### Comment · Reviewer_mWNa · 2025-08-04
> > >
> > > Thanks for carefully answering my questions. I believe that the promised changes will have fixed the minor flaws I mentioned in my review. The interpretation of Figure 2 is now clear to me.

---

### Author Response · Authors · 2025-08-04
**Updated DropSens Rankings and Results on the LRGB Dataset**

We thank the reviewers very much for acknowledging our results and actively engaging in the review process!

We want to present some more results, and we hope you can take these into account when assessing our contributions:

**Updated Comparisons of DropSens with graph-rewiring methods**

We realized that our experimental setup was slightly different compared to the one used by Black et al. (2023); Karhadkar et al. (2023), in that they used a lot more number of maximum epochs for training compared to us (originally 300), and instead used early stopping to determine convergence. Using the same setup, we conducted 20 runs with each hyperparameter setting of DropSens; we are working on collecting 50 samples for the best configurations for each dataset. Kindly find the performance and updated ranks of DropSens below:

**GCN**

| Cora | CiteSeer | PubMed | Chameleon | Squirrel | Actor |
| :-: | :-: | :-: | :-: | :-: | :-: |
| $84.57$ (**1st**) | $71.70$ (4th) | $83.80$ (**1st**) | $52.06$ (**1st**) | $39.33$ (**1st**) | $22.63$ (7th) |

| Mutag | Proteins | Enzymes | Reddit | IMDb | Collab |
| :-: | :-: | :-: | :-: | :-: | :-: |
| $80.75$ (**1st**) | $73.17$ (**2nd**) | $50.86$ (**1st**) | $78.50$ (**1st**) | $49.05$ (6th) | $61.16$ (**1st**) |

**GIN**

| Mutag | Proteins | Enzymes | Reddit | IMDb | Collab |
| :-: | :-: | :-: | :-: | :-: | :-: |
| $79.75$ (**2nd**) | $69.02$ (8th) | $61.71$ (**1st**) | $88.48$ (**2nd**) | $62.40$ (7th) | $67.21$ (6th) |

Summary of updates in rankings:
1. Performance with **GCN on CiteSeer** went **down from 1st to 4th**.
2. Performance with **GCN on Mutag** went **up from 7th to 1st**.
3. Performance with **GCN on Proteins** went **up from 7th to 2nd**.
4. Performance with **GCN on Mutag** went **up from 8th to 2nd**.
5. Performance with **GCN on Enzymes** went **up from 6th to 1st**.
6. Performance with **GCN on Reddit** went **up from 6th to 2nd**.

These changes put **DropSens with GCN on the podium for 2 more graph-classification datasets**, totaling 5/6 datasets. Moreover, they put **DropSens with GIN on the podium for 3 graph-classification datasets**.

---

**Performance on PeptidesStruct (LRGB) dataset**

Below we present the Mean Absolute Error (averaged over 20 runs) of the best performing configurations of DropEdge and DropSens with GCN on the PeptidesStruct dataset (using early-stopping, as above):

| NoDrop | DropSens | DropEdge |
| :-: | :-: | :-: |
| $0.4181$ | $0.4148$ | $0.4278$ |

We see that **DropEdge results in a performance decline, while DropSens outperforms the NoDrop baseline**.

We are working on collecting the full set of results for PeptidesStruct and PeptidesFunc, but in the meantime, we hope these results address any concerns about the variety of datasets used in our experiments.

---

### Author Response · Authors · 2025-08-05
**Any Remaining Concerns?**

Hi, in light of the extended discussion period, we kindly invite the reviewers to share any remaining concerns. We’d be glad to do our best to address them during this time.

---

### Note · Authors · 2025-08-15

We thank the reviewers for actively engaging in the review process, and helping us genuinely improve the paper.

Below, we summarize the edits and additions made on top of our initial submission, thanks to the reviewers' suggestions:

1. We updated the comparison of DropSens against the graph-rewiring methods designed specifically for alleviating over-squashing, to match the implementation in Karhadkar et al. (2023). This **moved DropSens from bottom 3 to top 2 on 5 datasets**, while degrading its ranking on only one dataset.

2. Upon **Reviewer mWNa's suggestions**, we have included
    - a revision of Figure 1(b) to improve its readability,
    - results for DropSens in Table 1, and added a commentary on the results.

3. Upon **Reviewer SqpF's suggestions**, we have included
    - the computation time for exact dropping probabilities under DropSens, as a function of in-degree,
    - the computation time for edge masks for the real-world datasets used in our study,
    - an explanation for why adapting DropSens to GIN and GAT is non-trivial, and hence not done in our study,
    - results for DropSens on SyntheticZINC, demonstrating that it improves on DropEdge.

4. Upon **Reviewer 5aYk's suggestions**, we have included
    - an explanation for why adapting DropSens to GIN and GAT is non-trivial, and hence not done in our study,
    - a comparison between NoDrop, DropEdge and DropSens on the PeptidesStruct graph-regression dataset from the Long Range Graph Benchmark,
    - a revision to the abstract to clarify the specificity of DropSens to GCN,
    - a comparison between DropSens and other dropping methods,
    - added error bars for Figure 3.

5. Upon **Reviewer RHiq's suggestions**, we have included
    - a sensitivity analysis comparing DropSens with NoDrop and DropEdge on homophilic and heterophilic datasets,
    - a comparison between NoDrop, DropEdge and DropSens on the PeptidesStruct graph-regression dataset from the Long Range Graph Benchmark.

We sincerely thank the reviewers' for their engagement, hope that our rebuttal has clarified their concerns, and that they have updated their ratings/confidence accordingly!

---

### Decision · Program_Chairs · 2025-09-17

**Decision:**

Accept (poster)

**Comment:**

This paper investigates the impact of dropout-based methods (DropEdge, DropNode, etc.) on over-squashing in Graph Neural Networks, revealing that while these methods address over-smoothing, they can exacerbate over-squashing by reducing sensitivity to distant nodes. The authors propose DropSens, a sensitivity-aware variant that preserves long-range interactions while maintaining the benefits of edge dropping. The work received scores of 5 (accept), 4 (borderline accept), 4 (borderline accept), and 4 (borderline accept) from four engaged reviewers. Strengths include: (1) Novel theoretical insight - The paper provides the first systematic analysis of how dropout methods affect over-squashing through sensitivity analysis, revealing an important but previously unrecognized trade-off between addressing over-smoothing and preserving long-range interactions; (2) Strong theoretical foundation - The mathematical analysis extending sensitivity measures to edge-dropping scenarios is rigorous and well-motivated, leading naturally to the DropSens algorithm design; (3) Comprehensive experimental validation - The authors conducted extensive experiments across synthetic and real-world datasets, responded thoroughly to reviewer concerns by adding results on LRGB datasets, and provided detailed computational analysis; (4) Practical contribution - DropSens offers a principled solution that maintains simplicity while addressing the identified limitations. Limitations acknowledged: (1) Architecture specificity - DropSens is primarily designed for GCN architectures, with limited applicability to GIN/GAT due to their different message-passing schemes; (2) Scope constraints - While the experimental evaluation is solid, it focuses on medium-scale datasets with some computational limitations for larger benchmarks. The authors demonstrated exceptional responsiveness during the review process, conducting additional experiments, updating experimental setups to match prior work, and providing thorough theoretical explanations. All reviewers acknowledged the quality of the responses and the authors' transparency in reporting both positive and negative results. The work makes a valuable contribution to understanding the complex interplay between different GNN training techniques and their effects on fundamental graph learning challenges.